# Weakly-Supervised Contrastive Learning for Imprecise Class Labels

**Zi-Hao Zhou** [* 1 2]  **Jun-Jie Wang** [* 1 2]  **Tong Wei** [1 2]  **Min-Ling Zhang** [1 2]

## Abstract

Contrastive learning has achieved remarkable success in learning effective representations, with supervised contrastive learning often outperforming self-supervised approaches. However, in real-world scenarios, data annotations are often ambiguous or inaccurate, meaning that class labels may not reliably indicate whether two examples belong to the same class. This limitation restricts the applicability of supervised contrastive learning. To address this challenge, we introduce the concept of "continuous semantic similarity" to define positive and negative pairs. Instead of directly relying on imprecise class labels, we measure the semantic similarity between example pairs, which quantifies how closely they belong to the same category by iteratively refining weak supervisory signals. Based on this concept, we propose a graph-theoretic framework for weakly-supervised contrastive learning, where semantic similarity serves as the graph weights. Our framework is highly versatile and can be applied to many weakly-supervised learning scenarios. We demonstrate its effectiveness through experiments in two common settings, i.e., noisy label and partial label learning, where existing methods can be easily integrated to significantly improve performance. Theoretically, we establish an error bound for our approach, showing that it can approximate supervised contrastive learning under mild conditions. The implementation code is available at https://github.com/Speechless-10308/WSC.

*Equal contribution [1]School of Computer Science and Engineering, Southeast University, Nanjing, China [2]Key Laboratory of Computer Network and Information Integration (Southeast University), Ministry of Education, China. Correspondence to: Tong Wei <weit@seu.edu.cn>.

*Proceedings of the 42nd International Conference on Machine Learning*, Vancouver, Canada. PMLR 267, 2025. Copyright 2025 by the author(s).

## 1. Introduction

In recent years, there has been a resurgence of research in contrastive learning, which has led to significant advancements in the field of representation learning (He et al., 2020; Oord et al., 2018; Chen et al., 2020; Caron et al., 2020; Zbontar et al., 2021; Khosla et al., 2020). These works share a common underlying principle: they aim to pull an anchor and its corresponding "positive" examples closer together in the embedding space, while simultaneously pushing the anchor away from a set of "negative" examples. In self-supervised contrastive learning, positive examples are generated through various data augmentation techniques applied to the original samples, thereby creating different views or representations of the same data instance. Negative examples, on the other hand, are randomly selected from different samples, as illustrated in Figure 1(a). In contrast, fully-supervised contrastive learning utilizes additional supervisory information by treating samples from the same class as positive pairs, thereby constructing multiple positive examples (Khosla et al., 2020), as shown in Figure 1(b). However, real-world supervisory information is often inaccurate and ambiguous (Lin et al., 2023; Zhang et al., 2021b; Wang et al., 2022; Zhang et al., 2022; Guo et al., 2020; 2025), manifesting as weakly-supervised information such as noisy labels (Amsaleg et al., 2017; Yan et al., 2013) and partial labels (Luo and Orabona, 2010; Chen et al., 2018; Dong et al., 2023), which cannot directly indicate class membership between samples. As a result, traditional methods fail to effectively utilize weak supervisory information, significantly limiting the applicability of supervised contrastive learning in real-world tasks.

To address the limitations of existing methods, we introduce a more general concept of positive and negative examples: continuous semantic similarity. Samples with a higher likelihood of belonging to the same category are assigned higher semantic similarity, where values of 1 and 0 correspond to positive and negative pairs in traditional contrastive learning, respectively. This can be viewed as a continuous extension of the conventional positive-negative example framework. Furthermore, as illustrated in Figure 1(c), we extend the contrastive learning objective to align the feature similarity of two samples with their corresponding semantic similarity.

Specifically, we formulate a graph-theoretic framework for

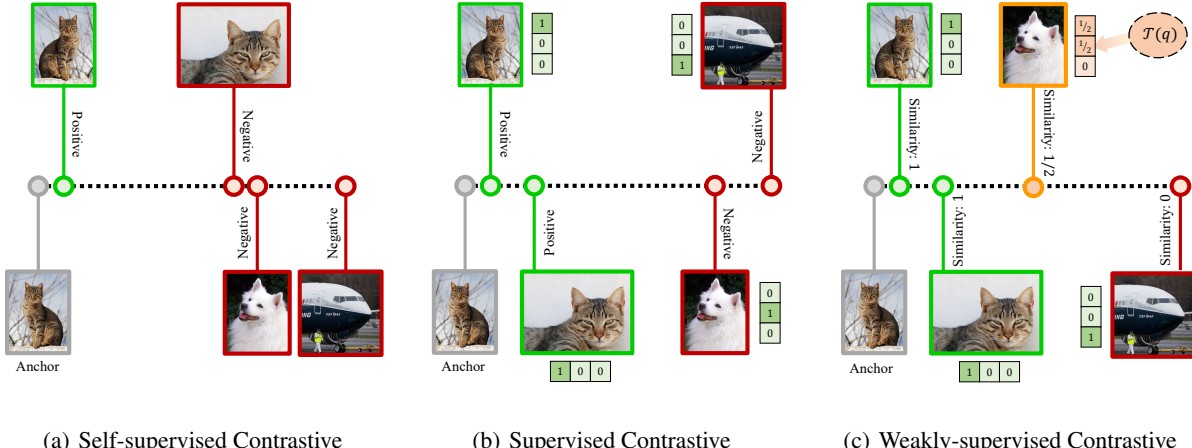

(a) Self-supervised Contrastive     (b) Supervised Contrastive     (c) Weakly-supervised Contrastive

*Figure 1.* (a) Self-supervised contrastive learning constructs positive example pairs by using different views of the same image and constructs negative example pairs by using different images. (b) Supervised contrastive learning further regards different images of the same class as additional positive example pairs. (c) Our proposed weakly-supervised contrastive learning abandons the concepts of discrete positive and negative examples.

weakly-supervised representation learning, where vertices represent augmented data points and edges are defined by their corresponding semantic similarity. The semantic similarity is constructed from two perspectives. First, data points derived from different views of the same instance are assigned a relatively high semantic similarity. Second, we leverage the corresponding weakly-supervised information to approximate whether two samples belong to the same class, thereby constructing the semantic similarity between data points from different instances after augmentation.

By modeling semantic similarities using the provided weakly-supervised information, our method is highly versatile and applicable to various weakly-supervised learning scenarios, including noisy label and partial label settings. The contrastive loss derived from our framework is straightforward to implement, stable during training, and can be seamlessly integrated as a plug-and-play component, consistently enhancing the performance of existing methods in weakly-supervised learning.

Additionally, we provide a comprehensive theoretical analysis of the method's performance, deriving the downstream error bound for feature learning using our contrastive loss under various weakly-supervised settings. The analysis shows that, under mild conditions, our framework can approximate the performance achievable with fully-supervised information, thus theoretically ensuring its effectiveness.

In summary, our main contributions are summarized below:

- We propose a unified framework for contrastive learning that leverages weakly-supervised information, including noisy and partial labels. By introducing the

concept of semantic similarity as a generalization of positive and negative examples, our framework offers new insights into how contrastive learning can effectively utilize ambiguous supervisory information.

- The loss derived from our framework is highly versatile and can be seamlessly integrated into a wide range of weakly-supervised learning settings, consistently leading to performance improvements. Notably, our approach shows substantial gains in challenging settings with high noise and partial label rates, achieving improvements of **6.8%** and **7.8%**, respectively, underscoring its robustness and effectiveness in addressing real-world weak supervision challenges.

- Our theoretical analysis offers a rigorous performance guarantee for the proposed framework, showing that it can approximate supervised contrastive learning under mild conditions and elucidating how and to what extent weakly-supervised information can enhance the effectiveness of representation learning.

## 2. Weakly-Supervised Contrastive Learning

We describe the setup and goal of weakly-supervised learning. Let $\mathcal{X}$ denote instance space and $\mathcal{Y} = \{y^1, y^2, \ldots, y^c\}$ denote label space. $\mathcal{D}$ denotes the joint distribution $(\mathcal{X}, \mathcal{Y})$. Furthermore, given $\mathcal{Q} = \{q^1, q^2, \ldots, q^v\}$ denotes weakly-supervised information space. We denote $\mathcal{D}_{\mathcal{Q}}$ as the joint distribution $(\mathcal{X}, \mathcal{Q})$ which generated form $\mathcal{D}$ with process $\mathcal{D}_{\mathcal{Q}}(x) = \mathcal{D}(x), \mathcal{D}_{\mathcal{Q}}(\boldsymbol{q} \mid x) = \boldsymbol{T}(x)\mathcal{D}(\boldsymbol{y} \mid x)$, where $\mathcal{D}_{\mathcal{Q}}(\boldsymbol{q} \mid x)$ and $\mathcal{D}(\boldsymbol{y} \mid x)$ denote $\left[\mathcal{D}_{\mathcal{Q}}(q^1 \mid x), \ldots, \mathcal{D}_{\mathcal{Q}}(q^v \mid x)\right]^T$ and $\left[\mathcal{D}(y^1 \mid x), \ldots, \mathcal{D}(y^c \mid x)\right]^T$, respectively and $\boldsymbol{T}(x) \in$

$\mathbb{R}^{v \times c}$ is the transition matrix with $\boldsymbol{T}(x)_{i,j} = \mathcal{D}_{\mathcal{Q}}(q^i \mid x, y^j)$. In this paper, we consider two popular weakly-supervised settings with different data generation processes, namely instance-independent and instance-dependent. The former satisfies the condition of $\boldsymbol{T}(x)$ is independent of the samples, whereas the latter does not. The objective of learning with weak supervision is to derive a multi-class predictive model $h$ from weakly-supervised training set $D = \{(x_i, q_i)\}_{i=1}^m \sim \mathcal{D}_{\mathcal{Q}}^m$. For brevity, we denote $\mathbb{P}$ as probability of joint distribution $(\mathcal{X}, \mathcal{Y}, \mathcal{Q})$ induced by $\mathcal{D}$ and $\mathcal{D}_{\mathcal{Q}}$ in the reminder of this paper.

## 2.1. A Graph-Theoretic View

To investigate representation learning in weakly-supervised settings, we adopt a novel view of contrastive learning (HaoChen et al., 2021), framing it as a graph spectral clustering problem where data points serve as vertices and classes correspond to connected sub-graphs. Preliminarily, we consider representation learning under self-supervised condition by modeling an augmentation graph $\mathcal{G}$ where the vertices represent all augmented samples, and the edge weights are constructed based on self-supervised connectivity (Chen et al., 2020; HaoChen et al., 2021):

$$w_{x,x'}^u = \mathbb{E}_{\tilde{x} \sim \mathbb{P}(\mathcal{X})}[\mathcal{A}(x \mid \tilde{x})\mathcal{A}(x' \mid \tilde{x})], \quad (1)$$

where $\mathcal{A}(\cdot \mid \tilde{x})$ denotes distribution of augmentations such as Gaussian blur, color distortion and random cropping from natural data $\tilde{x} \in \mathcal{X}$.

The optimal features are obtained through graph spectral clustering of the given graph, where dense intra-class connectivity plays a key role in ensuring good clustering properties (HaoChen et al., 2021; Sun et al., 2023). However, when the graph is constructed purely based on self-supervised connectivity, the assumption of dense intra-class connections implies that most distinct data points within the same class should share same augmented samples, a condition that may not hold in practice.

Therefore, it is natural to consider incorporating additional intra-class connections by leveraging label information to further improve the quality of the graph, as follows:

$$w_{x,x'}^l \triangleq \mathbb{E}_{(\tilde{x},\tilde{y}),(\tilde{x}',\tilde{y}') \sim \mathbb{P}(\mathcal{X},\mathcal{Y})^2} \\ [\mathbb{I}[\tilde{y} = \tilde{y}']\mathcal{A}(x \mid \tilde{x})\mathcal{A}(x' \mid \tilde{x}')], \quad (2)$$

where $\mathbb{I}[\cdot]$ denotes the indicator function.

With the additional graph connectivity defined above, we can construct improved augmentation graph with perturbation edge weight:

$$w_{x,x'} = \underbrace{\alpha w_{x,x'}^u}_{\text{Self-supervised}} + \underbrace{\beta w_{x,x'}^l}_{\text{Supervised.}} \quad (3)$$

However, in weakly-supervised learning settings, the aforementioned edge weights cannot be directly derived due to the absence of true labels. In the next subsection, we propose a framework that leverages weakly-supervised information to establish additional connectivity, thereby enabling weakly-supervised representation learning.

## 2.2. Representation and Semantic Similarity Matching

The main idea of our framework is to approximate $w_{x,x'}^l$ in Equation 3 using weakly-supervised information. The following proposition demonstrates the existence of such an approximation.

**Proposition 2.1.** *For any $\boldsymbol{S} : \mathcal{X} \to \mathbb{R}^{c \times v}$ that satisfies the condition: $\mathbb{P}(\boldsymbol{y} \mid x) = \boldsymbol{S}(x)\mathbb{P}(\boldsymbol{q} \mid x)$ holds almost everywhere in $\mathcal{X}$, the following equation holds:*

$$w_{x,x'}^{wl}(\boldsymbol{S}) \triangleq \mathbb{E}_{(\tilde{x},\tilde{q}),(\tilde{x}',\tilde{q}') \sim \mathbb{P}(\mathcal{X},\mathcal{Q})^2} \\ [S((\tilde{x},\tilde{q}),(\tilde{x}',\tilde{q}'))\mathcal{A}(x \mid \tilde{x})\mathcal{A}(x' \mid \tilde{x}')] = w_{x,x'}^l, \quad (4)$$

*where $S((\tilde{x},\tilde{q}),(\tilde{x}',\tilde{q}')) = \boldsymbol{S}(\tilde{x})_{:,\tilde{q}}^T \boldsymbol{S}(\tilde{x}')_{:,\tilde{q}'}$.*

**Remark:** Proposition 2.1 reformulates the goal of approximating supervised connections $w_{x,x'}^l$ using weakly-supervised information into constructing an $\boldsymbol{S}$ that satisfies specific properties. In the next subsection, we present methods for constructing such $\boldsymbol{S}$ that meets, or approximately meets these properties. Furthermore, $S((\tilde{x},\tilde{q}),(\tilde{x}',\tilde{q}'))$ in Equation 4 can be viewed as estimation of semantic similarity between two samples according to the corresponding weakly-supervised information. In this context, the graph in Equation 2 is discrete, edge existing only if they belong to the same class, while its approximation in Equation 4 represents a continuous graph, where the connection strength corresponds to the estimated semantic similarity. The proof of Proposition 2.1 is in Appendix B.1.

Based on the approximate $w_{x,x'}^{wl}(\boldsymbol{S})$ provided in Proposition 2.1, we now present a unified framework for weakly-supervised representation learning. To this end, we begin by providing a formal definition of the perturbation augmentation graph.

**Definition 2.2.** (Perturbation augmentation graph) We refer a graph where the vertices represent all augmented samples and adjacency matrix $\boldsymbol{A}$ constructed by perturbation connectivity as perturbation augmentation graph. Specifically, giving two augmentation data points $x, x'$:

$$\boldsymbol{A}_{x,x'} = \alpha w_{x,x'}^u + \beta w_{x,x'}^{wl}(\boldsymbol{S}). \quad (5)$$

Let $\boldsymbol{F}$ be an embedding matrix with $\boldsymbol{F}_{x,:} \in \mathbb{R}^d$ is embedding of sample $x$, we consider optimal features derived from the graph spectral clustering of the given such graph in Definition 2.2 (Chung, 1997; HaoChen et al., 2021):

$$\boldsymbol{F}^* = \arg\min \left\| \widetilde{\boldsymbol{A}} - \boldsymbol{F}\boldsymbol{F}^T \right\|_F, \quad (6)$$

where $\widetilde{A}$ is normalized adjacency matrix of $A$ with $\widetilde{A}_{x,x'} = \frac{A_{x,x'}}{\sqrt{A_x A_{x'}}}$, $A_x = \sum_{x'} A_{x,x'}$.

Now, if we view each row of $F$ as a scaled version of learned feature embedding $f : \mathcal{X} \to \mathbb{R}^d$, the above optimal features can be obtained through minimize an end-to-end contrastive learning loss. We formalize this connection in following proposition.

**Proposition 2.3.** *We define $F_{x,:} = \sqrt{A_x} f_x$ and weakly-supervised contrastive loss $\mathcal{L}_{wsc}$ as follow:*

$$
\begin{aligned}
\mathcal{L}_{wsc}(f) \triangleq & - 2\alpha\mathcal{L}_1(f) - 2\beta\mathcal{L}_2(f) \\
& + \alpha^2 \mathcal{L}_3(f) + \beta^2 \mathcal{L}_4(f) + 2\alpha\beta\mathcal{L}_5(f),
\end{aligned} \tag{7}
$$

*where*

$$
\mathcal{L}_1(f) \triangleq \mathbb{E}_{\tilde{x}\sim\mathbb{P}(\mathcal{X}), x,x'\sim\mathcal{A}(\cdot|\tilde{x})} \left[ f_x f_{x'}^T \right], \tag{8}
$$

$$
\mathcal{L}_2(f) \triangleq \mathbb{E}_{(\tilde{x},\tilde{q}),(\tilde{x}',\tilde{q}')\sim\mathbb{P}(\mathcal{X},\mathcal{Q})^2, x\sim\mathcal{A}(\cdot|\tilde{x}), x'\sim\mathcal{A}(\cdot|\tilde{x}')} \left[ S((\tilde{x},\tilde{q}),(\tilde{x}',\tilde{q}')) f_x f_{x'}^T \right], \tag{9}
$$

$$
\mathcal{L}_3(f) \triangleq \mathbb{E}_{\tilde{x},\tilde{x}'\sim\mathbb{P}(\mathcal{X})^2, x\sim\mathcal{A}(\cdot|\tilde{x}), x'\sim\mathcal{A}(\cdot|\tilde{x}')} \left[ \left( f_x f_{x'}^T \right)^2 \right], \tag{10}
$$

$$
\mathcal{L}_4(f) \triangleq \mathbb{E}_{(\tilde{x},\tilde{q}),(\tilde{x}',\tilde{q}')\sim\mathbb{P}(\mathcal{X},\mathcal{Q})^2, x\sim\mathcal{A}(\cdot|\tilde{x}), x'\sim\mathcal{A}(\cdot|\tilde{x}')} \left[ S'(\tilde{x},\tilde{q}) S'(\tilde{x}',\tilde{q}') \left( f_x f_{x'}^T \right)^2 \right], \tag{11}
$$

$$
\mathcal{L}_5(f) \triangleq \mathbb{E}_{(\tilde{x},\tilde{q})\sim\mathbb{P}(\mathcal{X},\mathcal{Q}), \tilde{x}'\sim\mathbb{P}(\mathcal{X}), x\sim\mathcal{A}(\cdot|\tilde{x}), x'\sim\mathcal{A}(\cdot|\tilde{x}')} \left[ S'(\tilde{x},\tilde{q}) \left( f_x f_{x'}^T \right)^2 \right], \tag{12}
$$

*where* $S\left((\tilde{x},\tilde{q}),(\tilde{x}',\tilde{q}')\right) = S(\tilde{x})_{:,\tilde{q}}^T S(\tilde{x}')_{:,\tilde{q}'}, S'(\tilde{x},\tilde{q}) = S(\tilde{x})_{:,\tilde{q}}^T \mathbb{P}(y)$.

*Then the following equation holds:*

$$
\left\| \widetilde{A} - FF^T \right\|_F = \mathcal{L}_{wsc}(f) + Const. \tag{13}
$$

*Furthermore, when assuming uniform class distribution $\mathbb{P}(y) = \left[\frac{1}{c}, \ldots, \frac{1}{c}\right]^T$, $\mathcal{L}_{wsc}(f)$ can be rewritten as:*

$$
\mathcal{L}_{wsc}(f) \triangleq -2\alpha\mathcal{L}_1(f) - 2\beta\mathcal{L}_2(f) + \left(\alpha + \frac{\beta}{c}\right)^2 \mathcal{L}_3(f) \tag{14}
$$

**Remark:** By the Eckart–Young–Mirsky theorem (Eckart and Young, 1936), $F^*$ in Equation 6 consists of the top-$d$ eigenvectors of $\widetilde{A}$. This property enables clustering performance analysis via the connectivity of $\widetilde{A}$. Proposition 2.3 outlines an end-to-end method for training a neural network to approximate these optimal features. The pseudo code for computing $\mathcal{L}_{wsc}$ in Equation 7 with batch data under a uniform class distribution is given in Algorithm 1, with the general version (without the uniform class assumption) provided in Appendix E. We can intuitively interpret $\mathcal{L}_{wsc}$

---

**Algorithm 1** The calculation of $\mathcal{L}_{wsc}$ in uniform class distribution case

**Input:** features of two augmentation views of batch data with corresponding weakly-supervised information $X_Q^1, X_Q^2 \in \mathbb{R}^{B_Q \times d}, Q \in \mathcal{Q}^{B_Q}$, features of two augmentation view of batch unlabeled data $X_U^1, X_U^2 \in \mathbb{R}^{B_U \times d}$, $S$ in Proposition 2.1, coefficients $\alpha, \beta$

**Output:** batch data estimated loss $\widehat{\mathcal{L}}_{wsc}$

Compute $S(X) \in \mathbb{R}^{c \times B_Q}, S(X)_{:,x} = S(x)_{:,q}$

Compute $\widehat{\mathcal{L}}_1 = \frac{\text{Tr}\left[X_Q^1 (X_Q^2)^T\right] + Tr\left[X_U^1 (X_U^2)^T\right]}{B_Q + B_U}$

Compute $\widehat{\mathcal{L}}_2 = \frac{\left\| (S(X)^T S(X)) \otimes (X_Q^1 (X_Q^2)^T) \right\|_1}{B_Q^2}$

Take $X^1 = \left[X_Q^1; X_U^1\right], X^2 = \left[X_Q^2; X_U^2\right]$

Compute $\widehat{\mathcal{L}}_3 = \frac{\left\| X^1 (X^2)^T \right\|_2^2}{(B_Q + B_U)^2}$

**Return:** $\widehat{\mathcal{L}}_{wsc} = -2\alpha\widehat{\mathcal{L}}_1 - 2\beta\widehat{\mathcal{L}}_2 + (\alpha + \beta/c)^2 \widehat{\mathcal{L}}_3$

---

as follows: $\mathcal{L}_1$ and $\mathcal{L}_2$ push the features of positive pairs to be closer, with $\mathcal{L}_1$ pulling closer features of two views of the same image, and $\mathcal{L}_2$ pulling closer features of different images with different weights which equal to continuous adjacency matrix $S(X)S(X)^T$ in Algorithm 1 of the sampled sub-graph. $\mathcal{L}_3, \mathcal{L}_4, \mathcal{L}_5$ are regularization terms to prevents feature collapse. Finally, $\mathcal{L}_{wsc}$ depends solely on $S$, hence it can be adapted to multiple weakly-supervised setting by constructing the corresponding $S$. The proof is provided in Appendix B.2.

### 2.3. Instantiating under Various Weakly Information

In this paper, we consider two typical paradigms of weakly supervision: noisy label learning (NLL), partial label learning (PLL). For noisy label leaning, we have $\mathcal{Q} = \mathcal{Y}$ but mistakes exist $\mathcal{D}_{\mathcal{Q}}(T(X) \neq I_{c \times c}) > 0$. For partial label learning, we have $\mathcal{Q} = 2^{\mathcal{Y}} \setminus \emptyset$ and $\mathcal{D}_{\mathcal{Q}}(y \in q \mid x) = 1, \forall x \in \mathcal{X}$. In the following text, we show how to instantiate our framework into these two settings.

Reviewing Proposition 2.1, a good $S$ ought to satisfy or approximately satisfy the condition $\mathbb{P}(y \mid x) = S(x)\mathbb{P}(q \mid x)$. In this subsection, we provide methodologies for constructing it in various settings, thereby completing the instantiation of the framework. Next, we will separately discuss instance-independent and instance-dependent settings.

**Instance-Independent Setting.** For this setting, we have instance-independent transition matrix $T$: there exists a $T$ satisfies $T = T(x)$ holding almost everywhere in $\mathcal{X}$. The following proposition illustrates that $S$ can be constructed by estimating the transition matrix $T$.

**Proposition 2.4.** *Under instance-independent assumption, we have the sufficient condition for $S(x) = S, \forall x \in \mathcal{X}$ to*

*satisfy condition in Proposition 2.1 as follows:*

$$\boldsymbol{ST} = \boldsymbol{I}_{c \times c}. \tag{15}$$

*where $\boldsymbol{I}_{c \times c}$ denote the identity matrix of size $c \times c$.*

*Proof.* When $\boldsymbol{S}$ satisfies the above condition, we have following equation holds almost everywhere in $\mathcal{X}$:

$$\boldsymbol{S}\mathbb{P}(\boldsymbol{q} \mid x) = \boldsymbol{ST}\mathbb{P}(\boldsymbol{y} \mid x) = \mathbb{P}(\boldsymbol{y} \mid x). \tag{16}$$

□

Inspired by Proposition 2.4, we can use any matrix $\boldsymbol{S}$ for $\mathcal{L}_{wsc}$ that satisfies $\boldsymbol{S}\widehat{\boldsymbol{T}} = \boldsymbol{I}$, where $\widehat{\boldsymbol{T}}$ is an estimate of $\boldsymbol{T}$. In the noisy label setting, methods such as Liu and Tao 2016; Patrini et al. 2017; Xia et al. 2019; Li et al. 2021; Lin et al. 2023 can estimate $\widehat{\boldsymbol{T}}$, and $\widehat{\boldsymbol{S}}$ is its inverse. In the partial label setting, $\boldsymbol{T} \in \mathbb{R}^{(2^c-1) \times c}$ is hard to estimate due to the curse of dimensionality, but with the assumption where labels are independent adopted into the candidate set (Cour et al., 2011; Lv et al., 2020; Feng et al., 2020b), $\widehat{\boldsymbol{T}}$ can be estimated by $c$ times of mixing proportion estimation (Garg et al., 2021), and $\widehat{\boldsymbol{S}}$ can be derived through pseudo-inverse or constructive methods. See more details in Appendix F.

**Instance-Dependent Setting.** In this setting, a universal transition matrix $\boldsymbol{T}$ applicable to all samples does not exist, thereby rendering the methods discussed previously ineffective. However, according to Bayes theorem, we have:

$$\mathbb{P}(y \mid x) = \sum_{q \in \mathcal{Q}} \mathbb{P}(y, q \mid x) = \sum_{q \in \mathcal{Q}} \mathbb{P}(y \mid x, q)\mathbb{P}(q \mid x). \tag{17}$$

Thus, we can construct $\widehat{\boldsymbol{S}}$ as $\widehat{\boldsymbol{S}}(x)_{y,q} = \widehat{\mathbb{P}}(y \mid x, q)$ where $\widehat{\mathbb{P}}(y \mid x, q)$ can be any estimation of $\mathbb{P}(y \mid x, q)$. Specifically, it is obtained through the current predictions of the model. This process is also called self-labeling and has been widely discussed. Specifically, in the context of noisy labels, related studies include Liu et al. 2020; Li et al. 2022; Ortego et al. 2021; Liu et al. 2022; Xiao et al. 2023; Qiao et al. 2024; for partial labels, related studies are conducted by Xu et al. 2021; Wang et al. 2022; Wu et al. 2022; Zhang et al. 2022; Qiao et al. 2023a; 2024. See more details in Appendix F.

# 3. Theoretical Understanding

We now present a theoretical analysis of the performance of features learned from weakly-supervised data. Following the traditional theoretical framework (Chen et al., 2020; HaoChen et al., 2021), we use the linear probe to evaluate the performance of representation learning. Concretely, we use a linear classifier with weights $\boldsymbol{B} \in \mathbb{R}^{d \times c}$ and predict $h_{f,\boldsymbol{B}}(x) = \arg\max_{i \in \mathcal{Y}} \boldsymbol{B}_{:,i}^T f(x)$ for an augmented data

$x$. Then given naturally data $\widetilde{x} \in \mathcal{X}$, we ensemble the predictions on augmented data and predict:

$$\tilde{h}_{f,\boldsymbol{B}}(\tilde{x}) = \arg\max_{i \in \mathcal{Y}} \mathbb{E}_{x \sim \mathcal{A}(\cdot \mid \tilde{x})} \left[\mathbb{I}\left[h_{f,\boldsymbol{B}}(x) = i\right]\right]. \tag{18}$$

We define linear probe error of representation $f$ as follow:

$$\varepsilon(f) = \min_{\boldsymbol{B} \in \mathbb{R}^{d \times c}} \mathbb{E}_{(\tilde{x},\tilde{y}) \sim \mathbb{P}(\mathcal{X},\mathcal{Y})} \left[\mathbb{I}\left[\tilde{h}_{f,\boldsymbol{B}}(\tilde{x}) \neq \tilde{y}\right]\right]. \tag{19}$$

Regarding the given constructed $\widehat{\boldsymbol{S}}$ and training set $D = D_{\mathcal{Q}} \cup D_{\mathcal{U}}$, where $D_{\mathcal{Q}}$ is set of training samples with corresponding weakly-supervised information and $D_{\mathcal{U}}$ is set of unlabeled samples, we learn $f$ through:

$$\widehat{f}(D) = \arg\min \widehat{\mathcal{L}}_{wsc}(f), \tag{20}$$

where $\widehat{\mathcal{L}}_{wsc}(f)$ is the estimate of $\mathcal{L}_{wsc}(f)$ using data $D_{\mathcal{Q}}$ and $D_{\mathcal{U}}$ with constructed $\widehat{\boldsymbol{S}}$.

In this section, we examine $\varepsilon(\widehat{f}(D))$ through three steps:

- We consider $\varepsilon(f^*)$, where $f^*$ is derived from the spectral clustering of the perturbation graph $\boldsymbol{A}$.

- We consider the gap between $\mathcal{L}_{wsc}(f^*)$ and $\mathcal{L}_{wsc}(\widehat{f}(D))$. Specifically, such a gap is caused by two aspects. On the one hand, there is the estimation error resulting from the limited samples. On the other hand, the constructed $\widehat{\boldsymbol{S}}$ cannot fully satisfy the properties in Proposition 2.1.

- We control $\varepsilon(\widehat{f}(D)) - \varepsilon(f^*)$ through $\mathcal{L}_{wsc}(\widehat{f}(D)) - \mathcal{L}_{wsc}(f^*)$ to obtain the minimum downstream error.

## 3.1. How Can Supervised Information Help Representation Learning

In this subsection, we examine $\varepsilon(f^*)$ where $f^*$ is derived from graph spectral clustering of $\boldsymbol{A}$ with entries $\boldsymbol{A}_{x,x'} = \alpha w_{x,x'}^u + \beta w_{x,x'}^{wl}(\boldsymbol{S})$. The two properties of a graph serve as crucial elements that have a significant impact on performance. Specifically, these properties are the density of connections within a class and the sparsity of connections between classes. The following several definitions respectively characterize these properties of the graph.

**Definition 3.1.** (Data points from the different class hardly share augmented data) We refer data augmentation $\mathcal{A}$ as $\gamma$-*consistent* under joint distribution $\mathbb{P}(\mathcal{X}, \mathcal{Y})$ if there exist a pseudo labeler $\hat{y}(x)$ for augmentation data such that:

$$\mathbb{E}_{(\tilde{x},\tilde{y}) \sim \mathbb{P}(\mathcal{X},\mathcal{Y}), x \sim \mathcal{A}(\cdot \mid \tilde{x})} \left[\mathbb{I}\left[\tilde{y} = \hat{y}(x)\right]\right] \leq \gamma. \tag{21}$$

Small $\gamma$ in Definition 3.1 implies the sparsity of connections between different classes in $\boldsymbol{A}^u$ which constructed by self-supervised connectivity. Next, to formulate density of connections within a class, we introduce Dirichlet conductance and sparsest partition which are standard in spectral graph theory.

**Definition 3.2.** (Dirichlet conductance) For a graph $\mathcal{G}(\mathcal{X}, w)$ and a subset $\Omega \subseteq \mathcal{X}$, we define the Dirichlet conductance of $\Omega$ as:

$$\Phi_{\mathcal{G}}(\Omega) = \frac{\sum_{x \in \Omega, x' \notin \Omega} w_{x,x'}}{\sum_{x \in \Omega} w_x}. \tag{22}$$

**Definition 3.3.** (Sparsest partition) For a graph $\mathcal{G}(\mathcal{X}, w)$ and an integer $i \in [2, |\mathcal{X}|]$, we define the sparsest $i$-partition:

$$\rho_i = \min_{\Omega_1, \ldots, \Omega_i} \max\{\Phi_{\mathcal{G}}(\Omega_1), \ldots, \Phi_{\mathcal{G}}(\Omega_i)\}, \tag{23}$$

where $\Omega_1, \ldots, \Omega_i$ are non-empty sets that form a partition of $\mathcal{X}$.

Intuitively, Dirichlet conductance represents the fraction of edges from $\Omega$ to its complement, and sparsest partition represent the number of edges between many disjoint subsets. When sparsity of connections between classes holds, we might expect $\rho_c \approx 0$. Furthermore, any $\rho_d$ where $d > c$ needs to break one class into many pieces, hence a large $\rho_d$ reflects density of connections within a class. Giving such definition for two main properties of graph, the following theorem bounds linear probe error of feature derived from perturbation graph.

**Theorem 3.4.** *Given augmentation graph $\mathcal{G}$ with perturbation adjacency matrix $\boldsymbol{A} = \alpha \boldsymbol{A}^u + \beta \boldsymbol{A}^{wl}(\boldsymbol{S})$ where $\boldsymbol{S}$ stratify properties in Proposition 2.1, if $\mathcal{A}$ is $\gamma$-consistent with $\gamma > 0$ and representation dimension $d > 2c$, we have:*

$$\varepsilon(f^*) \preceq \widetilde{O}\left(\gamma\left(\alpha^* \rho^u_{\lfloor \frac{d}{2} \rfloor} + \beta^*\left(1 - \frac{2c}{d}\right)\right)^{-2}\right), \tag{24}$$

*where $\rho^u$ is sparsest partition of self-supervised connectivity graph $\boldsymbol{A}^u$, and $\alpha^* = \frac{\alpha}{\alpha + \beta/c}$, $\beta^* = \frac{\beta/c}{\alpha + \beta/c}$.*

**Remark:** Theorem 3.4 bounds linear probe error of feature derived from perturbation graph through properties of self-supervised connectivity graph and perturbation coefficients. Theorem 3.4 demonstrates that the perturbation graph increases the density of intra-class connections without adding extra-class connections, thereby improving the performance of feature. Specifically, when $\beta^*$ in Theorem 3.4 equals to zero, the theorem degenerates into Theorem 3.8. in HaoChen et al. 2021 which bounds linear probe error of feature derived from self-supervised connectivity graph. When assuming $\rho^u_{d/2} < 1 - \frac{2c}{d}$, bounds in Equation 24 decrease as $\beta^*$ increases. Such a condition is almost always satisfied in reality, due to $\rho_i \leq \widetilde{O}(1 - \frac{1}{i})$ for every graph and density of the graph constructed using self-connections is usually much smaller than this upper bound (HaoChen et al., 2021). The proof of Theorem 3.4 is in Appendix C.1.

### 3.2. How Weakly-Supervised Information Approximates Supervised Information

In this subsection, we first consider the gap between $\mathcal{L}_{wsc}(\widehat{f}(D))$ and $\mathcal{L}_{wsc}(f^*)$. $\widehat{f}(D)$ is an optimal feature that minimizes empirical loss $\widehat{\mathcal{L}}_{wsc}(f; D)$ defined as:

$$\widehat{\mathcal{L}}_{wsc}(f; D) = \mathbb{E}_{\boldsymbol{X}^1_Q, \boldsymbol{X}^2_Q \sim f(\mathcal{A}(\cdot | D_Q)^2) \boldsymbol{X}^1_U, \boldsymbol{X}^2_U \sim f(\mathcal{A}(\cdot | D_U)^2}$$
$$\left[\widehat{\mathcal{L}}_{wsc}(f; \boldsymbol{X}^1_Q, \boldsymbol{X}^2_Q, \boldsymbol{X}^1_U, \boldsymbol{X}^2_U, \widehat{\boldsymbol{S}}, Q)\right], \tag{25}$$

where $\widehat{\boldsymbol{S}}$ is constructed to approximately satisfy the property in Proposition 2.1 and $\widehat{\mathcal{L}}_{wsc}$ in expectation is computed through Algorithm 1.

To this end, we decouple $\mathcal{L}_{wsc}(\widehat{f}(D)) - \mathcal{L}_{wsc}(f^*)$ into two parts, the finite sample error and the approximation error caused by the constructed $\widehat{\boldsymbol{S}}$. To analyze the first part, we introduce the Rademacher complexity which is a standard concept in generalization error analysis.

**Definition 3.5.** (Maximal possible empirical Rademacher complexity for feature extractor) Let $\mathcal{F}$ be a hypothesis class of feature extractors from $\mathcal{X}$ to $\mathbb{R}^d$, for $n \in \mathbb{Z}^+$, we define Maximal possible empirical Rademacher complexity $\widehat{\mathcal{R}}_n(\mathcal{F})$ for feature extractor under $\mathcal{X}$ as:

$$\widehat{\mathcal{R}}_n(\mathcal{F}) = \max_{i \in [d]} \max_{\{x_1, \ldots, x_n\} \in \mathcal{X}^n} \mathbb{E}_{\sigma}\left[\sup_{f \in \mathcal{F}} \frac{1}{n} \sum_{j=1}^{n} \sigma_j f_i(x_j)\right], \tag{26}$$

where $\sigma = \{\sigma_1, \ldots, \sigma_n\}$ are $n$ Rademacher variables with $\sigma_i$ independently uniform variable taking value in $\{+1, -1\}$.

To analyze the second part, we introduce expected bias for approximated $\widehat{\boldsymbol{S}}$.

**Definition 3.6.** (Expected bias for $\widehat{\boldsymbol{S}}$) We defined expected bias $\Delta(\widehat{\boldsymbol{S}})$ under $\mathcal{D}(\mathcal{X})$ as follow:

$$\Delta(\widehat{\boldsymbol{S}}) = \mathbb{E}_{x \sim \mathbb{P}(\mathcal{X})}\left[\left\|\mathbb{P}(\boldsymbol{y} \mid x) - \widehat{\boldsymbol{S}}(x)\mathbb{P}(\boldsymbol{q} \mid x)\right\|_1\right]. \tag{27}$$

Utilizing the above two definitions, we have following theorem bounds the gap $\mathcal{L}_{wsc}(\widehat{f}(D)) - \mathcal{L}_{wsc}(f^*)$.

**Theorem 3.7.** *Given perturbation coefficients $\alpha, \beta$ satisfy $\alpha + \beta/c = 1$, and assuming $\|\mathcal{F}\|_\infty < +\infty$, for a random training dataset $D = D_Q \cup D_U$ with $n_q$ and $n_u$ samples respectively. Then with probability at least $1 - \delta$, we have:*

$$\mathcal{L}_{wsc}(\widehat{f}(D)) - \mathcal{L}_{wsc}(f^*) \leq (\alpha + 1)\eta_0 \widehat{\mathcal{R}}_{\frac{n_u + n_q}{2}}(\mathcal{F})$$
$$+ (\alpha + 1)\eta_1 \eta(n_u + n_q, \delta) + \beta \eta_2 \Delta(\widehat{\boldsymbol{S}})$$
$$+ \beta \sup_{x \in \mathcal{X}} \left\|\widehat{\boldsymbol{S}}(x)^T \widehat{\boldsymbol{S}}(x)\right\|_\infty \left(\eta_3 \widehat{\mathcal{R}}_{\frac{n_q}{2}}(\mathcal{F}) + \eta_4 \eta(n_q, \delta)\right), \tag{28}$$

*where $\eta_0, \eta_1, \eta_2, \eta_3, \eta_4$ are constants related to $\|\mathcal{F}\|_\infty$ and feature dimension $d$ and $\eta(n, \delta) = \sqrt{\frac{\log 2/\delta}{n}} + \frac{\delta}{2}$.*

**Remark:** Theorem 3.7 demonstrates that in our framework, the features learned through finite samples and weak supervision information can approximate the features derived from the perturbation graph which incorporates supervision information and thus has been proved in the Theorem 3.4 to possess good clustering properties. Specifically, the approximation error can be regarded as two parts. The first part, comprising the first and second terms in Equation 28, is the finite sample error when using all the training samples to approximate the self-supervised graph, which is derived through the standard finite sample approximation analysis. The second part is the error when using the samples with weakly supervision information to approximate the perturbed part. The third and fourth terms in the Equation 28 describe two aspects of this error respectively. The third term describes the estimation error brought about by the bias of $\widehat{S}$, which can be regarded as the bias term of the error. The fourth term describes the approximation error due to limited samples and can be regarded as the variance term of the error. $\sup_{x \in \mathcal{X}} \left| \widehat{S}(x)\widehat{S}(x)^T \right|$ illustrates that the presence of weakly-supervised information exacerbates the learning difficulty, even when an unbiased estimate of $\widehat{S}$ is utilized. From this perspective, the two terms can be regarded as the trade-off between variance and bias. The proof of Theorem 3.7 is in Appendix C.2.

Theorem D.7 in HaoChen et al. 2021 characterize the error propagation from pre-training to the downstream task, combine with our Theorems 3.4 and 3.7, we can obtain final $\epsilon(\widehat{f}(D))$ as follows:

**Corollary 3.8.** *(Main results: linear probe error of feature learned from our framework) Given representation dimension $d > 4r$, and $\alpha, \beta$ satisfy $\alpha + \frac{\beta}{c} = 1$, $\mathcal{A}$ is $\gamma$-consistent with $\gamma > 0$, then with probability at least $1 - \delta$, we have:*

$$\varepsilon(\widehat{f}(D)) \leq \widetilde{O}\left( \gamma \left( \alpha^* \rho^u_{\lfloor \frac{d}{2} \rfloor} + \beta^* \left( 1 - \frac{2c}{d} \right) \right)^{-2} \right)$$
$$+ (\alpha + 1) \left( \eta_0' \widehat{\mathcal{R}}_{\frac{n}{2}}(\mathcal{F}) + \eta_1' \eta(n, \delta) \right) + \beta \eta_2' \Delta(\widehat{S})$$
$$+ \beta \sup_{x \in \mathcal{X}} \left\| \widehat{S}(x)^T \widehat{S}(x) \right\|_\infty \left( \eta_3' \widehat{\mathcal{R}}_{\frac{n_q}{2}}(\mathcal{F}) + \eta_4' \eta(n_q, \delta) \right),$$
$$(29)$$

*where $[\eta_0', \eta_1', \eta_2', \eta_3', \eta_4'] = \frac{d}{\Delta_\lambda^2}[\eta_0, \eta_1, \eta_2, \eta_3, \eta_4]$ with $\Delta_\lambda$ is the eigenvalue gap between the $\frac{3}{4}d$-th and the $d$-th eigenvalue of perturbation graph, and $n = n_u + n_q$.*

*Proof.* By substituting Theorems 3.4 and 3.7 into Theorem D.7 in HaoChen et al. 2021, the corollary can be directly obtained. The restatement of Theorem D.7 in HaoChen et al. 2021 can been found in Appendix C.3. $\square$

**Remark:** Corollary 3.8 demonstrates that the performance of the features learned within our framework is determined by two aspects. Firstly, the quality of the clustering structure of the constructed augmented graph. Secondly, the error arising from approximating the constructed graph using finite samples with weakly-supervised information. As $\beta$ increases, the clustering structure of the constructed graph improves, but the approximation error also increase accordingly. This suggests that in practical representation learning, selecting an appropriate $\beta$ is crucial for effectively integrating self-supervised and supervised information.

## 4. Experiment

In this section, we empirically validate the efficacy of our framework across two paradigms of weakly-supervised learning, namely noisy label learning (NLL), partial label learning (PLL). For implementation, our weakly-supervised contrastive (WSC) loss is combined only with the simplest baseline. Detailed implementation can be found in Appendix E.

### 4.1. Learning with Noisy Labels

**Dataset and Experimental Setting.** For the case of noisy label learning, following the settings in Li et al. 2020; Liu et al. 2020, we verify the effectiveness of our method on CIFAR-10 and CIFAR-100 (Krizhevsky, 2012) with two types of label noise: *symmetric* and *asymmetric*. Symmetric noise refers to the random reassignment of labels within the training set according to a predefined noise ratio. In contrast, asymmetric noise is designed to replicate real-world label noise, where labels are replaced exclusively by those of similar classes (*e.g.*, dog $\leftrightarrow$ cat). In order to fully verify the effectiveness of the proposed method, we select ten baselines for NLL comparison under the same experimental setting: DivideMix (Li et al., 2020), ELR (Liu et al., 2020), SOP (Liu et al., 2022), GFWS (Chen et al., 2024) and ProMix (Xiao et al., 2023), which do not incorporate contrastive learning, as well as MOIT (Ortego et al., 2021), Sel-CL (Li et al., 2022), and TCL (Huang et al., 2023), which leverage a contrastive learning module. Due to space limitations, the implementation details, along with additional comparisons on the instance-dependent noisy dataset CIFAR-10N (Wei et al., 2022) and the realistic, larger-scale instance-dependent noisy dataset Clothing1M (Xiao et al., 2015), are provided in Appendix E.1.

**Experimental Result.** Table 1 shows the classification accuracy for each comparative approach. Most of the experimental results are the same as those reported in their original paper, except for ProMix, which we reproduce due to differences in our settings. The proposed method demonstrates competitive performance across various settings and datasets. It outperforms existing approaches in most sce-

*Table 1.* Comparisons with each methods on simulated NLL datasets. Each run has been three times with different randomly generated noise, and the mean of the last five epochs are reported.

| Dataset | CIFAR-10 | | | | CIFAR-100 | | | | |
|---|---|---|---|---|---|---|---|---|---|
| Noise Type | Sym. | | | Asym. | Sym. | | | | Asym. |
| Noise Ratio | 0.5 | 0.8 | 0.9 | 0.4 | 0.2 | 0.5 | 0.8 | 0.9 | 0.4 |
| CE | 80.70 | 65.80 | 42.70 | 82.20 | 58.10 | 47.10 | 23.80 | 3.50 | 43.34 |
| DivideMix | 94.60 | 93.20 | 76.00 | 93.40 | 77.10 | 74.60 | 60.20 | 31.00 | 55.57 |
| ELR | 94.80 | 93.30 | 78.70 | 93.00 | 77.90 | 73.80 | 60.80 | 33.40 | 69.94 |
| SOP | 95.50 | 94.00 | - | 93.80 | 78.80 | 75.90 | 63.30 | - | 69.53 |
| GFWS | **96.60** | 94.12 | 84.22 | 94.75 | 77.49 | 75.51 | 66.46 | 45.82 | 75.82 |
| ULAREF | 94.31 | 91.47 | - | 92.56 | 76.16 | 72.39 | 54.72 | - | 76.11 |
| ProMix | 93.23 | 83.11 | - | 89.83 | 75.43 | 71.64 | 43.35 | - | 72.13 |
| MOIT | 90.00 | 79.00 | 69.60 | 92.00 | 73.00 | 64.60 | 46.50 | 36.00 | 71.55 |
| Sel-CL | 93.90 | 89.20 | 81.90 | 93.40 | 76.50 | 72.40 | 57.70 | 29.30 | 74.20 |
| TCL | 93.90 | 92.50 | 89.40 | 92.60 | 78.00 | 73.30 | 65.00 | 54.50 | 73.10 |
| WSC (Ours) | 95.79$_{\pm0.19}$ | **94.62$_{\pm0.07}$** | **90.93$_{\pm0.14}$** | **95.22$_{\pm0.09}$** | **79.21$_{\pm0.13}$** | **77.51$_{\pm0.17}$** | **71.92$_{\pm0.17}$** | **61.32$_{\pm0.15}$** | **76.31$_{\pm0.32}$** |

*Table 2.* Comparisons with each methods on simulated PLL datasets. Each run has been repeated three times with different randomly generated partial labels, and the mean and standard deviation values of the last five epochs are reported.

| Dataset | CIFAR-10 | | | | CIFAR-100 | | | | CUB-200 |
|---|---|---|---|---|---|---|---|---|---|
| Partial Ratio | 0.5 | 0.6 | 0.7 | 0.8 | 0.05 | 0.1 | 0.2 | 0.3 | 0.05 |
| LWS | 85.30$_{\pm0.36}$ | 80.33$_{\pm1.33}$ | 72.11$_{\pm0.58}$ | 58.49$_{\pm0.33}$ | 54.78$_{\pm0.26}$ | 50.44$_{\pm0.38}$ | 32.44$_{\pm0.48}$ | 25.49$_{\pm0.58}$ | 39.74$_{\pm0.43}$ |
| PRODEN | 89.82$_{\pm0.38}$ | 87.44$_{\pm0.35}$ | 86.44$_{\pm0.20}$ | 85.78$_{\pm0.55}$ | 72.65$_{\pm0.09}$ | 71.05$_{\pm0.23}$ | 68.44$_{\pm0.79}$ | 55.21$_{\pm0.44}$ | 62.56$_{\pm0.10}$ |
| CC | 82.30$_{\pm0.28}$ | 80.08$_{\pm0.07}$ | 77.15$_{\pm0.45}$ | 75.94$_{\pm0.88}$ | 63.74$_{\pm0.51}$ | 57.55$_{\pm0.19}$ | 50.41$_{\pm0.47}$ | 40.28$_{\pm0.29}$ | 55.61$_{\pm0.02}$ |
| MSE | 75.47$_{\pm0.85}$ | 73.64$_{\pm0.28}$ | 69.09$_{\pm0.67}$ | 64.32$_{\pm0.33}$ | 51.17$_{\pm0.25}$ | 47.33$_{\pm0.94}$ | 42.55$_{\pm0.33}$ | 35.11$_{\pm0.49}$ | 22.07$_{\pm2.36}$ |
| GFWS | 95.22$_{\pm0.08}$ | 95.01$_{\pm0.03}$ | 94.21$_{\pm0.14}$ | 93.58$_{\pm0.21}$ | 76.89$_{\pm0.32}$ | 75.95$_{\pm0.10}$ | 73.18$_{\pm0.51}$ | 60.25$_{\pm0.37}$ | 70.77$_{\pm0.20}$ |
| RCR | 95.01$_{\pm0.03}$ | 94.37$_{\pm0.07}$ | 93.28$_{\pm0.05}$ | 91.67$_{\pm0.10}$ | 77.01$_{\pm0.22}$ | 75.85$_{\pm0.35}$ | 72.58$_{\pm0.31}$ | 57.48$_{\pm0.95}$ | - |
| PiCO | 94.63$_{\pm0.21}$ | 94.25$_{\pm0.31}$ | 93.68$_{\pm0.06}$ | 92.01$_{\pm0.18}$ | 74.19$_{\pm0.28}$ | 72.74$_{\pm0.64}$ | 70.89$_{\pm0.44}$ | 61.35$_{\pm0.88}$ | 72.12$_{\pm0.74}$ |
| WSC (Ours) | **95.41$_{\pm0.08}$** | **95.22$_{\pm0.10}$** | **95.03$_{\pm0.04}$** | **94.41$_{\pm0.05}$** | **77.88$_{\pm0.19}$** | **77.26$_{\pm0.30}$** | **75.13$_{\pm0.24}$** | **69.15$_{\pm0.05}$** | **74.55$_{\pm0.17}$** |

narios, including those based on other contrastive learning techniques. Notably, our method shows significant improvements under high noise rates. For instance, on CIFAR-100 with a 90% noise rate, it surpasses the previous best TCL by **6.85%**, highlighting the substantial performance gains our framework offers for NLL.

### 4.2. Learning with Partial Labels

**Dataset and Experimental Setting.** For the case of partial label learning, following Wang et al. 2022, we verify the effectiveness of our method on CIFAR-10 (Krizhevsky, 2012), CIFAR-100 (Krizhevsky, 2012) and CUB-200 (Wah et al., 2011b) with different partial ratios. The partial label datasets are generated by flipping the negative labels to candidate labels with a specified partial ratio. In other words, all $c - 1$ negative labels can be uniformly aggregated into the ground truth label to form the set of candidate labels. We choose seven baselines for PLL using same experiment setting for comparison: LWS (Wen et al., 2021), PRODEN (Lv et al., 2020), CC (Feng et al., 2020b), MSE (Feng et al., 2020a), RCR (Wu et al., 2022) and GFWS (Chen et al.,

2024), which do not incorporate contrastive learning, as well as PiCO (Wang et al., 2022), which employs a contrastive learning module. The implementation details, along with additional comparisons using different partial label ratios on CUB-200 and the hierarchical-generated partial label dataset CIFAR-100-H (Wang et al., 2022), are provided in Appendix E.2.

**Experimental Result.** Table 2 shows the main results for PLL. Our proposed method achieves the best performance across almost all settings compared to the baseline methods. It is worth noting that our method achieves a larger performance gap among the previous methods when the partial rate is large. Especially on CIFAR-100 with a partial ratio of 0.3, the proposed method outperforms previous best method by **7.8%**, which is a strong evidence that our method can achieve better results on these more difficult datasets.

## 5. Conclusion

This paper introduces a novel approach for contrastive learning that incorporates weakly-supervised information

through graph spectral theory. Extensive experiments and theoretical analysis validate its effectiveness. We propose semantic similarity as a continuous measure of class membership, offering new insights into utilizing ambiguous supervision in contrastive learning. Our framework has the potential to be extended to a wider range of scenarios, including bag-level weak supervision and multi-modal matching.

## Acknowledgements

This work was supported by the National Science Foundation of China (62206049, 62176055), and the Big Data Computing Center of Southeast University. We would like to thank anonymous reviewers for their constructive suggestions.

## Impact Statement

Our work may have broader implications, including the potential for increased unemployment among traditional data annotators as annotation quality standards decline. Additionally, careful attention is needed to address privacy concerns, as using raw data from web scraping can effectively train high-accuracy models.

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

# Supplementary Material

## A. Notation

Table 3. List of common mathematical symbols used in this paper.

| Symbol | Definition |
|---|---|
| $\mathcal{X}, \mathcal{Y} = \{y^1, \ldots, y^c\}$ | Instance space and label space respectively |
| $\mathcal{Q} = \{q^1, \ldots, q^v\}$ | Weakly-supervised information space |
| $\mathcal{D}$ | Joint distribution on $(\mathcal{X}, \mathcal{Y})$ for target pattern |
| $\mathcal{D}_\mathcal{Q}$ | Joint distribution on $(\mathcal{X}, \mathcal{Q})$ for weakly-supervised pattern which generated form $\mathcal{D}$ |
| $\boldsymbol{T}(x) \in \mathbb{R}^{v \times c}$ | Transition matrix with $\boldsymbol{T}(x)_{i,j} = \mathcal{D}_\mathcal{Q}(q^i \mid x, y^j)$ |
| $\mathcal{D}(\boldsymbol{y} \mid x)$ | The abbreviation of vector $\left[\mathcal{D}(y^1 \mid x), \ldots, \mathcal{D}(y^c \mid x)\right]^T$ |
| $\mathcal{D}_\mathcal{Q}(\boldsymbol{q} \mid x)$ | The abbreviation of vector $\left[\mathcal{D}_\mathcal{Q}(q^1 \mid x), \ldots, \mathcal{D}_\mathcal{Q}(q^v \mid x)\right]^T$ |
| $\mathbb{P}$ | The abbreviation of probability on joint distribution $(\mathcal{X}, \mathcal{Y}, \mathcal{Q})$ induced by $\mathcal{D}$ and $\mathcal{D}_\mathcal{Q}$ |
| $D_\mathcal{Q}$ | Set of training samples with corresponding weakly-supervised information |
| $D_\mathcal{U}$ | Set of unlabeled training samples |
| $D = D_\mathcal{Q} \cup D_\mathcal{U}$ | Final training dataset |
| $n, n_q, n_u$ | Size of $D, D_\mathcal{Q}, D_\mathcal{U}$, respectively |
| $(\tilde{x}, \tilde{y}), (\tilde{x}', \tilde{y}')$ | Two naturally data points with their label sampling form $\mathcal{D}$ |
| $x, x'$ | Two augmentation data points form $\tilde{x}$ and $\tilde{x}'$ respectively |
| $\mathcal{A}(\cdots \mid \tilde{x})$ | Distribution of the data augmentation |
| $w_{x,x'}$ | The edge weight of the edge connecting the two augmented samples constructed |
| $w_{x,x'}^u$ | The edge weight constructed through self-supervised connectivity |
| $w_{x,x'}^l$ | The edge weight constructed through fully-supervised |
| $w_{x,x'}^{wl}$ | The edge weight constructed through weakly-supervised connectivity |
| $\alpha, \beta$ | perturbation coefficients for integrating self-supervised and supervised information. |
| $\boldsymbol{A}$ | Adjacency matrix of constructed augmentation graph |
| $\widetilde{\boldsymbol{A}}$ | Normalized adjacency matrix of constructed augmentation graph |
| $\boldsymbol{S} : \mathcal{X} \to \mathbb{R}^{c \times v}$ | The recovery matrix used for compute the semantic similarity of samples |
| $S\left((\tilde{x}, \tilde{q}), (\tilde{x}', \tilde{q}')\right)$ | Computed semantic similarity of samples with corresponding weakly-supervised information |
| $\widehat{\boldsymbol{S}} : \mathcal{X} \to \mathbb{R}^{c \times v}$ | Constructed approximate recovery matrix |
| $\boldsymbol{F}$ | Embedding matrix with $\boldsymbol{F}_{x,:} \in \mathbb{R}^d$ is embedding of sample x |
| $f : \mathcal{X} \to \mathbb{R}^d$ | Learned feature embedding |
| $\boldsymbol{B} \in \mathbb{R}^{d \times c}$ | Linear classifier weight |
| $h_{f,\boldsymbol{B}}(x)$ | Prediction for augmentation data $x$ with feature $f$ and classifier $\boldsymbol{B}$ |
| $\widetilde{h}_{f,\boldsymbol{B}}(\tilde{x})$ | Ensemble prediction for naturally data $\tilde{x}$ with feature $f$ and classifier $\boldsymbol{B}$ |
| $g(x)$ | Posterior category probability give $x$ predicted by neural networks |
| $\varepsilon(f)$ | Linear probe error of feature embedding $f$ |
| $\rho$ | Sparsest partition of a graph in Definition. 3.3 |
| $\rho^u$ | Sparsest partition of self-supervised connectivity graph $\boldsymbol{A}^u$ |
| $\eta_0, \ldots, \eta_4$ | Constant only related to $\mathcal{F}_\infty$ and features dimension $d$ |
| $\Delta_\lambda$ | Eigenvalue gap between the $\lfloor 3d/4 \rfloor$-th and the $d$-th eigenvalue of graph $\boldsymbol{A}$ |
| $\eta_0', \ldots, \eta_4'$ | Proportional amplification of constant $\eta_0, \ldots, \eta_4$ |

# B. Proof of Section. 2

In this section, we provide the proof details of the proposition in the Section. 2.

## B.1. Proof of Proposition. 2.1

**Proposition B.1.** *(Recap of Proposition 2.1). For any $\boldsymbol{S} : \mathcal{X} \to \mathbb{R}^{c \times n}$ that satisfies the condition: $\mathbb{P}(\boldsymbol{y} \mid x) = \boldsymbol{S}(x)\mathbb{P}(\boldsymbol{q} \mid x)$ holds almost everywhere in $\mathcal{X}$, the following equation holds:*

$$w_{x,x'}^{wl}(\boldsymbol{S}) \triangleq \mathbb{E}_{(\tilde{x},\tilde{q}),(\tilde{x}',\tilde{q}') \sim \mathbb{P}(\mathcal{X},\mathcal{Q})^2} \left[ S\left((\tilde{x},\tilde{q}),(\tilde{x}',\tilde{q}')\right) \mathcal{A}(x \mid \tilde{x})\mathcal{A}(x' \mid \tilde{x}') \right] = w_{x,x'}^l, \tag{30}$$

*where $S\left((\tilde{x},\tilde{q}),(\tilde{x}',\tilde{q}')\right) = \boldsymbol{S}(\tilde{x})_{:,\tilde{q}}^T \boldsymbol{S}(\tilde{x}')_{:,\tilde{q}'}$.*

*Proof.* The proof is direct, on the one hand, we can expand $w_{x,x'}^l$ and obtain:

$$\begin{aligned}
w_{x,x'}^l &= \mathbb{E}_{(\tilde{x},\tilde{y}),(\tilde{x}',\tilde{y}') \sim \mathbb{P}(\mathcal{X},\mathcal{Y})^2} \left[ \mathbb{I}\left[\tilde{y} = \tilde{y}'\right] \mathcal{A}(x \mid \tilde{x})\mathcal{A}(x' \mid \tilde{x}') \right] \\
&= \int \sum_{\tilde{y},\tilde{y}' \in \mathcal{Y}^2} \mathbb{P}(\tilde{x},\tilde{y})\mathbb{P}(\tilde{x}',\tilde{y}')\mathbb{I}\left[\tilde{y} = \tilde{y}'\right] \mathcal{A}(x \mid \tilde{x})\mathcal{A}(x' \mid \tilde{x}')\mathrm{d}\tilde{x}\mathrm{d}\tilde{x}' \\
&= \int \mathbb{P}(\tilde{x})\mathbb{P}(\tilde{x}')\mathrm{d}\tilde{x}\mathrm{d}\tilde{x}' \sum_{\tilde{y},\tilde{y}' \in \mathcal{Y}^2} \mathbb{P}(\tilde{y} \mid \tilde{x})\mathbb{P}(\tilde{y}' \mid \tilde{x}')\mathbb{I}\left[\tilde{y} = \tilde{y}'\right] \mathcal{A}(x \mid \tilde{x})\mathcal{A}(x' \mid \tilde{x}') \\
&= \int \mathbb{P}(\tilde{x})\mathbb{P}(\tilde{x}')\mathrm{d}\tilde{x}\mathrm{d}\tilde{x}' \sum_{\tilde{y} \in \mathcal{Y}} \mathbb{P}(\tilde{y} \mid \tilde{x})\mathbb{P}(\tilde{y} \mid \tilde{x}')\mathcal{A}(x \mid \tilde{x})\mathcal{A}(x' \mid \tilde{x}') \\
&= \int \mathbb{P}(\tilde{x})\mathbb{P}(\tilde{x}')\mathrm{d}\tilde{x}\mathrm{d}\tilde{x}' \left( \mathbb{P}(\boldsymbol{y} \mid \tilde{x})^T\mathbb{P}(\boldsymbol{y} \mid \tilde{x}') \right) \mathcal{A}(x \mid \tilde{x})\mathcal{A}(x' \mid \tilde{x}') \\
&= \mathbb{E}_{\tilde{x},\tilde{x}' \sim \mathcal{X}^2} \left[ \mathbb{P}(\boldsymbol{y} \mid \tilde{x})^T\mathbb{P}(\boldsymbol{y} \mid \tilde{x}')\mathcal{A}(x \mid \tilde{x})\mathcal{A}(x' \mid \tilde{x}') \right].
\end{aligned} \tag{31}$$

On the other hand, we can expand the $w_{x,x'}^{wl}$ and obtain:

$$\begin{aligned}
w_{x,x'}^{wl} &= \mathbb{E}_{(\tilde{x},\tilde{q}),(\tilde{x}',\tilde{q}') \sim \mathbb{P}(\mathcal{X},\mathcal{Q})^2} \left[ S\left((\tilde{x},\tilde{q}),(\tilde{x}',\tilde{q}')\right) \mathcal{A}(x \mid \tilde{x})\mathcal{A}(x' \mid \tilde{x}') \right] \\
&= \int \sum_{\tilde{q},\tilde{q}' \in \mathcal{Q}^2} \mathbb{P}(\tilde{x},\tilde{q})\mathbb{P}(\tilde{x}',\tilde{q}')S\left((\tilde{x},\tilde{q}),(\tilde{x}',\tilde{q}')\right) \mathcal{A}(x \mid \tilde{x})\mathcal{A}(x' \mid \tilde{x}')\mathrm{d}\tilde{x}\mathrm{d}\tilde{x}' \\
&= \int \mathbb{P}(\tilde{x})\mathbb{P}(\tilde{x}')\mathrm{d}\tilde{x}\mathrm{d}\tilde{x}' \sum_{\tilde{q},\tilde{q}' \in \mathcal{Q}^2} \mathbb{P}(\tilde{q} \mid \tilde{x})\mathbb{P}(\tilde{q}' \mid \tilde{x}')\boldsymbol{S}(\tilde{x})_{:,\tilde{q}}^T\boldsymbol{S}(\tilde{x}')_{:,\tilde{q}'}\mathcal{A}(x \mid \tilde{x})\mathcal{A}(x' \mid \tilde{x}') \\
&= \int \mathbb{P}(\tilde{x})\mathbb{P}(\tilde{x}')\mathrm{d}\tilde{x}\mathrm{d}\tilde{x}'\mathbb{P}(\boldsymbol{q} \mid \tilde{x})^T \left( \boldsymbol{S}(\tilde{x})^T\boldsymbol{S}(\tilde{x}') \right) \mathbb{P}(\boldsymbol{q} \mid \tilde{x}')\mathcal{A}(x \mid \tilde{x})\mathcal{A}(x' \mid \tilde{x}') \\
&= \mathbb{E}_{\tilde{x},\tilde{x}' \sim \mathcal{X}^2} \left[ \mathbb{P}(\boldsymbol{q} \mid \tilde{x})^T \left( \boldsymbol{S}(\tilde{x})^T\boldsymbol{S}(\tilde{x}') \right) \mathbb{P}(\boldsymbol{q} \mid \tilde{x}')\mathcal{A}(x \mid \tilde{x})\mathcal{A}(x' \mid \tilde{x}') \right].
\end{aligned} \tag{32}$$

When $\boldsymbol{S}$ satisfies the condition of $\mathbb{P}(\boldsymbol{y} \mid x) = \boldsymbol{S}(x)\mathbb{P}(\boldsymbol{q} \mid x)$ holds almost everywhere in $\mathcal{X}$, we have follow equation holds almost everywhere in $\mathcal{X}^2$:

$$\mathbb{P}(\boldsymbol{q} \mid \tilde{x})^T \left( \boldsymbol{S}(\tilde{x})^T\boldsymbol{S}(\tilde{x}') \right) \mathbb{P}(\boldsymbol{q} \mid \tilde{x}') = \left( \boldsymbol{S}(\tilde{x})\mathbb{P}(\boldsymbol{q} \mid \tilde{x}) \right)^T \left( \boldsymbol{S}(\tilde{x}')\mathbb{P}(\boldsymbol{q} \mid \tilde{x}') \right) = \mathbb{P}(\boldsymbol{y} \mid \tilde{x})^T\mathbb{P}(\boldsymbol{y} \mid \tilde{x}') \tag{33}$$

Combining Equations 31, 32, and 33, we finish the proof. $\qquad\square$

## B.2. Proof of Proposition. 2.3

**Proposition B.2.** *(Recap of Proposition. 2.3). We define $\boldsymbol{F}_{x,:} = \sqrt{\boldsymbol{A}_x}f_x$ and weakly-supervised contrastive loss $\mathcal{L}_{wsc}$ as follows:*

$$\mathcal{L}_{wsc}(f) \triangleq -2\alpha\mathcal{L}_1(f) - 2\beta\mathcal{L}_2(f) + \alpha^2\mathcal{L}_3(f) + \beta^2\mathcal{L}_4(f) + 2\alpha\beta\mathcal{L}_5(f), \tag{34}$$

*where:*

$$\mathcal{L}_1(f) \triangleq \mathbb{E}_{\tilde{x} \sim \mathbb{P}(\mathcal{X}),x,x' \sim \mathcal{A}(\cdot|\tilde{x})} \left[ f_x f_{x'}^T \right], \tag{35}$$

$$\mathcal{L}_2(f) \triangleq \mathbb{E}_{(\tilde{x},\tilde{q}),(\tilde{x}',\tilde{q}')\sim\mathbb{P}(\mathcal{X},\mathcal{Q})^2, x\sim\mathcal{A}(\cdot|\tilde{x}), x'\sim\mathcal{A}(\cdot|\tilde{x}')}\left[S((\tilde{x},\tilde{q}),(\tilde{x}',\tilde{q}'))f_x f_{x'}^T\right], \tag{36}$$

$$\mathcal{L}_3(f) \triangleq \mathbb{E}_{\tilde{x},\tilde{x}'\sim\mathbb{P}(\mathcal{X})^2, x\sim\mathcal{A}(\cdot|\tilde{x}), x'\sim\mathcal{A}(\cdot|\tilde{x}')}\left[\left(f_x f_{x'}^T\right)^2\right], \tag{37}$$

$$\mathcal{L}_4(f) \triangleq \mathbb{E}_{(\tilde{x},\tilde{q}),(\tilde{x}',\tilde{q}')\sim\mathbb{P}(\mathcal{X},\mathcal{Q})^2, x\sim\mathcal{A}(\cdot|\tilde{x}), x'\sim\mathcal{A}(\cdot|\tilde{x}')}\left[S'(\tilde{x},\tilde{q})S'(\tilde{x}',\tilde{q}')\left(f_x f_{x'}^T\right)^2\right], \tag{38}$$

$$\mathcal{L}_5(f) \triangleq \mathbb{E}_{(\tilde{x},\tilde{q})\sim\mathbb{P}(\mathcal{X},\mathcal{Q}), \tilde{x}'\sim\mathbb{P}(\mathcal{X}), x\sim\mathcal{A}(\cdot|\tilde{x}), x'\sim\mathcal{A}(\cdot|\tilde{x}')}\left[S'(\tilde{x},\tilde{q})\left(f_x f_{x'}^T\right)^2\right], \tag{39}$$

*where* $S\left((\tilde{x},\tilde{q}),(\tilde{x}',\tilde{q}')\right) = \boldsymbol{S}(\tilde{x})_{:,\tilde{q}}^T \boldsymbol{S}(\tilde{x}')_{:,\tilde{q}'}, S'(\tilde{x},\tilde{q}) = \boldsymbol{S}(\tilde{x})_{:,\tilde{q}}^T\mathbb{P}(\boldsymbol{y}).$

*Then the following equation holds:*

$$\left\|\widetilde{\boldsymbol{A}} - \boldsymbol{F}\boldsymbol{F}^T\right\|_F = \mathcal{L}_{wsc}(f) + Const, \tag{40}$$

*Where* $\boldsymbol{A}_{x,x'} = \alpha w_{x,x'}^u + \beta w_{x,x'}^{wl}$, *and* $\widetilde{\boldsymbol{A}}_{x,x'} = \frac{\boldsymbol{A}_{x,x'}}{\sqrt{\boldsymbol{A}_x}\sqrt{\boldsymbol{A}_{x'}}}$.

*Furthermore, when assuming uniform class distribution* $\mathbb{P}(\boldsymbol{y}) = \left[\frac{1}{c}, \ldots, \frac{1}{c}\right]^T$, *above* $\mathcal{L}_{wsc}(f)$ *can also rewrite as:*

$$\mathcal{L}_{wsc}(f) \triangleq -2\alpha\mathcal{L}_1(f) - 2\beta\mathcal{L}_2(f) + \left(\alpha + \frac{\beta}{c}\right)^2 \mathcal{L}_3(f). \tag{41}$$

*Proof.* First we can expand $\left\|\widetilde{\boldsymbol{A}} - \boldsymbol{F}\boldsymbol{F}^T\right\|_F$ as follows:

$$\begin{aligned}
\left\|\widetilde{\boldsymbol{A}} - \boldsymbol{F}\boldsymbol{F}^T\right\|_F &= \sum_{x,x'}\left(\frac{\boldsymbol{A}_{x,x'}}{\sqrt{\boldsymbol{A}_x}\sqrt{\boldsymbol{A}_{x'}}} - \boldsymbol{F}_{x,:}\boldsymbol{F}_{x',:}^T\right)^2 \\
&= \sum_{x,x'}\left(\boldsymbol{F}_{x,:}\boldsymbol{F}_{x',:}^T\right)^2 - 2\frac{\boldsymbol{A}_{x,x'}}{\sqrt{\boldsymbol{A}_x}\sqrt{\boldsymbol{A}_{x'}}}\boldsymbol{F}_{x,:}\boldsymbol{F}_{x',:}^T + Const \\
&= \sum_{x,x'}\boldsymbol{A}_x\boldsymbol{A}_{x'}\left(f_x f_{x'}^T\right)^2 - 2\boldsymbol{A}_{x,x'}f_x f_{x'}^T + Const.
\end{aligned} \tag{42}$$

Next, We analyze the two terms in the RHS of Equation 42 separately. We first expand the second term in RHS of Equation 42 as follows:

$$\begin{aligned}
\sum_{x,x'} -2\boldsymbol{A}_{x,x'}f_x f_{x'}^T &= \sum_{x,x'} -2(\alpha w_{x,x'}^u + \beta w_{x,x'}^{wl})f_x f_{x'}^T \\
&= -2\alpha\sum_{x,x'}\mathbb{E}_{\tilde{x}\sim\mathbb{P}(X)}[\mathcal{A}(x\mid\tilde{x})\mathcal{A}(x'\mid\tilde{x})]f_x f_{x'}^T \\
&\quad -2\beta\sum_{x,x'}\mathbb{E}_{(\tilde{x},\tilde{q}),(\tilde{x}',\tilde{q}')\sim\mathbb{P}(\mathcal{X},\mathcal{Q})^2}\left[S\left((\tilde{x},\tilde{q}),(\tilde{x}',\tilde{q}')\right)\mathcal{A}(x\mid\tilde{x})\mathcal{A}(x'\mid\tilde{x}')\right]f_x f_{x'}^T \\
&= -2\alpha\mathbb{E}_{\tilde{x}\sim\mathbb{P}(X)}\left[\sum_{x,x'}\mathcal{A}(x\mid\tilde{x})\mathcal{A}(x'\mid\tilde{x})f_x f_{x'}^T\right] \\
&\quad -2\beta\mathbb{E}_{(\tilde{x},\tilde{q}),(\tilde{x}',\tilde{q}')\sim\mathbb{P}(\mathcal{X},\mathcal{Q})^2}\left[\sum_{x,x'}S\left((\tilde{x},\tilde{q}),(\tilde{x}',\tilde{q}')\right)\mathcal{A}(x\mid\tilde{x})\mathcal{A}(x'\mid\tilde{x}')f_x f_{x'}^T\right] \\
&= -2\alpha\mathbb{E}_{\tilde{x}\sim\mathbb{P}(X)}\left[\mathbb{E}_{x\sim\mathcal{A}(\cdot|\tilde{x}), x'\sim\mathcal{A}(\cdot|\tilde{x})}\left[f_x f_{x'}^T\right]\right] \\
&\quad -2\beta\mathbb{E}_{(\tilde{x},\tilde{q}),(\tilde{x}',\tilde{q}')\sim\mathbb{P}(\mathcal{X},\mathcal{Q})^2}\left[S\left((\tilde{x},\tilde{q}),(\tilde{x}',\tilde{q}')\right)\mathbb{E}_{x\sim\mathcal{A}(\cdot|\tilde{x}), x'\sim\mathcal{A}(\cdot|\tilde{x}')}\left[f_x f_{x'}^T\right]\right].
\end{aligned} \tag{43}$$

By substituting the definitions of $\mathcal{L}_1(f)$ and $\mathcal{L}_2(f)$ into the Equation 43, we have:

$$\sum_{x,x'} -2\boldsymbol{A}_{x,x'}f_x f_{x'}^T = -2\alpha\mathcal{L}_1(f) - 2\beta\mathcal{L}_2(f) \tag{44}$$

On the other hand, we have:

$$\boldsymbol{A}_x = \sum_{x'} A_{x,x'} = \alpha \sum_{x'} w^u_{x,x'} + \beta \sum_{x'} w^{wl}_{x,x'}$$

$$= \alpha \mathbb{E}_{\tilde{x} \sim \mathbb{P}(X)} \left[ \sum_{x'} \mathcal{A}(x \mid \tilde{x}) \mathcal{A}(x' \mid \tilde{x}) \right]$$

$$\beta \mathbb{E}_{(\tilde{x},\tilde{q}),(\tilde{x}',\tilde{q}') \sim \mathbb{P}(\mathcal{X},\mathcal{Q})^2} \left[ \sum_{x'} S\left( (\tilde{x},\tilde{q}),(\tilde{x}',\tilde{q}') \right) \mathcal{A}(x \mid \tilde{x}) \mathcal{A}(x' \mid \tilde{x}') \right] \tag{45}$$

$$= \alpha \mathbb{E}_{\tilde{x} \sim \mathbb{P}(X)} \left[ \mathcal{A}(x \mid \tilde{x}) \right]$$

$$+ \beta \mathbb{E}_{(\tilde{x},\tilde{q}),(\tilde{x}',\tilde{q}') \sim \mathbb{P}(\mathcal{X},\mathcal{Q})^2} \left[ \sum_{x'} S\left( (\tilde{x},\tilde{q}),(\tilde{x}',\tilde{q}') \right) \mathcal{A}(x \mid \tilde{x}) \mathcal{A}(x' \mid \tilde{x}') \right].$$

By the definition of $S\left( (\tilde{x},\tilde{q}),(\tilde{x}',\tilde{q}') \right)$, we can rewrite the second term of RHS of Equation 45 as follows:

$$\mathbb{E}_{(\tilde{x},\tilde{q}),(\tilde{x}',\tilde{q}') \sim \mathbb{P}(\mathcal{X},\mathcal{Q})^2} \left[ \sum_{x'} S\left( (\tilde{x},\tilde{q}),(\tilde{x}',\tilde{q}') \right) \mathcal{A}(x \mid \tilde{x}) \mathcal{A}(x' \mid \tilde{x}') \right]$$

$$= \mathbb{E}_{(\tilde{x},\tilde{q}) \sim \mathbb{P}(\mathcal{X},\mathcal{Q})} \left[ \mathcal{A}(x \mid \tilde{x}) \boldsymbol{S}(\tilde{x})^T_{:,\tilde{q}} \mathbb{E}_{\tilde{x}' \sim \mathcal{X}} \left[ \boldsymbol{S}(\tilde{x}') \mathbb{P}(\boldsymbol{q} \mid \tilde{x}') \right] \sum_{x'} \mathcal{A}(x' \mid \tilde{x}') \right] \tag{46}$$

$$= \mathbb{E}_{(\tilde{x},\tilde{q}) \sim \mathbb{P}(\mathcal{X},\mathcal{Q})} \left[ \mathcal{A}(x \mid \tilde{x}) \boldsymbol{S}(\tilde{x})^T_{:,\tilde{q}} \mathbb{E}_{\tilde{x}' \sim \mathcal{X}} \left[ \mathbb{P}(\boldsymbol{y} \mid \tilde{x}') \right] \right]$$

$$= \mathbb{E}_{(\tilde{x},\tilde{q}) \sim \mathbb{P}(\mathcal{X},\mathcal{Q})} \left[ \mathcal{A}(x \mid \tilde{x}) \boldsymbol{S}(\tilde{x})^T_{:,\tilde{q}} \mathbb{P}(\boldsymbol{y}) \right]$$

$$= \mathbb{E}_{(\tilde{x},\tilde{q}) \sim \mathbb{P}(\mathcal{X},\mathcal{Q})} \left[ S'(\tilde{x},\tilde{q}) \mathcal{A}(x \mid \tilde{x}) \right],$$

where we defined $S'(\tilde{x},\tilde{q}) = \boldsymbol{S}(\tilde{x})^T_{:,\tilde{q}} \mathbb{P}(\boldsymbol{y})$ before. By substituting Equation 46 into Equation 45, we have:

$$\boldsymbol{A}_x = \alpha \mathbb{E}_{\tilde{x} \sim \mathbb{P}(X)} \left[ \mathcal{A}(x \mid \tilde{x}) \right] + \beta \mathbb{E}_{(\tilde{x},\tilde{q}) \sim \mathbb{P}(\mathcal{X},\mathcal{Q})} \left[ S'(\tilde{x},\tilde{q}) \mathcal{A}(x \mid \tilde{x}) \right]. \tag{47}$$

Using results of Equation 47, we can expand the first term in RHS of Equation 42 as follows:

$$\sum_{x,x'} \boldsymbol{A}_x \boldsymbol{A}_{x'} \left( f_x f^T_{x'} \right)^2 = \sum_{x,x'} \left( \alpha \mathbb{E}_{\tilde{x} \sim \mathbb{P}(\mathcal{X})} \left[ \mathcal{A}(x \mid \tilde{x}) \right] + \beta \mathbb{E}_{(\tilde{x},\tilde{q}) \sim \mathbb{P}(\mathcal{X},\mathcal{Q})} [S'(\tilde{x},\tilde{q}) \mathcal{A}(x \mid \tilde{x})] \right)$$

$$\left( \alpha \mathbb{E}_{\tilde{x}' \sim \mathbb{P}(\mathcal{X})} [\mathcal{A}(x' \mid \tilde{x}')] + \beta E_{(\tilde{x}',\tilde{q}') \sim \mathbb{P}(\mathcal{X},\mathcal{Q})} [S'(\tilde{x}',\tilde{q}') \mathcal{A}(x' \mid \tilde{x}')] \right) \left( f_x f^T_{x'} \right)^2$$

$$= \alpha^2 \mathbb{E}_{\tilde{x},\tilde{x}' \sim \mathbb{P}(\mathcal{X}^2)} \left[ \sum_{x,x'} \mathcal{A}(x \mid \tilde{x}) \mathcal{A}(x' \mid \tilde{x}') \left( f_x f^T_{x'} \right)^2 \right] \tag{48}$$

$$+ \beta^2 \mathbb{E}_{(\tilde{x},\tilde{q}),(\tilde{x}',\tilde{q}') \sim \mathbb{P}(\mathcal{X},\mathcal{Q})^2} \left[ \sum_{x,x'} S'(\tilde{x},\tilde{q}) S'(\tilde{x}',\tilde{q}') \mathcal{A}(x \mid \tilde{x}) \mathcal{A}(x' \mid \tilde{x}') \left( f_x f^T_{x'} \right)^2 \right]$$

$$+ 2\alpha\beta \mathbb{E}_{(\tilde{x},\tilde{q}) \sim \mathbb{P}(\mathcal{X},\mathcal{Q}),\tilde{x}' \sim \mathbb{P}(\mathcal{X})} \left[ S'(\tilde{x},\tilde{q}) \sum_{x,x'} \mathcal{A}(x \mid \tilde{x}) \mathcal{A}(x' \mid \tilde{x}') \left( f_x f^T_{x'} \right)^2 \right].$$

By substituting the definitions of $\mathcal{L}_3(f)$, $\mathcal{L}_4(f)$ and $\mathcal{L}_5(f)$ into RHS of Equation 48, we rewrite its three terms as follows, respectively:

$$\alpha^2 \mathbb{E}_{\tilde{x},\tilde{x}' \sim \mathbb{P}(\mathcal{X}^2)} \left[ \sum_{x,x'} \mathcal{A}(x \mid \tilde{x}) \mathcal{A}(x' \mid \tilde{x}') \left( f_x f^T_{x'} \right)^2 \right] = \alpha^2 \mathbb{E}_{\tilde{x},\tilde{x}' \sim \mathbb{P}(\mathcal{X})^2, x \sim \mathcal{A}(\cdot|\tilde{x}), x' \sim \mathcal{A}(\cdot|\tilde{x}')} \left[ \left( f_x f^T_{x'} \right)^2 \right] = \alpha^2 \mathcal{L}_3(f). \tag{49}$$

$$\beta^2 \mathbb{E}_{(\tilde{x},\tilde{q}),(\tilde{x}',\tilde{q}') \sim \mathbb{P}(\mathcal{X},\mathcal{Q})^2} \left[ \sum_{x,x'} S'(\tilde{x},\tilde{q}) S'(\tilde{x}',\tilde{q}') \mathcal{A}(x \mid \tilde{x}) \mathcal{A}(x' \mid \tilde{x}') \left( f_x f^T_{x'} \right)^2 \right] \tag{50}$$

$$= \beta^2 \mathbb{E}_{(\tilde{x},\tilde{q}),(\tilde{x}',\tilde{q}') \sim \mathbb{P}(\mathcal{X},\mathcal{Q})^2, x \sim \mathcal{A}(\cdot|\tilde{x}), x' \sim \mathcal{A}(\cdot|\tilde{x}')} \left[ S'(\tilde{x},\tilde{q}) S'(\tilde{x}',\tilde{q}') \left( f_x f^T_{x'} \right)^2 \right] = \beta^2 \mathcal{L}_4(f).$$

$$2\alpha\beta\mathbb{E}_{(\tilde{x},\tilde{q})\sim\mathbb{P}(\mathcal{X},\mathcal{Q}),\tilde{x}'\sim\mathbb{P}(\mathcal{X})}\left[S'(\tilde{x},\tilde{q})\sum_{x,x'}\mathcal{A}(x\mid\tilde{x})\mathcal{A}(x'\mid\tilde{x}')\left(f_x f_{x'}^T\right)^2\right]$$

$$=2\alpha\beta\mathbb{E}_{(\tilde{x},\tilde{q})\sim\mathbb{P}(\mathcal{X},\mathcal{Q}),\tilde{x}'\sim\mathbb{P}(\mathcal{X}),x\sim\mathcal{A}(\cdot|\tilde{x}),x'\sim\mathcal{A}(\cdot|\tilde{x}')}\left[S'(\tilde{x},\tilde{q})\left(f_x f_{x'}^T\right)^2\right]=2\alpha\beta\mathcal{L}_5(f). \tag{51}$$

By combining the results of Equations 48, 49, 50, and 51, we obtain:

$$\sum_{x,x'}\boldsymbol{A}_x\boldsymbol{A}_{x'}\left(f_x f_{x'}^T\right)^2 = \alpha^2\mathcal{L}_3 + \beta^2\mathcal{L}_4 + 2\alpha\beta\mathcal{L}_5. \tag{52}$$

By substituting the results of Equations 44 and 52 into Equation 42, we can obtain the Equation 40. Furthermore, when assume uniform class distribution $\mathbb{P}(\boldsymbol{y})=\left[\frac{1}{c},\ldots,\frac{1}{c}\right]^T$, above $\mathcal{L}_4(f)$ can be rewritten as follows:

$$\mathcal{L}_4(f)=\mathbb{E}_{(\tilde{x},\tilde{q}),(\tilde{x}',\tilde{q}')\sim\mathbb{P}(\mathcal{X},\mathcal{Q})^2,x\sim\mathcal{A}(\cdot|\tilde{x}),x'\sim\mathcal{A}(\cdot|\tilde{x}')}\left[S'(\tilde{x},\tilde{q})S'(\tilde{x}',\tilde{q}')\left(f_x f_{x'}^T\right)^2\right]$$

$$=\mathbb{E}_{\tilde{x},\tilde{x}'\sim\mathbb{P}(\mathcal{X})^2,x\sim\mathcal{A}(\cdot|\tilde{x}),x'\sim\mathcal{A}(\cdot|\tilde{x}')}\left[\frac{1}{c^2}\sum_{\tilde{q}\in\mathcal{Q},\tilde{y}'\in\mathcal{Y}}\boldsymbol{S}(\tilde{x})_{\tilde{y},\tilde{q}}\mathbb{P}(\tilde{q}\mid\tilde{x})\sum_{\tilde{q}'\in\mathcal{Q},\tilde{y}'\in\mathcal{Y}}\boldsymbol{S}(\tilde{x}')_{\tilde{y}',\tilde{q}'}\mathbb{P}(\tilde{q}'\mid\tilde{x}')\left(f_x f_{x'}^T\right)^2\right] \tag{53}$$

$$=\mathbb{E}_{\tilde{x},\tilde{x}'\sim\mathbb{P}(\mathcal{X})^2,x\sim\mathcal{A}(\cdot|\tilde{x}),x'\sim\mathcal{A}(\cdot|\tilde{x}')}\left[\frac{1}{c^2}\sum_{\tilde{y}\in\mathcal{Y}}(\boldsymbol{S}(\tilde{x})\mathbb{P}(\boldsymbol{q}\mid\tilde{x}))_{\tilde{y}}\sum_{\tilde{y}'\in\mathcal{Y}}(\boldsymbol{S}(\tilde{x}')\mathbb{P}(\boldsymbol{q}\mid\tilde{x}'))_{\tilde{y}'}\left(f_x f_{x'}^T\right)^2\right]$$

$$=\frac{1}{c^2}\mathbb{E}_{\tilde{x},\tilde{x}'\sim\mathbb{P}(\mathcal{X})^2,x\sim\mathcal{A}(\cdot|\tilde{x}),x'\sim\mathcal{A}(\cdot|\tilde{x}')}\left[\left(f_x f_{x'}^T\right)^2\right]=\frac{1}{c^2}\mathcal{L}_3(f).$$

Similarly, $\mathcal{L}_5(f)$ can be rewritten as follows:

$$\mathcal{L}_5(f)=\mathbb{E}_{(\tilde{x},\tilde{q})\sim\mathbb{P}(\mathcal{X},\mathcal{Q}),\tilde{x}'\sim\mathbb{P}(\mathcal{X}),x\sim\mathcal{A}(\cdot|\tilde{x}),x'\sim\mathcal{A}(\cdot|\tilde{x}')}\left[S'(\tilde{x},\tilde{q})\left(f_x f_{x'}^T\right)^2\right]$$

$$=\mathbb{E}_{\tilde{x},\tilde{x}'\sim\mathbb{P}(\mathcal{X})^2,x\sim\mathcal{A}(\cdot|\tilde{x}),x'\sim\mathcal{A}(\cdot|\tilde{x}')}\left[\frac{1}{c}\sum_{\tilde{q}\in\mathcal{Q},\tilde{y}'\in\mathcal{Y}}\boldsymbol{S}(\tilde{x})_{\tilde{y},\tilde{q}}\mathbb{P}(\tilde{q}\mid\tilde{x})\left(f_x f_{x'}^T\right)^2\right] \tag{54}$$

$$=\frac{1}{c}\mathbb{E}_{\tilde{x},\tilde{x}'\sim\mathbb{P}(\mathcal{X})^2,x\sim\mathcal{A}(\cdot|\tilde{x}),x'\sim\mathcal{A}(\cdot|\tilde{x}')}\left[\sum_{\tilde{y}\in\mathcal{Y}}(\boldsymbol{S}(\tilde{x})\mathbb{P}(\boldsymbol{q}\mid\tilde{x}))_{\tilde{y}}\left(f_x f_{x'}^T\right)^2\right]$$

$$=\frac{1}{c}\mathbb{E}_{\tilde{x},\tilde{x}'\sim\mathbb{P}(\mathcal{X})^2,x\sim\mathcal{A}(\cdot|\tilde{x}),x'\sim\mathcal{A}(\cdot|\tilde{x}')}\left[\left(f_x f_{x'}^T\right)^2\right]=\frac{1}{c}\mathcal{L}_3(f).$$

Combining the results of Equations 53 and 54, $\mathcal{L}_{wsc}(f)$ can be rewritten as follows thus finishing the proof of Equation 41.

$$\mathcal{L}_{wsc}(f)=-2\alpha\mathcal{L}_1(f)-2\beta\mathcal{L}_2(f)+\alpha^2\mathcal{L}_3(f)+\beta^2\mathcal{L}_4(f)+2\alpha\beta\mathcal{L}_5(f)$$

$$=-2\alpha\mathcal{L}_1(f)-2\beta\mathcal{L}_2(f)+\left(\alpha^2+\left(\frac{\beta}{c}\right)^2+2\frac{\alpha\beta}{c}\right)\mathcal{L}_3(f) \tag{55}$$

$$=-2\alpha\mathcal{L}_1(f)-2\beta\mathcal{L}_2(f)+\left(\alpha+\frac{\beta}{c}\right)^2\mathcal{L}_3(f).$$

$\square$

# C. Proof of Section. 3

In this section, we provide the proof detail of theorems in the Section. 3. Our theoretical framework mainly follow theoretical framework in HaoChen et al. 2021. In order to eliminate technical complexity, we assume vertices of augmentation graph have to be a finite but exponentially large set. This is a reasonable assumption given that modern computers store data with finite bits so the possible number of all data has to be finite, and results can been easier generalized to the infinite case through Theorem F.2 in HaoChen et al. 2021. Moreover, without loss of generality, we assume a uniform class distribution to simplify the proof.

## C.1. Proof of Theorem 3.4

**Theorem C.1.** *(Recap of Theorem 3.4) Given augmentation graph $\mathcal{G}$ with perturbation adjacency matrix $\boldsymbol{A} = \alpha \boldsymbol{A}^u + \beta \boldsymbol{A}^{wl}(\boldsymbol{S})$ where $\boldsymbol{S}$ stratify properties in Proposition. 2.1, if $\mathcal{A}$ is $\gamma$-consistent with $\gamma > 0$ and representation dimension $d > 2c$, we have:*

$$\varepsilon(f^*) \preceq \tilde{O}\left(\gamma\left(\alpha^* \rho^u_{\lfloor \frac{d}{2} \rfloor} + \beta^*\left(1 - \frac{2c}{d}\right)\right)^{-2}\right), \tag{56}$$

*where $\rho^u$ is sparsest partition of self-supervised connectivity graph $\boldsymbol{A}^u$, and $\alpha^* = \frac{\alpha}{\alpha + \beta/c}$, $\beta^* = \frac{\beta/c}{\alpha + \beta/c}$.*

**Lemma C.2.** *For any proportionality constant $\eta > 0$, then linear probe error of features derived from $\eta \boldsymbol{A}$ is equal to features derived from $\boldsymbol{A}$.*

*Proof.* Assuming that $g^*$ is features derived form $\eta \boldsymbol{A}$, then through results of Proposition. B.2, we have $g^*_x = \frac{\boldsymbol{G}^*_x}{\sqrt{\eta \boldsymbol{A}_x}}$ where $\boldsymbol{G}^*$ define as follows:

$$\boldsymbol{G}^* = \arg\min \left\|\eta \widetilde{\boldsymbol{A}} - \boldsymbol{G}\boldsymbol{G}^T\right\|_F. \tag{57}$$

Since $\eta \widetilde{\boldsymbol{A}}_{x,x'} = \frac{\eta \boldsymbol{A}_{x,x'}}{\sqrt{\eta \boldsymbol{A}_x}\sqrt{\eta \boldsymbol{A}_{x'}}} = \frac{\boldsymbol{A}}{\sqrt{\boldsymbol{A}_x}\sqrt{\boldsymbol{A}_{x'}}} = \widetilde{\boldsymbol{A}}_{x,x'}$, we have:

$$\boldsymbol{G}^* = \arg\min \left\|\widetilde{\boldsymbol{A}} - \boldsymbol{G}\boldsymbol{G}^T\right\|_F = \boldsymbol{F}^*. \tag{58}$$

Hence features derived form $\eta \boldsymbol{A}$ can be rewritten as:

$$g^*_x = \frac{\boldsymbol{G}^*_x}{\sqrt{\eta \boldsymbol{A}_x}} = \frac{\boldsymbol{F}^*_x}{\sqrt{\eta \boldsymbol{A}_x}} = \frac{f^*_x}{\sqrt{\eta}}. \tag{59}$$

For any linear classifier $\boldsymbol{B} \in \mathbb{R}^{d \times c}$, we have:

$$h_{g^*, \boldsymbol{B}}(x) = \arg\max_{i \in \mathcal{Y}} \boldsymbol{B}^T_{:,i} g^*(x) = \arg\max_{i \in \mathcal{Y}} \frac{1}{\sqrt{\eta}} \boldsymbol{B}^T_{:,i} f^*(x) = h_{f^*, \boldsymbol{B}}(x). \tag{60}$$

In the end, we finish the proof with:

$$\varepsilon(g^*) == \min_{\boldsymbol{B} \in \mathbb{R}^{d \times c}} \mathbb{E}_{(\tilde{x}, \tilde{y}) \sim \mathbb{P}(\mathcal{X}, \mathcal{Y})}\left[\mathbb{I}\left[\tilde{h}_{g^*, \boldsymbol{B}}(\tilde{x}) \neq \tilde{y}\right]\right] = \min_{\boldsymbol{B} \in \mathbb{R}^{d \times c}} \mathbb{E}_{(\tilde{x}, \tilde{y}) \sim \mathbb{P}(\mathcal{X}, \mathcal{Y})}\left[\mathbb{I}\left[\tilde{h}_{f^*, \boldsymbol{B}}(\tilde{x}) \neq \tilde{y}\right]\right] = \varepsilon(f^*). \tag{61}$$

$\square$

Using results of Lemma C.2, we can assume $\alpha + \frac{\beta}{c} = 1$ in Theorem C.1 (When $\alpha + \frac{\beta}{c} \neq 1$, we consider perturbation graph with $\boldsymbol{A} = \frac{\alpha}{\alpha + \beta/c} \boldsymbol{A}^u + \frac{\beta}{\alpha + \beta/c} \boldsymbol{A}^{wl}$ ).

We are now ready to formally prove the theorem. Our proof relies partly on the theoretical results in HaoChen et al. 2021. For the sake of completeness, we will restate some of the theoretical results in HaoChen et al. 2021 in this section.

**Lemma C.3.** *If $\mathcal{A}$ is $\gamma$-consistent augmentation distribution, and $\hat{y}(x)$ is corresponding pseudo labeler for augmentation data such that satisfy condition in Definition 3.1, then for any linear classifier $\boldsymbol{B}$ and feature $f$, we have:*

$$\mathbb{E}_{(\tilde{x}, \tilde{y}) \sim \mathbb{P}(\mathcal{X}, \mathcal{Y})}\left[\mathbb{I}\left[\tilde{h}_{f, \boldsymbol{B}}(\tilde{x}) \neq \tilde{y}\right]\right] \leq 2\left(\mathbb{E}_{\tilde{x} \sim \mathbb{P}(\mathcal{X}), x \sim \mathcal{A}(\cdot | \tilde{x})}\left[\mathbb{I}\left[h_{f, \boldsymbol{B}}(x) \neq \hat{y}(x)\right]\right] + \gamma\right) \tag{62}$$

*Proof.* For any $\tilde{x}$ that satisfy $\tilde{h}_{f,\boldsymbol{B}}(\tilde{x}) \neq \tilde{y}$, we claim:

$$\mathbb{E}_{x \sim \mathcal{A}(\cdot|\tilde{x})} \left[ \mathbb{I} \left[ h_{f,\boldsymbol{B}}(x) \neq \tilde{y} \right] \right] \geq \frac{1}{2}. \tag{63}$$

Hence, we have:

$$
\begin{aligned}
\mathbb{E}_{(\tilde{x},\tilde{y}) \sim \mathbb{P}(\mathcal{X},\mathcal{Y})} \left[ \mathbb{I} \left[ \tilde{h}_{f,\boldsymbol{B}}(\tilde{x}) \neq \tilde{y} \right] \right] &\leq 2 \mathbb{E}_{(\tilde{x},\tilde{y}) \sim \mathbb{P}(\mathcal{X},\mathcal{Y}), x \sim \mathcal{A}(\cdot|\tilde{x})} \left[ \mathbb{I} \left[ h_{f,\boldsymbol{B}}(x) \neq \tilde{y} \right] \right] \\
&= 2 \mathbb{E}_{(\tilde{x},\tilde{y}) \sim \mathbb{P}(\mathcal{X},\mathcal{Y}), x \sim \mathcal{A}(\cdot|\tilde{x})} \left[ \mathbb{I} \left[ h_{f,\boldsymbol{B}}(x) \neq \tilde{y} \right] \left( \mathbb{I} \left[ h_{f,\boldsymbol{B}}(x) = \hat{y}(x) \right] + \mathbb{I} \left[ h_{f,\boldsymbol{B}}(x) \neq \hat{y}(x) \right] \right) \right] \\
&\leq 2 \mathbb{E}_{(\tilde{x},\tilde{y}) \sim \mathbb{P}(\mathcal{X},\mathcal{Y}), x \sim \mathcal{A}(\cdot|\tilde{x})} \left[ \mathbb{I} \left[ \hat{y}(x) \neq \tilde{y} \right] + \mathbb{I} \left[ h_{f,\boldsymbol{B}}(x) \neq \hat{y}(x) \right] \right] \\
&\leq 2 \left( \gamma + \mathbb{E}_{\tilde{x} \sim \mathbb{P}(\mathcal{X}), x \sim \mathcal{A}(\cdot|\tilde{x})} \left[ \mathbb{I} \left[ h_{f,\boldsymbol{B}}(x) \neq \hat{y}(x) \right] \right] \right).
\end{aligned}
\tag{64}
$$

$\square$

Using results of Lemma C.3, the problem has been transformed into analyzing $\mathbb{E}_{\tilde{x} \sim \mathbb{P}(\mathcal{X}), x \sim \mathcal{A}(\cdot|\tilde{x})} \left[ \mathbb{I} \left[ h_{f,\boldsymbol{B}}(x) \neq \hat{y}(x) \right] \right]$. We proceed with the following analysis.

**Lemma C.4.** *Given a real symmetric matrix $\boldsymbol{L}$ with size of $N$, and $\vartheta_i, i = 1, 2, \ldots d, d < N$ be the top-d smallest unit-norm eigenvector of $\boldsymbol{L}$ with eigenvalue $\lambda_i$ (make them orthogonal in case of repeated eigenvalue), then for any vector $\mu \in \mathbb{R}^N$, there exists a vector $b \in \mathbb{R}^d$, such that:*

$$\left\| \mu - \sum_{i=1}^{d} b_i \vartheta_i \right\|_2^2 \leq \frac{\mathcal{R}(\mu)}{\lambda_{d+1}} \|\mu\|_2^2, \tag{65}$$

*where $\mathcal{R}(\mu) = \frac{\mu^T \boldsymbol{L} \mu}{\mu^T \mu}$ is the Rayleigh quotient of $u$.*

*Proof.* We first decompose $\mu$ on the basis of the space spanned by the eigenvectors $\mu = \sum_{i=1}^{N} b_i \vartheta_i$, then we have:

$$\mathcal{R}(\mu) = \frac{\sum_{i=1}^{N} \lambda_i b_i^2}{\|\mu\|_2^2} \geq \frac{\lambda_{d+1} \sum_{i=d+1}^{N} b_i^2}{\|\mu\|_2^2} = \frac{\lambda_{d+1}}{\|\mu\|_2^2} \left\| \mu - \sum_{i=1}^{d} b_i \vartheta_i \right\|_2^2 \tag{66}$$

$\square$

**Lemma C.5.** *Given the pseudo-label indicator function $\boldsymbol{U}$ define as $\boldsymbol{U}_{x,i} = \sqrt{\boldsymbol{A}_x} \mathbb{I}[\hat{y}(x) = i]$ and feature $\boldsymbol{F}^*$ derived form perturbation graph $\boldsymbol{A}$ with perturbation coefficients $\alpha$ and $\beta$ satisfy $\alpha + \beta/c = 1$, then we have:*

$$\min_{\boldsymbol{B} \in \mathbb{R}^{d \times c}} \|\boldsymbol{U} - \boldsymbol{F}^* \boldsymbol{B}\|_2^2 \leq \frac{2\gamma}{\lambda_{d+1}}, \tag{67}$$

*where $\lambda_{d+1}$ is (d+1)-th smallest eigenvalue of normalized Laplacian matrix $\boldsymbol{I} - \widetilde{\boldsymbol{A}}$ of perturbation augmentation graph.*

*Proof.* According to the Eckart-Young-Mirsky theorem (Eckart and Young, 1936), $\boldsymbol{F}^*$ contains scaling of the top-$d$ smaller eigenvectors of $\boldsymbol{I} - \widetilde{\boldsymbol{A}}$. Using results of Lemma C.4, we have:

$$
\begin{aligned}
\min_{\boldsymbol{B} \in \mathbb{R}^{d \times c}} \|\boldsymbol{U} - \boldsymbol{F}^* \boldsymbol{B}\|_2^2 = \sum_{i=1}^{c} \min_{b \in \mathbb{R}^d} \|\boldsymbol{U}_{:,i} - \boldsymbol{F}^* b\| &\leq \sum_{i=1}^{c} \frac{1}{\lambda_{d+1}} \boldsymbol{U}_{:,i}^T (\boldsymbol{I} - \widetilde{\boldsymbol{A}}) \boldsymbol{U}_{:,i} \\
&= \frac{1}{\lambda_{d+1}} \sum_{i=1}^{c} \left( \sum_x \boldsymbol{A}_x \mathbb{I}[\hat{y}(x) = i] - \sum_{x,x'} \boldsymbol{A}_{x,x'} \mathbb{I}[\hat{y}(x) = i] \mathbb{I}[\hat{y}(x') = i] \right) \\
&= \frac{1}{\lambda_{d+1}} \sum_{i=1}^{c} \left( \sum_{x,x'} \boldsymbol{A}_{x,x'} \mathbb{I}[(\hat{y}(x) = i \wedge \hat{y}(x') \neq i) \vee (\hat{y}(x) \neq i \wedge \hat{y}(x') = i)] \right) \\
&= \frac{1}{\lambda_{d+1}} \sum_{x,x'} \boldsymbol{A}_{x,x'} \mathbb{I}[\hat{y}(x) \neq \hat{y}(x')]
\end{aligned}
\tag{68}
$$

Furthermore, we have:

$$
\begin{aligned}
\sum_{x,x'} \boldsymbol{A}_{x,x'} \mathbb{I}[\hat{y}(x) \neq \hat{y}(x')] &= \alpha \sum_{x,x'} w^u_{x,x'} \mathbb{I}[\hat{y}(x) \neq \hat{y}(x')] + \beta \sum_{x,x'} w^{wl}_{x,x'}(\boldsymbol{S}) \mathbb{I}[\hat{y}(x) \neq \hat{y}(x')] \\
&= \alpha \sum_{x,x'} w^u_{x,x'} \mathbb{I}[\hat{y}(x) \neq \hat{y}(x')] + \beta \sum_{x,x'} w^l_{x,x'} \mathbb{I}[\hat{y}(x) \neq \hat{y}(x')] \\
&= \alpha \mathbb{E}_{\tilde{x} \sim \mathbb{P}(\mathcal{X})} \left[ \sum_{x,x'} \mathbb{I}[\hat{y}(x) \neq \hat{y}(x')] \mathcal{A}(x \mid \tilde{x}) \mathcal{A}(x' \mid \tilde{x}) \right] \\
&+ \beta \mathbb{E}_{(\tilde{x},\tilde{y}),(\tilde{x}',\tilde{y}') \sim \mathbb{P}(\mathcal{X},\mathcal{Y})^2} \left[ \sum_{x,x'} \mathbb{I}[\hat{y}(x) \neq \hat{y}(x')] \mathbb{I}[\tilde{y} = \tilde{y}'] \mathcal{A}(x \mid \tilde{x}) \mathcal{A}(x' \mid \tilde{x}') \right] \\
&\leq \alpha \mathbb{E}_{\tilde{x},\tilde{y} \sim \mathbb{P}(\mathcal{X},\mathcal{Y})} \left[ \sum_{x,x'} (\mathbb{I}[\hat{y}(x) \neq \tilde{y}] + \mathbb{I}[\hat{y}(x') \neq \tilde{y}]) \mathcal{A}(x \mid \tilde{x}) \mathcal{A}(x' \mid \tilde{x}) \right] \\
&+ \beta \mathbb{E}_{(\tilde{x},\tilde{y}),(\tilde{x}',\tilde{y}') \sim \mathbb{P}(\mathcal{X},\mathcal{Y})^2} \left[ \sum_{x,x'} (\mathbb{I}[\hat{y}(x) \neq \tilde{y}] + \mathbb{I}[\hat{y}(x') \neq \tilde{y}']) \mathbb{I}[\tilde{y} = \tilde{y}'] \mathcal{A}(x \mid \tilde{x}) \mathcal{A}(x' \mid \tilde{x}') \right] \\
&= 2\alpha \mathbb{E}_{\tilde{x},\tilde{y} \sim \mathbb{P}(\mathcal{X},\mathcal{Y}), x \sim \mathcal{A}(\cdot|\tilde{x})} \left[ \mathbb{I}[\tilde{y} = \hat{y}(x)] \right] \\
&+ 2\beta \mathbb{E}_{\tilde{x},\tilde{y} \sim \mathbb{P}(\mathcal{X},\mathcal{Y}), x \sim \mathcal{A}(\cdot|\tilde{x}), \tilde{y}' \sim \mathbb{P}(\mathcal{Y})} \left[ \mathbb{I}[\tilde{y} = \hat{y}(x)] \mathbb{I}[\tilde{y} = \tilde{y}'] \right] \\
&\leq 2(\alpha + \beta/c)\gamma = 2\gamma,
\end{aligned}
\tag{69}
$$

here the first inequality holds because:

$$
\mathbb{I}[\hat{y}(x) \neq \hat{y}(x')] \leq \mathbb{I}[\hat{y}(x) \neq \tilde{y}] + \mathbb{I}[\hat{y}(x') \neq \tilde{y}].
\tag{70}
$$

$$
\mathbb{I}[\hat{y}(x) \neq \hat{y}(x')] \mathbb{I}[\tilde{y} = \tilde{y}'] \leq (\mathbb{I}[\hat{y}(x) \neq \tilde{y}] + \mathbb{I}[\hat{y}(x') \neq \tilde{y}']) \mathbb{I}[\tilde{y} = \tilde{y}'].
\tag{71}
$$

Finally, we substitute Formula 69 into Formula 68 to complete the proof. □

**Lemma C.6.** *Given the pseudo-label indicator function $\boldsymbol{V}$ define as $\boldsymbol{V}_{x,i} = \mathbb{I}[\hat{y}(x) = i]$ and feature $f^*(x) = \frac{1}{\sqrt{\boldsymbol{A}_x}} \left( \boldsymbol{F}^*_{x,:} \right)^T$ then we have:*

$$
\min_{\boldsymbol{B} \in \mathbb{R}^{d \times c}} \mathbb{E}_{\tilde{x} \sim \mathbb{P}(\mathcal{X}), x \sim \mathcal{A}(\cdot|\tilde{x})} \left[ \mathbb{I}[h_{f^*,\boldsymbol{B}}(x) \neq \hat{y}(x)] \right] \leq 2 \min_{\boldsymbol{B} \in \mathbb{R}^{d \times c}} \mathbb{E}_{\tilde{x} \sim \mathbb{P}(\mathcal{X}), x \sim \mathcal{A}(\cdot|\tilde{x})} \left[ \left\| \boldsymbol{V}^T_{x,:} - \boldsymbol{B}^T f^*(x) \right\|^2_2 \right] \leq \frac{4\gamma}{\lambda_{d+1}}.
\tag{72}
$$

*Proof.* Firstly, we prove the first inequality in the Formula 72. For any $x$, $h_{f^*,\boldsymbol{B}}(x) \neq \hat{y}(x)$ there exists $y'$ such that $(\boldsymbol{B}^T f^*(x))_{y'} > (\boldsymbol{B}^T f^*(x))_{\hat{y}(x)}$, hence, we have:

$$
\begin{aligned}
\left\| \boldsymbol{V}^T_{x,:} - \boldsymbol{B}^T f^*(x) \right\|^2_2 &\geq (1 - (\boldsymbol{B}^T f^*(x))_{\hat{y}(x)})^2 + ((\boldsymbol{B}^T f^*(x))_{y'})^2 \\
&\geq \frac{1}{2}(1 - (\boldsymbol{B}^T f^*(x))_{\hat{y}(x)} + (\boldsymbol{B}^T f^*(x))_{y'})^2 \geq \frac{1}{2}.
\end{aligned}
\tag{73}
$$

Next, we can expand $\mathbb{E}_{\tilde{x} \sim \mathbb{P}(\mathcal{X}), x \sim \mathcal{A}(\cdot|\tilde{x})} \left[ \left\| \boldsymbol{V}^T_{x,:} - \boldsymbol{B}^T f^*(x) \right\|^2_2 \right]$ and obtain:

$$
\mathbb{E}_{\tilde{x} \sim \mathbb{P}(\mathcal{X}), x \sim \mathcal{A}(\cdot|\tilde{x})} \left[ \left\| \boldsymbol{V}^T_{x,:} - \boldsymbol{B}^T f^*(x) \right\|^2_2 \right] = \sum_x \mathbb{P}(x) \left\| \boldsymbol{V}^T_{x,:} - \boldsymbol{B}^T f^*(x) \right\|^2_2
\tag{74}
$$

We claim $\mathbb{P}(x) = \boldsymbol{A}_x$ when $\alpha + \beta/c = 1$:

$$
\begin{aligned}
\boldsymbol{A}_x &= \alpha \sum_{x'} w^u_{x,x'} + \beta \sum_{x'} w^l_{x,x'} \\
&= \alpha \mathbb{E}_{\tilde{x} \sim \mathbb{P}(\mathcal{X})}[\mathcal{A}(x \mid \tilde{x}) \sum_{x'} \mathcal{A}(x' \mid \tilde{x})] + \beta \mathbb{E}_{(\tilde{x},\tilde{y}),(\tilde{x}',\tilde{y}') \sim \mathbb{P}(\mathcal{X},\mathcal{Y})^2} \left[ \mathbb{I}\left[\tilde{y} = \tilde{y}'\right] \mathcal{A}(x \mid \tilde{x}) \sum_{x'} \mathcal{A}(x' \mid \tilde{x}') \right] \\
&= \alpha \mathbb{E}_{\tilde{x} \sim \mathbb{P}(\mathcal{X})}[\mathcal{A}(x \mid \tilde{x})] + \beta \mathbb{E}_{(\tilde{x},\tilde{y}) \sim \mathbb{P}(\mathcal{X},\mathcal{Y}),\tilde{y}' \sim \mathbb{P}(\mathcal{Y})} \left[ \mathbb{I}\left[\tilde{y} = \tilde{y}'\right] \mathcal{A}(x \mid \tilde{x}) \right] \\
&= \alpha \mathbb{P}(x) + \frac{\beta}{c} \mathbb{P}(x) = \mathbb{P}(x)
\end{aligned}
\tag{75}
$$

By substituting Equation 75 into Equation 74, we have:

$$
\begin{aligned}
\mathbb{E}_{\tilde{x} \sim \mathbb{P}(\mathcal{X}), x \sim \mathcal{A}(\cdot \mid \tilde{x})} \left[ \left\| \boldsymbol{V}^T_{x,:} - \boldsymbol{B}^T f^*(x) \right\|^2_2 \right] &= \sum_x \boldsymbol{A}_x \left\| \boldsymbol{V}^T_{x,:} - \boldsymbol{B}^T f^*(x) \right\|^2_2 \\
&= \sum_x \left\| \boldsymbol{U}^T_{x,:} - \boldsymbol{B}^T (F^*_{x,:})^T \right\|^2_2 \\
&= \left\| \boldsymbol{U} - \boldsymbol{F}^* \boldsymbol{B} \right\|^2_2.
\end{aligned}
\tag{76}
$$

Using the results of Lemma C.5, we finish the proof of the second inequality in the Formula 72. $\qquad \square$

Combining Lemmas C.2, C.3, C.6 , we have prove under condition in Theorem $C.1$:

$$
\varepsilon(f^*) \leq 2 \left( \frac{4}{\lambda_{d+1}} + 1 \right) \gamma,
\tag{77}
$$

where $\lambda_{d+1}$ is $d + 1$-th smallest eigenvalue of normalized Laplacian matrix of perturbation graph.

The following lemma establishes the relationship between this eigenvalue and graph connectivity, allowing us to get rid of the dependency on $\lambda_{d+1}$.

**Lemma C.7.** *(Higher-order Cheeger's inequality, Proposition. 1.2 in Louis and Makarychev 2014) Let $\mathcal{G} = (V, E)$ be a weight graph with $|V| = N$. Then for any $t \in [N]$ and $\zeta > 0$ such that $(1 + \zeta)t \in [N]$, there exist a partition $\Omega_1, \ldots, \Omega_t$ of $V$ with:*

$$
\Phi_{\mathcal{G}}(\Omega_i) \preceq \text{poly}\left( \frac{1}{\zeta} \right) \sqrt{\lambda_{(1+\zeta)t} \log t}, \forall i \in [t].
\tag{78}
$$

In Lemma C.7, we take $\zeta = 1$ and $(1 + \zeta)t = 2t = d + 1$, there exist a partition $\Omega_i, \ldots, \Omega_{\lfloor d/2 \rfloor}$ such that $\max_{i \in [\lfloor d/2 \rfloor]} \Phi_{\mathcal{G}}(\Omega_i) \preceq \sqrt{\lambda_{d+1} \log(d+1)}$. We take $\rho$ as sparest partition of perturbation graph. By Definition 3.3, we have $\rho_{\lfloor d/2 \rfloor} \leq \max_{i \in [\lfloor d/2 \rfloor]} \Phi_{\mathcal{G}}(\Omega_i) \preceq \sqrt{\lambda_{d+1} \log(d+1)}$ which leads to there exist a constant $\eta(d)$ such that $\frac{1}{\lambda_{d+1}} \leq \eta(d) \frac{1}{\rho^2_{\lfloor d/2 \rfloor}}$. The following lemma bounds the sparest partition of perturbation graph.

**Lemma C.8.** *Given perturbation graph with $\boldsymbol{A}_{x,x'} = w_{x,x'} = \alpha w^u_{x,x'} + \beta w^{wl}_{x,x'}(\boldsymbol{S})$ where $\alpha + \beta/c = 1$, and self-connectivity graph $\boldsymbol{A}^u_{x,x'}$ with sparest partition $\rho^u$, then we have lower bounds of sparest partition $\rho$ of graph $\boldsymbol{A}$ as follows:*

$$
\rho_d \geq \alpha \rho^u_d + \frac{\beta}{c}\left(1 - \frac{c}{d}\right).
\tag{79}
$$

*Proof.* Firstly, for any subset $\Omega$ of augmentation data points, we have:

$$
\begin{aligned}
\Phi_{\boldsymbol{A}}(\Omega) &= \frac{\sum_{x\in\Omega,x'\notin\Omega} w_{x,x'}}{\sum_{x\in\Omega} w_x} = \frac{\sum_{x\in\Omega,x'\notin\Omega} \alpha w_{x,x'}^u + \beta w_{x,x'}^{wl}(\boldsymbol{S})}{\sum_{x\in\Omega} w_x} \\
&= \frac{\sum_{x\in\Omega,x'\notin\Omega} \alpha w_{x,x'}^u + \beta w_{x,x'}^l}{\sum_{x\in\Omega} w_x} = \frac{\sum_{x\in\Omega,x'\notin\Omega} \alpha w_{x,x'}^u + \beta w_{x,x'}^l}{\mathbb{P}(\Omega)} \\
&= \alpha\frac{\sum_{x\in\Omega,x'\notin\Omega} w_{x,x'}^u}{\mathbb{P}(\Omega)} + \beta\frac{\sum_{x\in\Omega,x'\notin\Omega} w_{x,x'}^l}{\mathbb{P}(\Omega)} \\
&= \alpha\frac{\sum_{x\in\Omega,x'\notin\Omega} w_{x,x'}^u}{\sum_{x\in\Omega} w_x^u} + \frac{\beta}{c}\frac{\sum_{x\in\Omega,x'\notin\Omega} w_{x,x'}^l}{\sum_{x\in\Omega} w_x^l}. \\
&= \alpha\Phi_{\boldsymbol{A}^u}(\Omega) + \frac{\beta}{c}\Phi_{\boldsymbol{A}^l}(\Omega),
\end{aligned}
\tag{80}
$$

here the 4-th and 6-th equality derived from Equation 75.

Hence we have lower bounds of $\rho_d$ as follows:

$$
\begin{aligned}
\rho_d &= \min_{\Omega_1,\ldots,\Omega_d} \max\{\Phi_{\boldsymbol{A}}(\Omega_1),\ldots,\Phi_{\boldsymbol{A}}(\Omega_d)\} \\
&= \min_{\Omega_1,\ldots,\Omega_d} \max\{\alpha\Phi_{\boldsymbol{A}^u}(\Omega_1) + \frac{\beta}{c}\Phi_{\boldsymbol{A}^l}(\Omega_1),\ldots,\alpha\Phi_{\boldsymbol{A}^u}(\Omega_d) + \frac{\beta}{c}\Phi_{\boldsymbol{A}^l}(\Omega_d)\} \\
&\geq \alpha\min_{\Omega_1,\ldots,\Omega_d} \max\{\Phi_{\boldsymbol{A}^u}(\Omega_1),\ldots,\Phi_{\boldsymbol{A}^u}(\Omega_d)\} + \frac{\beta}{c}\min_{\Omega_1,\ldots,\Omega_d} \max\{\Phi_{\boldsymbol{A}^l}(\Omega_1),\ldots,\Phi_{\boldsymbol{A}^l}(\Omega_d)\} \\
&= \alpha\rho_d^u + \frac{\beta}{c}\rho_d^l,
\end{aligned}
\tag{81}
$$

here the inequality holds because taking the minimum separately is always less than taking the minimum simultaneously.

In the following text, we prove that $\rho_d^l \geq (1 - \frac{c}{d})$ to finish the proof of Lemma C.8. For any subset $\Omega$, we can compute $\Phi_{\boldsymbol{A}^l}(\Omega)$ as follows:

$$
\begin{aligned}
\Phi_{\boldsymbol{A}^l}(\Omega) &= \frac{\sum_{x\in\Omega,x'\notin\Omega} w_{x,x'}^l}{\sum_{x\in\Omega} w_x} = \frac{\sum_{x\in\Omega,x'\notin\Omega} w_{x,x'}^l}{\mathbb{P}(\Omega)/c} \\
&= \frac{\mathbb{E}_{(\tilde{x},\tilde{y}),(\tilde{x}',\tilde{y}')\sim\mathbb{P}(\mathcal{X},\mathcal{Y})^2}\left[\sum_{x\in\Omega,x'\notin\Omega} \mathbb{I}\left[\tilde{y}=\tilde{y}'\right]\mathcal{A}(x\mid\tilde{x})\mathcal{A}(x'\mid\tilde{x}')\right]}{\sum_{i=1}^c \frac{1}{c^2}\mathbb{P}(\Omega\mid\tilde{y}=c)} \\
&= \frac{\mathbb{E}_{(\tilde{x},\tilde{y}),(\tilde{x}',\tilde{y}')\sim\mathbb{P}(\mathcal{X},\mathcal{Y})^2}\left[\sum_{x\in\Omega,x'\notin\Omega} \mathbb{I}\left[\tilde{y}=\tilde{y}'\right]\mathcal{A}(x\mid\tilde{x})\mathcal{A}(x'\mid\tilde{x}')\right]}{\sum_{i=1}^c \frac{1}{c^2}\mathbb{P}(\Omega\mid\tilde{y}=c)} \\
&= \frac{\frac{1}{c^2}\sum_{i=1}^c \mathbb{E}_{(\tilde{x},\tilde{x}')\sim\mathbb{P}(\cdot\mid\tilde{y}=i)^2}\left[\sum_{x\in\Omega,x'\notin\Omega} \mathcal{A}(x\mid\tilde{x})\mathcal{A}(x'\mid\tilde{x}')\right]}{\sum_{i=1}^c \frac{1}{c^2}\mathbb{P}(\Omega\mid\tilde{y}=c)} \\
&= \frac{\sum_{i=1}^c \mathbb{E}_{(\tilde{x},\tilde{x}')\sim\mathbb{P}(\cdot\mid y=i)^2}\left[\sum_{x\in\Omega,x'\notin\Omega} \mathcal{A}(x\mid\tilde{x})\mathcal{A}(x'\mid\tilde{x}')\right]}{\sum_{i=1}^c \mathbb{P}(\Omega\mid\tilde{y}=i)} \\
&= \frac{\sum_{i=1}^c \mathbb{P}(\Omega\mid\tilde{y}=i)(1-\mathbb{P}(\Omega\mid\tilde{y}=i))}{\sum_{i=1}^c \mathbb{P}(\Omega\mid\tilde{y}=i)}.
\end{aligned}
\tag{82}
$$

For any partition $\Omega_1,\ldots,\Omega_d$ of augmentation graph, we claim that they must satisfy conditions as follows:

$$
\begin{aligned}
\sum_{i=1}^d \mathbb{P}(\Omega_i\mid\tilde{y}=j) &= 1, \quad \forall j\in[c] \\
\sum_{j=1}^c \mathbb{P}(\Omega_i\mid\tilde{y}=j) &> 0, \quad \forall i\in[d].
\end{aligned}
$$

Hence, when take $\mathbb{P}(\Omega_i \mid \tilde{y} = j) = \omega_{i,j}$, we have:

$$\rho_d^l \geq \min_w \Psi(\omega) = \min_\omega \max_{i \in [d]} \frac{\sum_{j=1}^c \omega_{i,j}(1 - \omega_{i,j})}{\sum_{j=1}^c \omega_{i,j}}$$

$$\text{s.t.} \quad \sum_{i=1}^d \omega_{i,j} = 1, \quad \forall j \in [c]$$

$$\sum_{j=1}^c \omega_{i,j} > 0, \quad \forall i \in [d] \tag{83}$$

$$0 \leq \omega_{i,j} \leq 1, \quad \forall i \in [c], j \in [d].$$

We claim that the $\omega^*$ that minimizes the constrained minimization problem defined by Formula 83 should satisfy the condition: there exists $i^*, j^* = \arg\min_{i,j:\omega_{i,j}^* > 0} \omega_{i,j}^*$ such that $\omega_{i^*,j}^* = 0, \forall j \in [c] \setminus j^*$ and $\Psi(\omega^*) = 1 - \omega_{i^*,j^*}^*$.

To view that, one the one hand, for any $\omega$ break this condition, we can construct a $w'$ such that $\Psi(\omega') < \Psi(\omega)$ as follows:

$$\omega_{i^*,j^*}' = 0, \; \omega_{1,j^*}' = \omega_{1,j^*} + \omega_{i^*,j^*}$$

$$\omega_{i,j}' = \omega_{i,j}, \quad otherwise, \tag{84}$$

here we assume $\omega_{1,j^*} > 0$. In order to verify $\Psi(\omega') < \Psi(\omega)$, we first notice that:

$$\frac{\sum_{j=1}^c \omega_{i^*,j}'(1 - \omega_{i^*,j}')}{\sum_{j=1}^c \omega_{i^*,j}'} = \frac{\sum_{j \neq j^*} \omega_{i^*,j}(1 - \omega_{i^*,j})}{\sum_{j \neq j^*} \omega_{i^*,j}} \leq \frac{\sum_{j=1}^c \omega_{i^*,j}(1 - \omega_{i^*,j})}{\sum_{j=1}^c \omega_{i^*,j}}, \tag{85}$$

here the inequality holds because for any $j \neq j^*$, we have $1 - \omega_{i^*,j} \leq 1 - \omega_{i^*,j^*}$.

Furthermore, we also notice that:

$$\frac{\sum_{j=1}^c \omega_{1,j}'(1 - \omega_{1,j}')}{\sum_{j=1}^c \omega_{1,j}'} = \frac{\sum_{j \neq j^*} \omega_{1,j}(1 - \omega_{1,j}) + (\omega_{1,j^*} + \omega_{i^*,j^*})(1 - \omega_{1,j^*} - \omega_{i^*,j^*})}{\sum_{j \neq j^*} \omega_{1,j} + \omega_{1,j^*} + \omega_{i^*,j^*}}$$

$$\leq \frac{\sum_{j \neq j^*} \omega_{1,j}(1 - \omega_{1,j}) + \omega_{1,j^*}(1 - \omega_{1,j^*})}{\sum_{j \neq j^*} \omega_{1,j} + \omega_{1,j^*}} = \frac{\sum_{j=1}^c \omega_{1,j}(1 - \omega_{1,j})}{\sum_{j \neq j^*} \omega_{1,j}}. \tag{86}$$

Combining Formulas 85 and 86, we have $\Psi(\omega') < \Psi(\omega)$.

On the other hand, when the condition is met, for any $i \neq i^*$, we have:

$$\frac{\sum_{j=1}^c \omega_{i,j}(1 - \omega_{i,j})}{\sum_{j=1}^c \omega_{i,j}} \leq 1 - w_{i^*,j^*}, \tag{87}$$

here the inequality holds because for any $j$ with $\omega_{i,j} \neq 0$, we have $1 - w_{i,j} < 1 - w_{i^*,j^*}$.

Utilizing the above results, we can rewrite the constrained optimization problem as follows:

$$\min_\omega 1 - \omega_{i^*,j^*}$$

$$\text{s.t.} \quad \sum_{i=1}^d \omega_{i,j} = 1, \quad \forall j \in [c]$$

$$\sum_{j=1}^c \omega_{i,j} > 0, \quad \forall i \in [d] \tag{88}$$

$$0 \leq \omega_{i,j} \leq 1, \quad \forall i \in [c], j \in [d]$$

$$\omega_{i^*,j} = 0, \forall j \in [c] \setminus j^*,$$

where $i^*, j^* = \arg\min_{i,j} \omega_{i,j}$. Furthermore, for any $\omega$ that satisfies the constraint of the above Formula 88, we have $c = \sum_{i,j} \omega_{i,j} \geq d\omega_{i^*,j^*}$ through there are at least $d$ non-zero elements in $\omega_{i,:}$ for any $i \in [d]$. Substituting this result into Equation 83, We have proved that $\rho_d^l \geq 1 - \frac{c}{d}$, thereby completing the proof. □

Finally, we combine the results of Lemma C.8 and Equation 77 to complete the final proof of Theorem C.1.

*Proof of Theorem C.1.* By substituting Formula 79 into Formula 77 using Lemma C.7, we obtain the following bound, thereby completing the proof of Theorem C.1.

$$
\begin{aligned}
\varepsilon(f^*) &\leq 2\left(\frac{4}{\lambda_{d+1}} + 1\right)\gamma \leq \eta(d)\frac{\gamma}{\rho_{\lfloor d/2 \rfloor}^2} \\
&\leq \eta(d)\left(\gamma\left(\alpha\rho_{\lfloor d/2 \rfloor}^u + \frac{\beta}{c}(1 - \frac{2c}{d})\right)^{-2}\right)
\end{aligned}
\tag{89}
$$

$\square$

## C.2. Proof of Theorem 3.7

**Theorem C.9.** *(Recap of Theorem 3.7) Given $\mathcal{F}$, assuming $\|\mathcal{F}\|_\infty = \kappa < +\infty$, for a random training dataset $D = D_\mathcal{Q} \cup D_\mathcal{U}$ with $n_q$ and $n_u$ samples respectively. Then with probability at least $1 - \delta$, we have:*

$$
\begin{aligned}
\mathcal{L}_{wsc}(\widehat{f}(D, \widehat{S})) - \mathcal{L}_{wsc}(f^*) &\leq \left(\alpha + (\alpha + \beta/c)^2\right)\left(\eta_0\widehat{\mathcal{R}}_{\frac{n_q+n_u}{2}}(\mathcal{F}) + \eta_1\left(\sqrt{\frac{\log 2/\delta}{n_q + n_u}} + \frac{\delta}{2}\right)\right) \\
&\quad + \beta \sup_{x \in \mathcal{X}}\left\|\widehat{S}(x)^T\widehat{S}(x)\right\|_\infty\left(\eta_2\widehat{\mathcal{R}}_{\frac{n_q}{2}}(\mathcal{F}) + \eta_3\left(\sqrt{\frac{\log 2/\delta}{n_q}} + \frac{\delta}{2}\right)\right) + \beta\eta_4\Delta(\widehat{S}),
\end{aligned}
\tag{90}
$$

*where $\eta_0 \preceq \kappa d + \kappa^2 d^2$; $\eta_1 \preceq \kappa^2 d + \kappa^4 d^2$; $\eta_2 \preceq \kappa d$; $\eta_3, \eta_4 \preceq \kappa^2 d$ are constants related to $\|\mathcal{F}\|_\infty$ and feature dimension $d$.*

**Definition C.10.** *(Empirical weakly-supervised spectral contrastive loss under uniform class distribution) Given constructed $\widehat{S} : \mathcal{X} \to \mathbb{R}^{c \times n}$ and considering training dataset with corresponding weakly-supervised information $D_\mathcal{Q} = \{(x_i, q_i)_{i=1}^{n_q}\}$ and training dataset without any supervised information $D_\mathcal{U} = \{(x_i)_{i=n_q+1}^{n_q+n_u}\}$, we define empirical weakly-supervised spectral contrastive under uniform class distribution as follows:*

$$
\begin{aligned}
\widehat{\mathcal{L}}_{wsc}(f; D, \widehat{S}) &= \mathbb{E}_{X_Q^1, X_Q^2 \sim f(\mathcal{A}(\cdot|D_\mathcal{Q})^2)), X_U^1, X_U^2 \sim f(\mathcal{A}(\cdot|D)^2}\left[\widehat{\mathcal{L}}_{wsc}(X_Q^1, X_Q^2, X_U^1, X_U^2, \widehat{S}, Q)\right] \\
&= \mathbb{E}_{x_i^1, x_i^2 \sim \mathcal{A}(\cdot|x_i)^2}\left[-2\alpha\frac{1}{n_u + n_q}\sum_{i=1}^{n_u+n_q} f(x_i^1)f(x_i^2)^T + (\alpha + \beta/c)^2\frac{1}{(n_u+n_q)^2}\sum_{i=1}^{n_u+n_q}\sum_{j=1}^{n_u+n_q}\left(f(x_i^1)f(x_i^2)^T\right)^2\right] \\
&\quad + \mathbb{E}_{x_i^1, x_i^2}\left[-2\beta\frac{1}{n_q^2}\sum_{i=1}^{n_q}\sum_{j=1}^{n_q}\widehat{S}(x_i)_{:,q_i}^T\widehat{S}(x_i)_{:,q_j}f(x_i^1)f(x_i^2)^T\right]
\end{aligned}
\tag{91}
$$

Giving $\widehat{f}(D, \widehat{S}) = \arg\min \widehat{\mathcal{L}}_{wsc}(f; D, \widehat{S})$ in Definition C.10. The follow lemma decompose the error bound of $\mathcal{L}_{wsc}(\widehat{f}(D, \widehat{S})) - \mathcal{L}_{wsc}(f^*)$.

**Lemma C.11.** *We define $\widehat{\mathcal{L}}_{wsc}(f; \widehat{S})$ as expected weakly-supervised spectral contrastive loss using constructed $\widehat{S}$, then we have:*

$$
\mathcal{L}_{wsc}(\widehat{f}(D, \widehat{S})) - \mathcal{L}_{wsc}(f^*) \leq 2\sup_{f \in \mathcal{F}}\left(\widehat{\mathcal{L}}_{wsc}(f; D, \widehat{S}) - \widehat{\mathcal{L}}_{wsc}(f; \widehat{S})\right) + 2\sup_{f \in \mathcal{F}}\left|\widehat{\mathcal{L}}_{wsc}(f; \widehat{S}) - \widehat{\mathcal{L}}_{wsc}(f)\right|
\tag{92}
$$

*Proof.* On the one hand, we have:

$$
\begin{aligned}
\left(\mathcal{L}_{wsc}(\widehat{f}(D, \widehat{S})) - \mathcal{L}_{wsc}(f^*)\right) &- \left(\widehat{\mathcal{L}}_{wsc}(\widehat{f}(D, \widehat{S}); \widehat{S}) - \widehat{\mathcal{L}}_{wsc}(f^*; \widehat{S})\right) \\
&\leq \left|\mathcal{L}_{wsc}(\widehat{f}(D, \widehat{S})) - \widehat{\mathcal{L}}_{wsc}(\widehat{f}(D, \widehat{S}); \widehat{S})\right| + \left|\mathcal{L}_{wsc}(f^*) - \widehat{\mathcal{L}}_{wsc}(f^*; \widehat{S})\right| \\
&\leq 2\sup_{f \in \mathcal{F}}\left|\widehat{\mathcal{L}}_{wsc}(f; \widehat{S}) - \mathcal{L}_{wsc}(f)\right|.
\end{aligned}
\tag{93}
$$

On the other hand, we have:

$$
\begin{aligned}
\widehat{\mathcal{L}}_{wsc}(\widehat{f}(D,\widehat{S});\widehat{S}) - \widehat{\mathcal{L}}_{wsc}(f^*;\widehat{S}) = {}& \widehat{\mathcal{L}}_{wsc}(\widehat{f}(D,\widehat{S});\widehat{S}) - \widehat{\mathcal{L}}_{wsc}(\widehat{f}(D,\widehat{S});D,\widehat{S}) \\
& + \widehat{\mathcal{L}}_{wsc}(\widehat{f}(D,\widehat{S});D,\widehat{S}) - \widehat{\mathcal{L}}_{wsc}(f^*;D,\widehat{S}) + \widehat{\mathcal{L}}_{wsc}(f^*;D,\widehat{S}) - \widehat{\mathcal{L}}_{wsc}(f^*;\widehat{S}) \\
& \leq 2 \sup_{f \in \mathcal{F}} \left( \widehat{\mathcal{L}}_{wsc}(f;D,\widehat{S}) - \widehat{\mathcal{L}}_{wsc}(f;\widehat{S}) \right).
\end{aligned}
\tag{94}
$$

Combining Formulas 93 and 94 completes the proof. $\qquad\square$

We first bounds the first term in Formula 92. It is direct to see that $\widehat{\mathcal{L}}_{wsc}(f;D,\widehat{S})$ is an unbiased estimator of $\widehat{\mathcal{L}}_{wsc}(f;\widehat{S})$, to make use of the Rademacher complexity theory to given a generalize bound of this term, we convert the non sum of i.i.d pairwise function to a sum of i.i.d form by using perturbations in U-process (Clémençon et al., 2008).

**Definition C.12.** (Sub-sampling for empirical weakly-supervised spectral contrastive loss) We sample tuples to calculate the empirical weakly-supervised spectral contrastive loss as follows:

$$
x_i^1, x_i^2 \sim \mathcal{A}(\cdot \mid x_i)^2, \quad i = 1, \ldots, n_q + n_u
\tag{95}
$$

$$
\begin{aligned}
\widehat{\mathcal{L}}_{wsc}(f;D,\widehat{S}) = {}& -2\alpha \frac{2}{n} \sum_{i=1}^{n/2} f(x_i^1) f(x_i^2)^T + (\alpha + \beta/c)^2 \frac{2}{n} \sum_{i=1}^{n/2} \left( f(x_i^1) f(x_{i+(n/2)}^2)^T \right)^2 \\
& -2\beta \frac{2}{n_q} \sum_{i=1}^{n_q/2} \widehat{S}(x_i)_{:,q_i}^T \widehat{S}(x_{i+n_q/2})_{:,q_{i+n_q/2}} f(x_i^1) f(x_{i+n_q/2}^2)^T \\
= {}& \widehat{\mathcal{L}}_{wsc}^u(f;D,\widehat{S}) + \widehat{\mathcal{L}}_{wsc}^l(f;D,\widehat{S})
\end{aligned}
\tag{96}
$$

The following lemma reveals the relationship between the Rademacher complexity of feature extractors and the Rademacher complexity of the loss defined on U-process sub-sampling.

**Lemma C.13.** *Let $\mathcal{F}$ be a hypothesis class of feature extractors from $\mathcal{X}$ to $\mathbb{R}$. Assume $\|\mathcal{F}\|_\infty = \kappa$ and let $\widehat{\mathcal{R}}_n(\mathcal{F})$ be maximal possible empirical Rademacher complexity for feature extractor in Definition 3.5, then we can bounds empirical Rademacher complexity on sub-sampling samples as follows:*

$$
\mathbb{E}_\sigma \left[ \sup_{f \in \mathcal{F}} \frac{1}{n/2} \sum_{i=1}^{n/2} \sigma_i f(x_i^1) f(x_i^2)^T \right] \leq 8d\kappa \widehat{\mathcal{R}}_{n/2}(\mathcal{F})
\tag{97}
$$

$$
\mathbb{E}_\sigma \left[ \sup_{f \in \mathcal{F}} \frac{1}{n/2} \sum_{i=1}^{n/2} \sigma_i \left( f(x_i^1) f(x_{i+n/2}^2)^T \right)^2 \right] \leq 16d^2\kappa^2 \widehat{\mathcal{R}}_{n/2}(\mathcal{F})
\tag{98}
$$

$$
\mathbb{E}_\sigma \left[ \sup_{f \in \mathcal{F}} \frac{1}{n_q/2} \sum_{i=1}^{n_q/2} \sigma_i \widehat{S}(x_i)_{:,q_i}^T \widehat{S}(x_{i+n_q/2})_{:,q_{i+n_q/2}} f(x_i^1) f(x_{i+n_q/2}^2)^T \right] \leq \sup_{x \in \mathcal{X}} \left\| \widehat{S}(x)^T \widehat{S}(x) \right\|_\infty 8d\kappa \widehat{\mathcal{R}}_{n_q/2}(\mathcal{F})
\tag{99}
$$

*Proof.* We first proof Formula 97 as follows:

$$
\begin{aligned}
\mathbb{E}_\sigma \left[ \sup_{f \in \mathcal{F}} \frac{1}{n/2} \sum_{i=1}^{n/2} \sigma_i f(x_i^1) f(x_i^2)^T \right] = {}& \mathbb{E}_\sigma \left[ \sup_{f \in \mathcal{F}} \frac{1}{n/2} \sum_{i=1}^{n/2} \sigma_i \sum_{j=1}^{d} f_j(x_i^1) f_j(x_i^2) \right] \\
& \leq d \max_{x_i^1, x_i^2} \max_{j \in [d]} \mathbb{E}_\sigma \left[ \sup_{f_j \in \mathcal{F}_j} \frac{1}{n/2} \sum_{i=1}^{n/2} \sigma_i f_j(x_i^1) f_j(x_i^2) \right].
\end{aligned}
\tag{100}
$$

For any $x_i^1, x_i^2$ and $j$, we use Talagrand's lemma to bound the end term of Formula 100:

$$
\mathbb{E}_\sigma \left[ \sup_{f_j \in \mathcal{F}_j} \frac{1}{n/2} \sum_{i=1}^{n/2} \sigma_i f_j(x_i^1) f_j(x_i^2) \right]
$$

$$
= \mathbb{E}_\sigma \left[ \sup_{f_j \in \mathcal{F}_j} \frac{1}{n/2} \sum_{i=1}^{n/2} \sigma_i \left( \frac{1}{2} \left( f_j(x_i^1) + f_j(x_i^2) \right)^2 - \frac{1}{2} \left( f_j(x_i^1) - f_j(x_i^2) \right)^2 \right) \right]
$$

$$
\leq \frac{1}{2} \mathbb{E}_\sigma \left[ \sup_{f_j \in \mathcal{F}_j} \frac{1}{n/2} \sum_{i=1}^{n/2} \sigma_i \left( f_j(x_i^1) + f_j(x_i^2) \right)^2 \right] + \frac{1}{2} \mathbb{E}_\sigma \left[ \sup_{f_j \in \mathcal{F}_j} \frac{1}{n/2} \sum_{i=1}^{n/2} \sigma_i \left( f_j(x_i^1) - f_j(x_i^2) \right)^2 \right]
$$

$$
\leq 4\kappa \left( \mathbb{E}_\sigma \left[ \sup_{f_j \in \mathcal{F}_j} \frac{1}{n/2} \sum_{i=1}^{n/2} \sigma_i f_j(x_i^1) \right] + \mathbb{E}_\sigma \left[ \sup_{f_j \in \mathcal{F}_j} \frac{1}{n/2} \sum_{i=1}^{n/2} \sigma_i f_j(x_i^2) \right] \right). \tag{101}
$$

Combining results of Formulas 100 and 101, we can get Formula 97.

Next, we also use Talagrand's lemma to bound LHS of Formula 98 as follows:

$$
\mathbb{E}_\sigma \left[ \sup_{f \in \mathcal{F}} \frac{1}{n/2} \sum_{i=1}^{n/2} \sigma_i \left( f(x_i^1) f(x_{i+n/2}^2)^T \right)^2 \right] \leq 2d\kappa \mathbb{E}_\sigma \left[ \sup_{f \in \mathcal{F}} \frac{1}{n/2} \sum_{i=1}^{n/2} \sigma_i f(x_i^1) f(x_{i+n/2}^2)^T \right]
$$

$$
\leq 2d^2 \kappa \max_{x_i^1, x_{i+n/2}^2} \max_{j \in [d]} \mathbb{E}_\sigma \left[ \sigma \frac{1}{n/2} \sum_{i=1}^{n/2} \sigma_i f_j(x_i^1) f_j(x_{i+n/2}^2) \right] \tag{102}
$$

$$
\leq 16 d^2 k^2 \widehat{\mathcal{R}}_{n/2}(\mathcal{F}),
$$

where the first inequality follows from Talagrand's lemma, and the second and third inequalities result from Formulas 100 and 101, respectively.

In the end, we consider $g_{\widehat{\boldsymbol{S}}, q}(f(x_i^1) f(x_{i+n_q/2}^2)) = \widehat{\boldsymbol{S}}(x_i)_{:,q_i}^T \widehat{\boldsymbol{S}}(x_i)_{:,q_j} f(x_i^1) f(x_{i+n_q/2}^2)^T$ is $\sup_{x \in \mathcal{X}} \left\| \widehat{\boldsymbol{S}}(x)^T \widehat{\boldsymbol{S}}(x) \right\|_\infty$ - Lipchitz continuous, hence we can use Talagrand's lemma to bound LHS of Formula 99:

$$
\mathbb{E}_\sigma \left[ \sup_{f \in \mathcal{F}} \frac{1}{n_q/2} \sum_{i=1}^{n_q/2} \sigma_i \widehat{\boldsymbol{S}}(x_i)_{:,q_i}^T \widehat{\boldsymbol{S}}(x_{i+n_q/2})_{:,q_{i+n_q/2}} f(x_i^1) f(x_{i+n_q/2}^2)^T \right]
$$

$$
\leq \sup_{x \in \mathcal{X}} \left\| \widehat{\boldsymbol{S}}(x)^T \widehat{\boldsymbol{S}}(x) \right\|_\infty \mathbb{E}_\sigma \left[ \sup_{f \in \mathcal{F}} \frac{1}{n_q/2} \sum_{i=1}^{n_q/2} \sigma_i f(x_i^1) f(x_{i+n_q/2}^2)^T \right] \tag{103}
$$

$$
\leq 8d\kappa \sup_{x \in \mathcal{X}} \left\| \widehat{\boldsymbol{S}}(x)^T \widehat{\boldsymbol{S}}(x) \right\|_\infty \widehat{\mathcal{R}}_{n_q/2}(\mathcal{F}),
$$

where the second inequality is directed results of Formula 97 which we have already proved in the previous text.

$\square$

**Lemma C.14.** *Let $\mathcal{F}$ be a hypothesis class of feature extractors from $\mathcal{X}$ to $\mathbb{R}$. Assuming $\|\mathcal{F}\|_\infty = \kappa$. Then with probability at least $1 - \delta$ over random training set $D_q$ and $D_u$, we have :*

$$
\sup_{f \in \mathcal{F}} \left( \widehat{\mathcal{L}}_{wsc}(f; D, \widehat{\boldsymbol{S}}) - \widehat{\mathcal{L}}_{wsc}(f; \widehat{\boldsymbol{S}}) \right) \leq \left( \alpha + (\alpha + \beta/c)^2 \right) \left( \eta_0 \widehat{\mathcal{R}}_{\frac{n_q + n_u}{2}}(\mathcal{F}) + \eta_1 \left( \sqrt{\frac{\log 2/\delta}{n_q + n_u}} + \frac{\delta}{2} \right) \right)
$$

$$
+ \beta \sup_{x \in \mathcal{X}} \left\| \widehat{\boldsymbol{S}}(x)^T \widehat{\boldsymbol{S}}(x) \right\|_\infty \left( \eta_2 \widehat{\mathcal{R}}_{\frac{n_q}{2}}(\mathcal{F}) + \eta_3 \left( \sqrt{\frac{\log 2/\delta}{n_q}} + \frac{\delta}{2} \right) \right), \tag{104}
$$

*where $\eta$ is constant related to $\|\mathcal{F}\|_\infty$ and feature dimension $d$.*

*Proof.* For convenience, we will abbreviate $\alpha + (\alpha + \beta/c)^2$ as $\eta(\alpha, \beta)$. On one hand, note that fact $-2\alpha f(x_i^1 f(x_i^2)^T) + (\alpha + \beta/c)^2 \left( f(x_i^1) f(x_{i+(n/2)}^2)^T \right)^2$ takes values in range $\left[ -2\alpha\kappa^2 d, 2\alpha\kappa^2 d + (\alpha + \beta/c)^2 \kappa^4 d^2 \right]$, we apply standard generalization analysis based on Rademacher complexity and get: with probability at least $1 - \frac{\delta^2}{4}$ over the randomness of $D$ and corresponding sub-sampling tuples, we have, for any $f \in \mathcal{F}$:

$$\widehat{\mathcal{L}}_{wsc}^u(f; D, \widehat{\boldsymbol{S}}) - \mathbb{E}_D \left[ \widehat{\mathcal{L}}_{wsc}^u(f; D, \widehat{\boldsymbol{S}}) \right] \leq \eta(\alpha, \beta) \left( 32 \left( \kappa d + \kappa^2 d^2 \right) \widehat{\mathcal{R}}_{n/2}(\mathcal{F}) + \left( 4\kappa^2 d + \kappa^4 d^2 \right) \sqrt{\frac{\log 2/\delta}{n_q + n_u}} \right). \quad (105)$$

This means with probability at least $1 - \delta$ over $D$, we have: with probability at least $1 - \delta/2$ over corresponding sub-sampling tuples condition on $D$, Formula 105 holds. Hence, with probability at least probability $1 - \delta/2$ over $D$, we have:

$$\widehat{\mathcal{L}}_{wsc}^u(f; D, \widehat{\boldsymbol{S}}) - \mathbb{E}_D \left[ \widehat{\mathcal{L}}_{wsc}^u(f; D, \widehat{\boldsymbol{S}}) \right] \leq \eta(\alpha, \beta) \left( 32 \left( \kappa d + \kappa^2 d^2 \right) \widehat{\mathcal{R}}_{n/2}(\mathcal{F}) + \left( 4\kappa^2 d + \kappa^4 d^2 \right) \left( \sqrt{\frac{\log 2/\delta}{n_q + n_u}} + \frac{\delta}{2} \right) \right).$$
$$(106)$$

On the other hand, note that fact $-2\beta f(x_i^1) f(x_{i+n_q/2}^2)^T$ takes values in range $\left[ -2\beta\kappa^2 d, 2\beta\kappa^2 d \right]$ and using the exact same analysis process as above, we can obtain: with probability at least $1 - \delta/2$ over the randomness of $D$, for any $f \in \mathcal{F}$:

$$\widehat{\mathcal{L}}_{wsc}^l(f; D, \widehat{\boldsymbol{S}}) - \mathbb{E}_D \left[ \widehat{\mathcal{L}}_{wsc}^l(f; D, \widehat{\boldsymbol{S}}) \right] \leq \beta \sup_{x \in \mathcal{X}} \left\| \widehat{\boldsymbol{S}}(x)^T \widehat{\boldsymbol{S}}(x) \right\|_\infty \left( 32\kappa d \widehat{\mathcal{R}}_{\frac{n_q}{2}}(\mathcal{F}) + 4\kappa^2 d \left( \sqrt{\frac{\log 2/\delta}{n_q}} + \frac{\delta}{2} \right) \right) \quad (107)$$

By combining Formulas 106 and 107 with facts $\widehat{\mathcal{L}}_{wsc}(f; \widehat{\boldsymbol{S}}) = \mathbb{E}_D \left[ \widehat{\mathcal{L}}_{wsc}^u(f; D, \widehat{\boldsymbol{S}}) + \widehat{\mathcal{L}}_{wsc}^l(f; D, \widehat{\boldsymbol{S}}) \right]$, The proof can be concluded. $\square$

We next bound the second term in Formula 92. The following lemma bounds $\sup_{f \in \mathcal{F}} \left| \widehat{\mathcal{L}}_{wsc}(f; \widehat{\boldsymbol{S}}) - \mathcal{L}_{wsc}(f) \right|$.

**Lemma C.15.** *Let $\mathcal{F}$ be a hypothesis class of feature extractors from $\mathcal{X}$ to $\mathbb{R}^d$ with $\|\mathcal{F}\|_\infty = \kappa$. Then we have:*

$$\sup_{f \in \mathcal{F}} \left| \widehat{\mathcal{L}}_{wsc}(f; \widehat{\boldsymbol{S}}) - \mathcal{L}_{wsc}(f) \right| \leq 6\kappa^2 d \Delta(\widehat{\boldsymbol{S}}) \quad (108)$$

*Proof.* We expand $\left| \widehat{\mathcal{L}}_{wsc}(f; \widehat{\boldsymbol{S}}) - \widehat{\mathcal{L}}_{wsc}(f) \right|$ and obtain for any $f \in \mathcal{F}$:

$$\left| \widehat{\mathcal{L}}_{wsc}(f; \widehat{\boldsymbol{S}}) - \widehat{\mathcal{L}}_{wsc}(f) \right| \leq 2\beta \left| E_{(\tilde{x}, \tilde{q}), (\tilde{x}', \tilde{q}') \sim \mathbb{P}(\mathcal{X}, \mathcal{Q})^2, x \sim \mathcal{A}(\cdot|\tilde{x}), x' \sim \mathcal{A}(\cdot|\tilde{x}')} \left[ \left( \widehat{\boldsymbol{S}}(\tilde{x})_{:,\tilde{q}}^T \widehat{\boldsymbol{S}}(\tilde{x}')_{:,\tilde{q}'} - \boldsymbol{S}(\tilde{x})_{:,\tilde{q}}^T \boldsymbol{S}(\tilde{x}')_{:,\tilde{q}'} \right) f_x f_{x'}^T \right] \right|$$

$$\leq 2\kappa^2 d\beta \left| E_{\tilde{x}, \tilde{x}' \sim \mathbb{P}(\mathcal{X})^2} \left[ \widehat{\boldsymbol{S}}(\tilde{x})_{:,\tilde{q}}^T \widehat{\boldsymbol{S}}(\tilde{x}')_{:,\tilde{q}'} - \boldsymbol{S}(\tilde{x})_{:,\tilde{q}}^T \boldsymbol{S}(\tilde{x}')_{:,\tilde{q}'} \right] \right|$$

$$= 2\kappa^2 d\beta \left| E_{\tilde{x}, \tilde{x}' \sim \mathbb{P}(\mathcal{X})^2} \left[ \mathbb{P}(\boldsymbol{q} \mid \tilde{x})^T \left( \widehat{\boldsymbol{S}}(\tilde{x})^T \widehat{\boldsymbol{S}}(\tilde{x}') - \boldsymbol{S}(\tilde{x})^T \boldsymbol{S}(\tilde{x}') \right) \mathbb{P}(\boldsymbol{q} \mid \tilde{x}') \right] \right|$$

$$= 2\kappa^2 d\beta \left| E_{\tilde{x}, \tilde{x}' \sim \mathbb{P}(\mathcal{X})^2} \left[ \mathbb{P}(\boldsymbol{q} \mid \tilde{x})^T \widehat{\boldsymbol{S}}(\tilde{x})^T \widehat{\boldsymbol{S}}(\tilde{x}') \mathbb{P}(\boldsymbol{q} \mid \tilde{x}') - \mathbb{P}(\boldsymbol{y} \mid \tilde{x})^T \mathbb{P}(\boldsymbol{y} \mid \tilde{x}') \right] \right|$$

$$\leq 6\kappa^2 d\beta E_{\tilde{x} \sim \mathbb{P}(\mathcal{X})} \left[ \left\| \mathbb{P}(\boldsymbol{y} \mid \tilde{x}) - \widehat{\boldsymbol{S}}(\tilde{x}) \mathbb{P}(\boldsymbol{q} \mid \tilde{x}) \right\|_1 \right] = 6\kappa^2 d\Delta(\widehat{\boldsymbol{S}}),$$
$$(109)$$

here the 5-th inequality holds because we have:

$$|\mathbb{P}(\boldsymbol{q} \mid \tilde{x})^T \widehat{\boldsymbol{S}}(\tilde{x})^T \widehat{\boldsymbol{S}}(\tilde{x}') \mathbb{P}(\boldsymbol{q} \mid \tilde{x}') - \mathbb{P}(\boldsymbol{y} \mid \tilde{x})^T \mathbb{P}(\boldsymbol{y} \mid \tilde{x}')| \leq |\mathbb{P}(\boldsymbol{y} \mid \tilde{x}')^T \left( \mathbb{P}(\boldsymbol{y} \mid \tilde{x}) - \widehat{\boldsymbol{S}}(\tilde{x}) \mathbb{P}(\boldsymbol{q} \mid \tilde{x}) \right) |$$

$$+ |\mathbb{P}(\boldsymbol{y} \mid \tilde{x})^T \left( \mathbb{P}(\boldsymbol{y} \mid \tilde{x}') - \widehat{\boldsymbol{S}}(\tilde{x}') \mathbb{P}(\boldsymbol{q} \mid \tilde{x}') \right) |$$

$$+ | \left( \mathbb{P}(\boldsymbol{y} \mid \tilde{x}) - \widehat{\boldsymbol{S}}(\tilde{x}) \mathbb{P}(\boldsymbol{q} \mid \tilde{x}) \right)^T \left( \mathbb{P}(\boldsymbol{y} \mid \tilde{x}') - \widehat{\boldsymbol{S}}(\tilde{x}') \mathbb{P}(\boldsymbol{q} \mid \tilde{x}') \right) |$$

$$\leq 2 \left\| \mathbb{P}(\boldsymbol{y} \mid \tilde{x}) - \widehat{\boldsymbol{S}}(\tilde{x}) \mathbb{P}(\boldsymbol{q} \mid \tilde{x}) \right\|_1 + \left\| \mathbb{P}(\boldsymbol{y} \mid \tilde{x}') - \widehat{\boldsymbol{S}}(\tilde{x}') \mathbb{P}(\boldsymbol{q} \mid \tilde{x}') \right\|_1$$
$$(110)$$

$\square$

Finally, we combine the results of Lemmas C.11, C.14, C.15 to complete the final proof of Theorem C.9.

*Proof of Theorem C.9.* Take $\eta_0 = 64(\kappa d + \kappa^2 d^2), \eta_1 = 8\kappa^2 d + 2\kappa^4 d^2, \eta_2 = 64\kappa d, \eta_3 = 8\kappa^2 d, \eta_4 = 24\kappa^2 d$. By substituting the results of Lemmas C.14 and C.11 into Formula 92, and then the proof can be completed. $\square$

### C.3. Detail of Corollary 3.8

By substituting Theorems 3.4 and 3.7 into Theorem D.7 in HaoChen et al. 2021, the corollary can be directly obtained. We restate Theorem D.7 in HaoChen et al. 2021 as follows:

**Theorem C.16.** *(Theorem D.7 in HaoChen et al. 2021) Assume representation dimension $d > 4r + 2$, Recall $\lambda_i$ be the $i$-th largest eigenvalue of the normalized adjacency matrix. Then, for any $\epsilon > 0$ and $\widehat{f} \in \mathcal{F}$ such that $\mathcal{L}_{wsc}(\widehat{f}) \leq \mathcal{L}_{wsc}(f^*) + \epsilon$, we have:*

$$\varepsilon(\widehat{f}) \leq \varepsilon(f^*) + \frac{d}{\Delta_\lambda^2}\epsilon, \tag{111}$$

*where $\Delta_\lambda \triangleq \lambda_{\lfloor 3d/4 \rfloor} - \lambda_d$ is the eigenvalue gap between the $\lfloor 3d/4 \rfloor$-th and the $d$-th eigenvalue.*

## D. Related Work

Many methods have been developed to solve various weakly-supervised learning problems. At the same time, research on representation learning has also made significant breakthroughs in recent years. In this section, we revisit these related work and divide them into two parts: weakly-supervised learning and representation learning.

### D.1. Weakly-Supervised Learning

In real world, the widely existing supervised information is weakly-supervised information which is either inaccurate or ambiguous. In order to enable effective and accurate learning from this weakly-supervised information, previous work has proposed a series of methods for weakly-supervised learning. The two typical paradigms related to weakly-supervised learning are noisy label learning (NLL) and partial label learning (PLL) , which is what we will introduce in this section.

**Noisy Label Learning (NLL).** Noisy labels may be caused by annotation errors, and overfitting to these label errors will lead to poor model performance (Zhang et al., 2021a; Wei et al., 2024; Li et al., 2023). Several strategies have been developed to mitigate the label noise. The mainstream of NLL methods cover the following aspects: using robust loss function (Zhang and Sabuncu, 2018; Wang et al., 2019), modeling the noise transfer matrix, performing sample selection and correcting incorrect labels. In particular, label correction methods have shown promising results than other methods in noisy label learning. Liu et al. 2020 prove that the model can accurately predict a subset of mislabeled examples during the early learning phase. This observation implies a potential strategy for rectifying the corresponding labels, which can be achieved by generating novel labels tantamount to either soft or hard pseudo-labels estimated via the model (Tanaka et al., 2018). Han et al. 2018 propose Co-teaching which trains two different networks for the label correction. Inspired by the fact that clean samples have a smaller loss in the early learning stage, Arazo et al. 2019 apply a mixture model to the losses of each sample to estimate the probability that a sample is mislabeled and correct the loss based on the prediction of the network. Similar to the previous two methods, DivideMix (Li et al., 2020), deploys two neural networks to conduct mutual sample selection and apply semi-supervised learning methodology, where the targets are computed based on the average predictions obtained from different data augmentations.

**Partial Label Learning (PLL).** Partial labels, while preventing the omission of the correct label, introduce greater ambiguity into the labeling process. The prior works can be divided into average-based strategies and identification-based strategies. The average-based methods usually treat all candidate labels equally. Lv et al. 2023 discussed and analyzed the robustness performance of different average-based loss. However, those methods may introduce the misleading false positive label into the training process. To overcome these limitations, identification-based strategies, which regard the correct label as a latent variable and aim to identify it from the candidate label set, have drawn intensive attention and achieved remarkable progress. PRODEN (Lv et al., 2020) and CC (Feng et al., 2020b) utilize the predictions generated by the predictive model as label information, thus assigning more weights to labels that are more likely to be correct. Wen et al. 2021 propose a family of loss functions for label disambiguation. Wu et al. 2022 perform supervised learning on non-candidate labels and employ consistency regularization on candidate labels.

In recent years, a more practical setting called instance-dependent partial label learning (IDPLL) has also received considerable attention. In IDPLL, the candidate set is generated from labels that have a certain semantic similarity with the true label, which also increases the difficulty of disambiguation. Qiao et al. 2023b propose to explicitly model the instance-dependent generation process by decomposing it into separate generation steps. Wu et al. 2024 introduce a novel rectification process to ensure that candidate labels consistently receive higher confidence than non-candidates. He et al. 2023 propose a two-stage framework employing normalized entropy for selective disambiguation of well-disambiguated examples. CEL (Yang et al., 2025) employs a class associative loss to enforce intra-candidate similarity and inter-set dissimilarity among the class-wise embeddings for each sample.

Some research has also turned to trying to use a more unified framework to solve both NLL and PLL problems. ULAREF (Qiao et al., 2024) improves the reliability of label refinement by globally detecting the reliability of the prediction model and locally enhancing the supervision signal, thereby improving the performance of the prediction model. GFWS (Chen et al., 2024) treat the ground-truth labels as latent variables and try to model the entire distribution of all possible labeling entailed by weakly-supervised information, thus allowing a unified solution to deal with NLL and PLL. Although these methods try to uniformly exploit the common features of weakly-supervised information to solve the weakly-supervised learning challenges, they all ignore the potential of high-quality representations to achieve label disambiguation.

### D.2. Representation Learning and Weakly-Supervised Representation Learning

In recent years, contrastive learning has become dominant in representation learning because they can learn more distinct representations. A plethora of works has explored the effectiveness of contrastive learning in unsupervised representation learning (He et al., 2020; Oord et al., 2018; Chen et al., 2020; Caron et al., 2020; Zbontar et al., 2021). Meanwhile, there are also many works trying to introduce more information into contrastive learning or bring new insights from different views (Khosla et al., 2020; HaoChen et al., 2021; Sun et al., 2023; Cui et al., 2023; Zhou et al., 2024).

Khosla et al. 2020 attempt to introduce supervised information into contrastive learning. The approach regards samples in same classes as positive samples and achieves significant performance improvements on multiple supervised learning tasks. Due to the success of contrastive learning, many researchers have made a lot of attempts to improve the performance on weakly-supervised learning incorporated with the advantages of contrastive learning. On the line of NLL, the role of contrastive learning is different. MOIT (Ortego et al., 2021) uses the agreement between the features learned by contrastive learning and the original labels to identify mislabeled samples. Sel-CL (Li et al., 2022) utilizes k-nn nearest neighbors to select confident sample pairs and use these sample pairs for supervised contrastive learning. TCL (Huang et al., 2023) uses a Gaussian mixture model to disambiguate labels, and then uses self-supervised contrastive learning to further learn more robust representations. For PLL, PiCO (Wang et al., 2022) integrates contrastive learning with prototype learning. The former facilitates the formation of well-structured clusters, which in turn enables prototype learning to acquire prototype representations. The latter assists in the selection of positive samples for contrastive learning.

On the other hand, some researchers try to view and explain contrastive learning from different perspectives. Inspired by the widespread application of spectral graph theory in the field of machine learning, HaoChen et al. 2021 first regard contrastive learning as a problem of graph clustering on augmentation graph and introduce a spectral contrastive loss, which greatly promoted the progress of unsupervised contrastive learning. Along this line, Sun et al. 2023 introduce supervised information to this spectral contrastive learning framework, transforming the augmentation graph in the original problem into a perturbation graph, thus achieving great performance improvement on open-world semi-supervise learning task.

Despite their success in the respective field, they all overlooked the importance of correctly utilizing weakly-supervised information. Relevant theories and experiments have verified that simply introducing inaccurate noise labels for contrastive learning is ineffective and even harmful (Cui et al., 2023). Therefore, it is necessary to design a special contrastive learning framework for weakly-supervised information. Overall, our method is the first approach to utilize weakly-supervised information for contrastive learning from a graph spectral theory perspective. Sufficient experiments and complete and rigorous theory guarantee the effectiveness of the proposed method.

## E. Implementation Details

In this section, we provide more details on the implementation of our approach. In general, our proposed framework contains three training strategies: supervised loss, strong-weak augmentation consistency regularization (Xie et al., 2020), and the proposed weakly-supervised contrastive learning (WSC) loss. Among them, the supervised loss varies with different settings,

while the other two losses do not change with the setting.

Specifically, given any sample $x$, its weakly-supervised information can be denoted as $q \in \mathcal{Q}$. Let the extracted feature embedding be $f(x)$, the final probability prediction of the neural network be $g(x)$, and the strong augmentation function be $\mathcal{A}_s(\cdot)$. Our method can be formalized as:

$$\mathcal{L}(x,q) = \mathcal{L}_{sup}(g(x),q) + \mathcal{L}_{CE}(g(\mathcal{A}_s(x)), \tilde{g}(\mathcal{A}_w(x))) + \mathcal{L}_{wsc}(f), \qquad (112)$$

where $\tilde{g}(x)$ means the prediction is made by a fixed copy of current model, indicating that the gradient is not propagated. The first term in equation 112 means the supervised loss which will be introduced next. The second term is the so called strong-weak augmentation consistency regularization, which consistently utilize the prediction of the weakly-augmented data to train the strongly-augmented data, and it is commonly used in weakly-supervised learning (Xie et al., 2020; Wang et al., 2022; Wu et al., 2022; Chen et al., 2024). The third term is the proposed WSC loss, which needs at least three inputs: the embeddings of the features of two different views of the sample and the constructed matrix $\widehat{S}$. In this paper, we compute this loss through Algorithm 1 which assumes a uniform class distribution, pseudo code of computing WSC loss without this assumption can be found in Algorithm 2. We left method for construct $\widehat{S}$ in next section.

### E.1. Noisy Label Learning

**Setup.** For CIFAR-10 and CIFAR-100, an 18-layer PreAct ResNet is employed and stochastic gradient descent (SGD) is utilized for training with a momentum value of 0.9, a weight decay factor of 0.001, and a batch size of 128 over a total of 250 epochs. The initial learning rate is set to 0.02 and adjusted by a cosine learning rate scheduler. In our framework, both the projection head and the classification head are configured as a two-layer Multilayer Perceptron (MLP) with a dimension of 256 and the number of classes. The parameters are set to $\alpha = 1$, $\beta = 12$ for CIFAR-10 and $\alpha = 2$, $\beta = 300$ for CIFAR-100. Additionally, for CIFAR-100, the last term of our loss function is scaled by a factor of 3. In addition, we also conduct experiments on CIFAR-10N and CIFAR-100N, where we use a 34-layer ResNet as the backbone for feature extraction, others are same as the previous one. Besides, a 50-layer ResNet pretrained with ImageNet-1K was used to train for 15 epochs on Clothing1M. The batch size are set to be 64 and the initial learning rate is 0.002 and then multiply by a factor of 0.1 in the 7th epoch. We also set $\alpha = 1$, $\beta = 28$. The detailed hyper-parameters are presented in Table 4.

*Table 4.* Hyper-parameters for **noisy label learning** used in experiments.

| Hyper-parameter | CIFAR-10 (CIFAR-10N) | CIFAR-100 (CIFAR-100N) | Clothing1M |
|---|---|---|---|
| Image Size | 32 | 32 | 224 |
| Model | PreAct-ResNet-18 (ResNet-34) | PreAct-ResNet-18 (ResNet-34) | ResNet-50 (ImageNet-1K Pretrained) |
| Batch Size | 128 | 128 | 64 |
| Learning Rate | 0.02 | 0.02 | 0.002 |
| Weight Decay | 1e-3 | 1e-3 | 1e-3 |
| LR Scheduler | Cosine | Cosine | MultiStep |
| Training Epochs | 250 | 250 | 15 |
| Classes | 10 | 100 | 14 |
| $\alpha$ | 1 | 2 | 1 |
| $\beta$ | 12(6) | 300 | 28 |

**Baselines.** The performance of proposed method for noisy label is compared against ten baselines:

- CE, which utilizes the standard cross-entropy loss directly to train the model in a batch.

- DivideMix (Li et al., 2020), which regards noisy instances as unlabeled data and employs the strategy involving label co-refinement and co-guessing.

- ELR (Liu et al., 2020), which focuses on early learning via regularization to preclude the memorization of incorrect labels.

- SOP (Liu et al., 2022), which models the noise with sparse over-parameterization and exploits implicit regularization.

*Table 5.* Comparisons with each methods on CIFAR-10N, CIFAR-100N and Clothing1M. Each runs has been repeated 3 times with different randomly-generated noise and we report the best accuracy.

| Dataset | CIFAR-10N | | | CIFAR-100N | | Clothing1M |
|---|---|---|---|---|---|---|
| Noisy Type | Random1 | Aggregate | Worst | Clean | Noisy | Ins. |
| CE | $85.02_{\pm0.65}$ | $87.77_{\pm0.38}$ | $77.69_{\pm1.55}$ | $76.70_{\pm0.74}$ | $55.50_{\pm0.66}$ | 69.10 |
| Forward | $86.88_{\pm0.50}$ | $88.24_{\pm0.22}$ | $79.79_{\pm0.46}$ | $76.18_{\pm0.37}$ | $57.01_{\pm1.03}$ | - |
| Co-teaching | $90.33_{\pm0.13}$ | $91.20_{\pm0.13}$ | $83.83_{\pm0.13}$ | $73.46_{\pm0.09}$ | $60.37_{\pm0.27}$ | - |
| ELR | $94.43_{\pm0.41}$ | $94.83_{\pm0.10}$ | $91.09_{\pm1.60}$ | $78.57_{\pm0.12}$ | $66.72_{\pm0.07}$ | 72.90 |
| GFWS | $94.86_{\pm0.07}$ | $95.30_{\pm0.03}$ | $93.55_{\pm0.14}$ | $78.53_{\pm0.21}$ | $68.07_{\pm0.33}$ | 74.02 |
| CORES | $94.45_{\pm0.14}$ | $95.25_{\pm0.09}$ | $91.66_{\pm0.09}$ | $73.87_{\pm0.16}$ | $55.72_{\pm0.42}$ | 73.20 |
| SOP | $95.28_{\pm0.13}$ | $95.61_{\pm0.13}$ | $93.24_{\pm0.21}$ | $78.91_{\pm0.43}$ | $67.81_{\pm0.23}$ | 73.50 |
| WSC (Ours) | $\mathbf{96.13_{\pm0.52}}$ | $\mathbf{96.50_{\pm0.72}}$ | $\mathbf{93.60_{\pm0.21}}$ | $\mathbf{81.31_{\pm0.16}}$ | $\mathbf{71.00_{\pm0.16}}$ | $\mathbf{74.75_{\pm0.18}}$ |

- GFWS (Chen et al., 2024), which is a unified framework that uses Expectation-Maximization to model noisy label as latent variables.

- ULAREF (Qiao et al., 2024), which trains the predictive model with refined labels through global detection and local enhancement.

- ProMix (Xiao et al., 2023), which meticulously selects, dynamically extends, and optimally utilizes clean sample sets within the devised semi-supervised learning framework.

- MOIT (Ortego et al., 2021), which jointly exploits contrastive learning and classification to enhance performance against label noise, with contributions including an ICL loss, a novel label noise detection method, and a fine-tuning strategy.

- Sel-CL (Li et al., 2022), which leverages nearest neighbors to select confident pairs for supervised contrastive learning.

- TCL (Huang et al., 2023), which disambiguate labels by a Gaussian mixture model and uses self-supervised contrastive learning to further learn more robust representations.

**Discussion.** Recent works on NLL can be categorized into three types, with methods based on the noise transition matrix demonstrating empirical effectiveness. In our proposed framework, we use the cross-entropy loss to align the model output conditioned by the estimated noisy transition matrix and the noisy label. Besides, in order to better learn from the noisy label, we follow Li et al. 2021 and impose the following regularization constraints on the estimated noise transition matrix $\boldsymbol{T}$. The above two terms will construct the supervised loss as:

$$\mathcal{L}_{sup}(g(x), q) = \mathcal{L}_{CE}(g(x)\boldsymbol{T}, q) + \lambda \log \det(\boldsymbol{T}), \tag{113}$$

where $\lambda$ is a regularization coefficient.

**Additional Experiments.** In addition to the results we present in the main paper which mainly focuses on the the simulated datasets, we also verify the effectiveness of the proposed method on the CIFAR-N dataset (Wei et al., 2022), which equips the training samples of CIFAR dataset with human-annotated noisy label. On CIFAR-10N, we used three noise types: *Aggregate*, *Random1*, and *Worst*, with corresponding noise rates of 9.03%, 17.23%, and 40.21%. For CIFAR-100N, we compared the model performance under *Clean* and *Noisy* settings with a noise rate of 40.21%. We also include a full comparison on Clothing1M which has more realistic and large-scale instance noise (Xiao et al., 2015).

The results shown in Table 5 show the superiority of our method on these datasets. It is particularly noteworthy that our method performs very well on CIFAR-100N, which should be due to the fact that our contrastive learning method specially designed for weakly supervised learning helps learn more and better feature representations. It is worth noting that since the noise type of these datasets is not suitable for estimation using a noise transition matrix, we ignore the second term of Equation 113 during training.

**Qualitative Results.** Figure 2 visualizes the learned representations under a high noise ratio. Compared to ELR+, which employs two distinct backbones to obtain more meaningful representations, our method generates well-defined structures with semantically meaningful information, leading to superior accuracy in these settings.

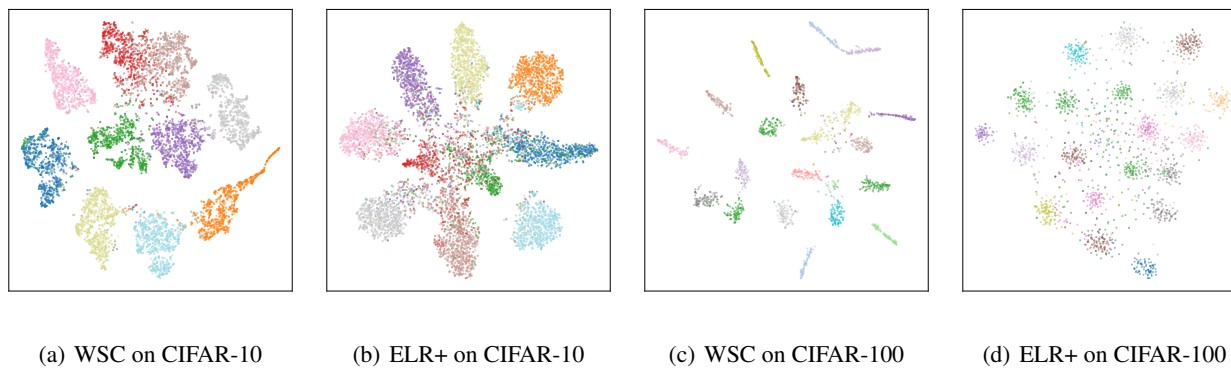

(a) WSC on CIFAR-10      (b) ELR+ on CIFAR-10      (c) WSC on CIFAR-100      (d) ELR+ on CIFAR-100

*Figure 2.* We present t-SNE visualizations of the learned representations on the CIFAR dataset with 90% symmetric noise. Figure 2(a) and Figure 2(b) illustrate the results for WSC and ELR+ on CIFAR-10, while Figure 2(c) and Figure 2(d) display the results for WSC and ELR+ on CIFAR-100 under the same noise condition, where only 20 categories with the highest accuracy for each model are shown.

### E.2. Partial Label Learning

**Setup.** For simulated PLL settings, we use Wide-ResNet-28-2 trained from scratch and ResNet-18 pre-trained with ImageNet-1K as our feature extractors on the CIFAR dataset and CUB-200 dataset, respectively. Following most of experimental setups conducted on the contrastive network, we use a 2-layer MLP that outputs 128-dimensional embeddings as our projection head. The batch size is 256 and the weight decay is 0.0001 across all the PLL settings. We set the initial learning rate to be 0.1 and adjust it by MultiStep and Cosine scheduler on CIFAR datasets and CUB-200 respectively. On CIFAR-10 and CIFAR-100 (CIFAR-100-H), we train our model for 200 epochs using parameters $\alpha = 1$, $\beta = 12$ for CIFAR-10 and $\alpha = 2$, $\beta = 300$ for CIFAR-100. For the CUB dataset, we train for 300 epochs with parameters $\alpha = 2$ and $\beta$ linearly increasing from 0 to 400. The detailed choice of hyper-parameters is provided in Table 6. Furthermore, for IDPLL settings, we use ResNet-34 pre-trained with ImageNet-1K as our feature extractors following (Yang et al., 2025). We train our model for 500 epochs and set the initial learning rate be 0.1 and adjust it by Cosine scheduler on both four fine-grained datasets. The detailed choice of hyper-parameters is provided in Table 7.

*Table 6.* Hyper-parameters for **simulated partial label learning** used in experiments.

| Hyper-parameter | CIFAR-10 | CIFAR-100 (CIFAR-100H) | CUB-200 |
|---|---|---|---|
| Image Size | 32 | 32 | 224 |
| Model | Wide-ResNet-28-2 | Wide-ResNet-28-2 | ResNet-18 (ImageNet-1K Pretrained) |
| Batch Size | 256 | 256 | 256 |
| Learning Rate | 0.1 | 0.1 | 0.01 |
| Weight Decay | 1e-4 | 1e-4 | 1e-5 |
| LR Scheduler | MultiStep | MultiStep | Cosine |
| Training Epochs | 200 | 200 | 300 |
| Classes | 10 | 100 | 200 |
| $\alpha$ | 1 | 2 | 2 |
| $\beta$ | 12 | 300 | 0-400 |

**Discussion.** We provide the detailed implementation of $\mathcal{L}_{sup}$ on PLL setting here. The average-based methods has been widely used in recent PLL works. Given the probability prediction $g(x)$, we can denote the classifier as $h(x) = \arg\min\limits_{i \in \mathcal{Y}} g_i(x)$.

Besides, the candidate set $q$ can be derived from the weakly supervised information, thus, the family of average partial label losses can be formally written as:

$$\mathcal{L}_{avg}(h(x), q) = \frac{1}{|q|} \sum_{i \in q} \ell(h(x), i), \tag{114}$$

Table 7. Hyper-parameters for **instance-dependent partial label learning** used in experiments.

| Hyper-parameter | CUB200 | CARS196 | DOGS120 | FGVC100 |
|---|---|---|---|---|
| Model | ResNet-34(Pretrained) | ResNet-34(Pretrained) | ResNet-34(Pretrained) | ResNet-34(Pretrained) |
| Image Size | 224 | 224 | 224 | 224 |
| Batch Size | 256 | 256 | 256 | 256 |
| Learning Rate | 0.01 | 0.01 | 0.01 | 0.01 |
| Weight Decay | 1e-5 | 1e-5 | 1e-5 | 1e-5 |
| LR Scheduler | Cosine | Cosine | Cosine | Cosine |
| Training Epochs | 500 | 500 | 500 | 500 |
| Classes | 200 | 196 | 120 | 100 |
| $\alpha$ | 2 | 2 | 2 | 2 |
| $\beta$ | 0-400 | 0-400 | 0-240 | 0-200 |

where $|\cdot|$ represents the cardinality. There are different loss functions $\ell$ that can be chosen, and we use the commonly used cross-entropy loss to implement our framework. The supervised loss can then be obtained from the above average partial label loss.

**Baselines.** We compare the proposed framework with seven methods handling partial labeled data:

- LWS (Wen et al., 2021), which weights candidate labels and non-candidate labels through a leverage parameter.

- PRODEN (Lv et al., 2020), which progressively identifies correct labels through the model output.

- CC (Feng et al., 2020b), which derives a classifier consistent risk estimator through a transition matrix.

- MSE (Feng et al., 2020a), which uses Mean Square Error loss to learn with multiple complementary labels.

- RCR (Wu et al., 2022), which performs supervised learning on non-candidate labels and employ consistency regularization on candidate labels.

- PiCO (Wang et al., 2022), which utilizes a contrastive loss term to enhance the model disambiguation ability.

We also include four IDPLL methods to compare:

- IDGP (Qiao et al., 2023b), which proposes to explicitly model the instance-dependent generation process by decomposing it into separate generation steps.

- DIRK (Wu et al., 2024), which introduces a novel rectification process to ensure that candidate labels consistently receive higher confidence than non-candidates.

- NPELL (He et al., 2023), which proposes a two-stage framework employing normalized entropy for selective disambiguation of well-disambiguated examples.

- CEL (Yang et al., 2025), which employs a class associative loss to enforce intra-candidate similarity and inter-set dissimilarity among the class-wise embeddings for each sample.

**Additional Experiments.** To further investigate the performance of our proposed method on fine-grained image classification tasks, we conduct additional experiments using CIFAR-100 with hierarchical labels (CIFAR-100-H) (Wang et al., 2022), where candidate labels are generated from the 20 superclasses in CIFAR-100. Furthermore, we extend our experiments on CUB-200, as presented in the main paper, to include a wider range of partial label ratios, specifically 0.01, 0.05, 0.1. We also extend our experiments on IDPLL setting. Following previous work (Wu et al., 2024; Yang et al., 2025), we conduct experiments on four fine-grained datasets: CUB-200 (Wah et al., 2011a), CARS-196 (Krause et al., 2013), DOGS-120 (Khosla et al., 2011), and FGVC-100 (Maji et al., 2013), and we employed the IDPLL noisy label generation method proposed by VALEN (Xu et al., 2021) to generate instance-dependent noisy labels which is the standard experimental protocol in IDPLL settings.

Table 8. Comparisons with each methods on simulated PLL datasets. Each runs has been repeated 3 times with different randomly-generated partial labels and we report the mean and std values of last 5 epochs.

| Dataset | CIFAR-100-H | | | CUB-200 | | |
|---|---|---|---|---|---|---|
| Partial Ratio | 0.1 | 0.5 | 0.8 | 0.01 | 0.05 | 0.1 |
| PiCO | $76.55_{\pm0.68}$ | $74.98_{\pm0.42}$ | $66.38_{\pm0.24}$ | $74.14_{\pm0.28}$ | $72.12_{\pm0.74}$ | $\mathbf{62.02_{\pm1.16}}$ |
| LWS | $63.88_{\pm0.10}$ | $59.37_{\pm0.25}$ | $40.33_{\pm0.19}$ | $73.74_{\pm0.23}$ | $39.74_{\pm0.43}$ | $12.30_{\pm0.77}$ |
| PRODEN | $67.42_{\pm0.91}$ | $64.89_{\pm0.75}$ | $50.41_{\pm0.38}$ | $72.34_{\pm0.04}$ | $62.56_{\pm0.10}$ | $35.89_{\pm0.05}$ |
| CC | $58.11_{\pm0.46}$ | $56.44_{\pm0.37}$ | $30.11_{\pm0.48}$ | $56.63_{\pm0.01}$ | $55.61_{\pm0.02}$ | $17.01_{\pm1.44}$ |
| MSE | $57.88_{\pm0.85}$ | $51.13_{\pm0.32}$ | $28.33_{\pm0.44}$ | $61.12_{\pm0.51}$ | $22.07_{\pm2.36}$ | $11.40_{\pm2.42}$ |
| GFWS | $77.10_{\pm0.18}$ | $76.08_{\pm0.25}$ | $61.35_{\pm0.71}$ | $73.19_{\pm0.42}$ | $70.77_{\pm0.20}$ | $47.44_{\pm0.15}$ |
| RCR | $77.38_{\pm0.33}$ | $75.78_{\pm0.25}$ | $65.31_{\pm0.19}$ | - | - | - |
| WSC (Ours) | $\mathbf{78.31_{\pm0.17}}$ | $\mathbf{76.67_{\pm0.11}}$ | $\mathbf{66.77_{\pm0.05}}$ | $\mathbf{76.88_{\pm0.39}}$ | $\mathbf{74.55_{\pm0.17}}$ | $56.13_{\pm0.34}$ |

Table 9. Comparisons with each methods on IDPLL settings with four fine-grained datasets. Each runs has been repeated 3 times with different randomly-generated partial labels and we report the mean and std values of last 5 epochs.

| Dataset | WSC (Ours) | CEL | DIRK | NEPLL | IDGP | PiCO | PRODEN |
|---|---|---|---|---|---|---|---|
| CUB-200 | $\mathbf{68.90_{\pm0.27}}$ | $68.60_{\pm0.10}$ | $66.60_{\pm1.07}$ | $62.88_{\pm1.66}$ | $58.16_{\pm0.58}$ | $58.56_{\pm0.90}$ | $65.05_{\pm0.14}$ |
| CARS-196 | $\mathbf{87.88_{\pm1.18}}$ | $86.22_{\pm0.08}$ | $85.31_{\pm0.77}$ | $85.05_{\pm0.17}$ | $79.56_{\pm0.46}$ | $70.15_{\pm1.63}$ | $83.35_{\pm0.05}$ |
| DOGS-120 | $\mathbf{79.75_{\pm0.32}}$ | $78.18_{\pm0.12}$ | $75.97_{\pm0.29}$ | $74.84_{\pm0.09}$ | $66.79_{\pm0.38}$ | $67.80_{\pm0.06}$ | $70.94_{\pm0.43}$ |
| FGVC-100 | $77.80_{\pm0.39}$ | $\mathbf{78.36_{\pm0.19}}$ | $76.86_{\pm3.54}$ | $75.36_{\pm0.59}$ | $72.48_{\pm0.86}$ | $63.52_{\pm0.94}$ | $69.34_{\pm0.39}$ |

Our method outperforms other methods except the experiment on CUB-200 with 0.1 partial ratio as shown in Table 8. The improvement can be attributed to PiCO's class prototype-based pseudo-labeling mechanism for disambiguation, a strategy that has proven effective in several fine-grained classification tasks. Moreover, our approach avoids the need for additional complex techniques. By incorporating our contrastive loss into the training objective, we achieve an **8.69%** improvement over GFWS in this setting, highlighting the potential for even greater performance gains. Similarity, our WSC method demonstrates strong performance in the instance-dependent partial label learning (IDPLL) setting as shown in Table 9 without introducing any complex modules or additional technical components. The results further validate that WSC can effectively address the challenges of instance-dependent label ambiguity while maintaining a straightforward implementation.

### E.3. Ablation Study

We conduct ablation experiments on CIFAR-100, as shown in Table 10. The results demonstrate that the proposed WSC loss consistently improves performance, especially under high noise rates, where it increases performance by **16.22%** and **8.99%**, respectively, further validating the effectiveness of the proposed method.

Table 10. Ablation studies of our proposed algorithm on CIFAR-100 with different ratio of noisy label and partial label.

| Components | | | Noisy Label | | | Partial Label | | |
|---|---|---|---|---|---|---|---|---|
| Supervised Loss | Consistency Regularizer | WSC Loss | 0.5 | 0.8 | 0.9 | 0.1 | 0.2 | 0.3 |
| ✓ | | | 53.89 | 41.35 | 20.11 | 61.50 | 55.22 | 35.44 |
| ✓ | ✓ | | 75.43 | 66.40 | 45.10 | 75.80 | 73.12 | 60.16 |
| ✓ | ✓ | ✓ | 77.51 | 71.92 | 61.32 | 77.26 | 75.13 | 69.15 |

# F. Details of Construction of Semantic Similarity under Different Scenarios

In this section, we will introduce more details respectively regarding the construction of $\widehat{S}$ in the scenarios of noisy label learning and partial label learning.

---

**Algorithm 2** batch data estimation of $\mathcal{L}_{wsc}$ under general scenario

---

**Input:** features of two augmentation view of batch data with corresponding weakly-supervised information $\boldsymbol{X}_Q^1, \boldsymbol{X}_Q^2 \in \mathbb{R}^{B_Q \times d}, Q \in \mathcal{Q}^{B_Q}$, features of two augmentation view of batch data without any supervised information $\boldsymbol{X}_U^1, \boldsymbol{X}_U^2 \in \mathbb{R}^{B_U \times d}$, $\boldsymbol{S}$ in Proposition. 2.1, class prior $\mathbb{P}(\boldsymbol{y}) \in \mathbb{R}^{c \times 1}$, proportional coefficients $\alpha, \beta$

**Output:** batch data estimated loss $\widehat{\mathcal{L}}_{wsc}$

Compute $\boldsymbol{S}(\boldsymbol{X}) \in \mathbb{R}^{c \times B_Q}, \boldsymbol{S}(\boldsymbol{X})_{:,x} = \boldsymbol{S}(x)_{:,q}$

Compute $\boldsymbol{S}'(\boldsymbol{X}) \in \mathbb{R}^{1 \times B_Q}, \boldsymbol{S}'(\boldsymbol{X}) = \mathbb{P}(\boldsymbol{y})^T \boldsymbol{S}(\boldsymbol{X})$

Compute $\widehat{\mathcal{L}}_1 = \frac{\mathrm{Tr}\left[\boldsymbol{X}_Q^1 (\boldsymbol{X}_Q^2)^T\right] + Tr\left[\boldsymbol{X}_U^1 (\boldsymbol{X}_U^2)^T\right]}{B_Q + B_U}$

Compute $\widehat{\mathcal{L}}_2 = \frac{\left\| (\boldsymbol{S}(\boldsymbol{X})^T \boldsymbol{S}(\boldsymbol{X})) \otimes (\boldsymbol{X}_Q^1 (\boldsymbol{X}_Q^2)^T) \right\|_1}{B_Q^2}$

Compute $\widehat{\mathcal{L}}_3 = \frac{\left\| \boldsymbol{X}_Q^1 (\boldsymbol{X}_Q^2)^T \right\|_2^2 + \left\| \boldsymbol{X}_U^1 (\boldsymbol{X}_U^2)^T \right\|_2^2}{B_Q^2 + B_U^2}$

Compute $\widehat{\mathcal{L}}_4 = \frac{\left\| \sqrt{\boldsymbol{S}'(\boldsymbol{X})^T \boldsymbol{S}'(\boldsymbol{X})} \otimes (\boldsymbol{X}_Q^1 (\boldsymbol{X}_Q^2)^T) \right\|_2^2}{B_Q^2}$

Take $\boldsymbol{X}_Q = [\boldsymbol{X}_Q^1; \boldsymbol{X}_Q^2], \boldsymbol{X}_U = [\boldsymbol{X}_U^1; \boldsymbol{X}_U^2], \boldsymbol{S}'(\boldsymbol{X}) = [\boldsymbol{S}'(\boldsymbol{X}), \boldsymbol{S}'(\boldsymbol{X})]$

Compute $\widehat{\mathcal{L}}_5 = \frac{\left\| \sqrt{\mathrm{Diag}(\boldsymbol{S}'(\boldsymbol{X}))} \boldsymbol{X}_Q \boldsymbol{X}_U^T \right\|_2^2}{4 B_Q \times B_U}$.

**Return:** $\widehat{\mathcal{L}}_{wsc}(\boldsymbol{X}_Q^1, \boldsymbol{X}_Q^2, \boldsymbol{X}_U^1, \boldsymbol{X}_U^2, \boldsymbol{S}, Q) = -2\alpha \widehat{\mathcal{L}}_1 - 2\beta \widehat{\mathcal{L}}_2 + \alpha^2 \widehat{\mathcal{L}}_3 + \beta^2 \widehat{\mathcal{L}}_4 + 2\alpha\beta \widehat{\mathcal{L}}_5$

---

### F.1. Details of Construction of Semantic Similarity under Noisy Label Setting

**Using Environment Information.** For instance-independent noisy label setting, there exists a $\boldsymbol{T}$ satisfies $\boldsymbol{T} = \boldsymbol{T}(x)$ holding almost everywhere in $\mathcal{X}$. Hence, inspired by our Proposition 2.4, in this setting, we can use $\widehat{\boldsymbol{S}}(x) = (\widehat{\boldsymbol{T}})^{-1}$ for any $x \in \mathcal{X}$, where $\widehat{\boldsymbol{T}}$ is estimated noisy transition matrix. Quite a number of studies have already investigated how to estimate the noise transition matrix (Liu and Tao, 2016; Patrini et al., 2017; Xia et al., 2019; Li et al., 2021; Zhang et al., 2021b; Lin et al., 2023). These methods are usually divided into two categories: anchor-based methods and anchor-free methods. The former (Liu and Tao, 2016; Patrini et al., 2017; Xia et al., 2019) usually fits the noise posterior directly, and then estimates the noise matrix by this estimated noise posterior probability on the data of reliable anchor points. In recent years, it is the latter (Li et al., 2021; Zhang et al., 2021b; Lin et al., 2023) that has achieved better results. They obtain the estimated noise matrix by searching for the matrix that minimizes a specific metric within the matrix family that can linearly represent all noise posteriors. Specifically, Li et al. 2021 uses volume of estimated noisy matrix as this specified metric and Zhang et al. 2021b uses total variation as this specified metric. Lin et al. 2023 additionally adopts a bi-level optimization to increase the robustness against the error of the estimation of the noise posterior and thus achieving better performance. All above methods can be used to construct $\widehat{\boldsymbol{S}}$.

**Using Both Environment and Samples Information.** However, the aforementioned way of constructing semantic similarity only takes into account the weakly-supervised information while ignoring the information of the samples themselves. Therefore, even if we obtain a completely accurate noise matrix through estimation, such a construction method will still add a variance term to the learning error ($\sup_{x \in \mathcal{X}} \left\| \widehat{\boldsymbol{S}}(x)^T \widehat{\boldsymbol{S}}(x) \right\|_\infty$ in Formula 29), thus resulting in sub-optimal performance in practice. Recalling Bayes theorem:

$$\mathbb{P}(y \mid x) = \sum_{q \in \mathcal{Q}} \mathbb{P}(y, q \mid x) = \sum_{q \in \mathcal{Q}} \mathbb{P}(y \mid x, q) \mathbb{P}(q \mid x). \tag{115}$$

Thus we can take $\widehat{\boldsymbol{S}}(x)_{y,q} = \widehat{\mathbb{P}}(y \mid x, q)$ where $\widehat{\mathbb{P}}(y \mid x, q)$ is estimation posterior with corresponding weakly-supervised information. With a estimated noisy transition matrix $\widehat{\boldsymbol{T}}$, we can estimate $\widehat{\mathbb{P}}(y \mid x, q)$ as follows:

$$\widehat{\mathbb{P}}(y \mid x, q) \propto \widehat{\mathbb{P}}(y, q \mid x) = \widehat{\mathbb{P}}(y \mid x) \widehat{\boldsymbol{T}}_{q,y} = g(\mathcal{A}_w(x)) \widehat{\boldsymbol{T}}_{q,y}, \tag{116}$$

where $g(\mathcal{A}_w(x))$ denotes posterior probability predicted by the current neural network.

Such a construction method can be regarded as constructing semantic similarity by integrating the information of the current neural network, the sample information and the estimated environmental information. Therefore, better performance has been achieved in practice.

**Using Samples Information.** For instance-dependent setting which we do not assume that all samples share the same noise transition matrix, the methods mentioned above become invalid. In such setting, we are unable to utilize the environmental information and can only rely on the properties of the neural network itself as well as the sample information for estimation $\widehat{\mathbb{P}}(y \mid x, q)$. This type of method has also been widely discussed in noisy label learning (Liu et al., 2020; Li et al., 2022; Ortego et al., 2021; Liu et al., 2022; Xiao et al., 2023; Huang et al., 2023; Qiao et al., 2024) and has achieved higher performance in practice. The main idea of this method can be summarized by the following equation:

$$\widehat{\mathbb{P}}(y \mid x, q) = \widehat{\mathbb{P}}(y \mid x, \widehat{y}) = \begin{cases} \mathbb{I}[\widehat{y} = y] & x \in D_r \\ g(\mathcal{A}_w(x))_y & x \notin D_r \end{cases}, \tag{117}$$

where the $\widehat{y}$ denotes corresponding noisy label, and $D_r$ is a "reliable" set of noisy labels which are screened out using a specific method.

Specifically, such screening methods are usually carried out based on empirical observation that the neural networks tend to learn easy (correct) samples first, and then start to fit onto the hard (corrupt) samples in the later phase of training, hence the samples with small loss values are presumed to be reliable, while those with large loss values are not. The specific implementation of screening is diverse. Any screening method can be used in our framework to construct the corresponding $\widehat{S}$. In this paper, we will not describe these methods in more details.

**Construct of Main Experiments.** To avoid introducing more details into the methods in the main paper, the $\widehat{S}$ selected in the main experiments of this paper only adopts the simplest form and does not use any screening mechanisms that have been proven to be effective. We simply take $\widehat{S}$ as $\widehat{S}(x)_{y,q} = \widehat{\mathbb{P}}(y \mid x, q) = g(\mathcal{A}_w(x))_y$ for any $x$. More experiment of using different $\widehat{S}$ can be referred to in the Table 11. For the sake of simplicity in this experiment, the first way to construct S is to use a fixed real noise matrix. The second way to construct $S$ is the same as the method used in our main paper. In the third approach, we construct $\widehat{\mathbb{P}}(y \mid x, q)$ as a convex combination of the original noisy label and the model predictions, with the weights predicted by a two-component Gaussian mixture model. This method is a commonly used label bootstrap technique, as demonstrated by (Huang et al., 2023). The second method achieves the best results, as it leverages both the overall environmental information and the sample information more effectively. Additionally, due to the accuracy of clean sample screening in the early learning stage, the third method achieves performance similar to the second method. In contrast, using the inverse of the noise transition matrix alone, even with the true noise transition matrix, leads to suboptimal results because it neglects the sample information.

### F.2. Details of Construction of Semantic Similarity under Partial Label Setting

**Using Environment Information.** For instance-independent partial label setting, unlike the situation of noisy label, it is generally quite difficult to directly estimate the transition matrix $T$ due to the curse of dimensionality. However, when using the commonly assumption that labels are independent adopted into the candidate set (Cour et al., 2011; Lv et al., 2020; Feng et al., 2020b), the problem can be simplified. We formally present this assumption as follows:

**Assumption F.1.** (Selected candidate at uniform and independent (SCUI) assumption) The partial label problem satisfies the SCUI assumption if its generation process meets follow equation:

$$\begin{aligned} \mathbb{P}(q \mid x, y^i) &= \mathbb{I}[y^i \in q] \prod_{y \in \mathcal{Y}, y \neq y^i} \mathbb{P}(y \notin q \mid x, y^i)^{\mathbb{I}[y \notin q]} \mathbb{P}(y \in q \mid x, y^i)^{\mathbb{I}[y \in q]} \\ &= \mathbb{I}[y^i \in q] \prod_{y \in \mathcal{Y}, y \neq y^i} (1 - \sigma_y)^{\mathbb{I}[y \notin q]} \sigma_y^{\mathbb{I}[y \in q]}, \end{aligned} \tag{118}$$

where $\mathbb{P}(y \in q \mid x, y^i) = \mu_y$ for any $y \neq y^i$.

Giving this assumption, we can decompose the partial label problem into $c$ times independent binary classification noisy label problems. When each $\sigma_y$ in known, it is easy to see follow $S$ will satisfy $ST = I_{c \times c}$:

$$S_{y,q} = \begin{cases} 1 & y \in q \\ \frac{-\sigma_y}{1 - \sigma_y} & y \notin q \end{cases}, \tag{119}$$

Hence, if we can get estimated $\widehat{\sigma}_y$ for any $y \in \mathcal{Y}$, we can take $\widehat{S}_{y,q}$ through Equation 119.

Now we consider how to estimate this $\sigma_y$. We assume a uniform class distribution and encode $q = l \in [0,1]^c$, where $l_i = 1$ indicates $y^i \in q$. The following equation holds:

$$
\begin{aligned}
\mathbb{P}(x \mid l_i = 1) &= \frac{\mathbb{P}(x, Y = y^i, l^i = 1) + \mathbb{P}(x, Y \neq y^i, l^i = 1)}{\mathbb{P}(l_i = 1)} \\
&= \frac{(1/c)\mathbb{P}(x \mid Y = y^i) + (1 - 1/c)\sigma_i \mathbb{P}(x \mid Y \neq y^i)}{(1/c) + (1 - 1/c)\sigma_i} \\
&= \frac{(1/c)\mathbb{P}(x \mid Y = y^i) + (1 - 1/c)\sigma_i \mathbb{P}(x \mid l^i = 0)}{(1/c) + (1 - 1/c)\sigma_i} \\
&= (1 - \theta_i)\mathbb{P}(x \mid Y = y^i) + \theta_i \mathbb{P}(x \mid l^i = 0),
\end{aligned}
\tag{120}
$$

where $\theta_i = \frac{(c-1)\sigma_y}{1 + (c-1)\sigma_y}$.

Using Equation 120, we transform the problem of estimating $\sigma_i$ into the problem of estimating $\theta_i$. The later is a mixture proportion estimation problem to estimate $\theta_i$ given samples sampled from the $\mathbb{P}(x \mid l_i = 1)$ and $\mathbb{P}(x \mid l_i = 0)$. Under many assumptions to ensure identifiability, including the irreducibility assumption (Scott et al., 2013; Garg et al., 2021), the anchor point assumption (Scott, 2015; Liu and Tao, 2016), the separability assumption (Ramaswamy et al., 2016), $\theta_i$ can been estimated through many off-the-shelf methods (Scott et al., 2013; Scott, 2015; Ramaswamy et al., 2016; Garg et al., 2021).

**Using Both Environment and Samples Information.** Similar to the noisy label setting, the above method for constructing semantic similarity overlooks the information contained in the samples themselves. To address this, we can incorporate sample information by applying Bayes' theorem. The estimation of $\widehat{\mathbb{P}}(y \mid x, q)$ can then be expressed as follows:

$$
\widehat{\mathbb{P}}(y \mid x, q) \propto \widehat{\mathbb{P}}(y \mid x)\mathbb{P}(q \mid x, y) = \widehat{\mathbb{P}}(y \mid x)\frac{\prod_{y' \in q}\sigma_{y'}\prod_{y' \notin q}(1 - \sigma_{y'})}{\sigma_y}\mathbb{I}[y \in q] \propto \frac{g(\mathcal{A}_w(x))_y}{\sigma_y}\mathbb{I}[y \in q].
\tag{121}
$$

Such a construction method can be regarded as constructing semantic similarity by integrating the information of the current neural network, the sample information and the estimated environmental information. Therefore, better performance has been achieved in practice. When $\sigma_y$ is unknown, we can simply assume that $\sigma_y = \sigma$ for any $y \in \mathcal{Y}$, Equation 121 can still be used to construct semantic similarity.

**Using Samples Information.** In the instance-dependent partial label setting proposed in recent years (Xu et al., 2021; Qiao et al., 2023a), the SCUI assumption does not hold, the methods mentioned above become invalid. To estimate $\widehat{\mathbb{P}}(y \mid x, q)$ in this situation, Xu et al. 2021 proposes a variational label enhancement method which relies solely on current neural network. Additionally, Qiao et al. 2023a parameterizes $\widehat{\mathbb{P}}(q \mid x, y)$ and uses maximum likelihood estimation to learn these parameters. Given this estimation $\widehat{\mathbb{P}}(q \mid x, q)$, we can also estimate $\widehat{\mathbb{P}}(y \mid x, q)$ using the Bayes theorem: $\widehat{\mathbb{P}}(y \mid x, q) \propto \widehat{\mathbb{P}}(y \mid x)\widehat{\mathbb{P}}(q \mid x, y)$.

**Construct of Main Experiments.** To avoid introducing more details into the methods in the main paper, the $\widehat{S}$ selected in the main experiments of this paper only adopts the simplest form. We simply assume that $\sigma_y = \sigma$ for any $y \in \mathcal{Y}$ and apply Equation 121. Specifically, we take $\widehat{S}(x)_{y,q} = \widehat{\mathbb{P}}(y \mid x, q) = \mathbb{I}[y \in q]\frac{g(\mathcal{A}_w(x))_y}{\sum_{y' \in q} g(\mathcal{A}_w(x))_{y'}}$ for any $x$. More experiments using different $\widehat{S}$ can be found in Table 11. For the sake of simplicity in this experiment, the first way to construct $S$ is to use a fixed real partial ratio. The second way to construct $S$ is the same as the method used in our main paper. In the third approach, we construct $\widehat{\mathbb{P}}(y \mid x, q)$ by using variational label enhancement method proposed in (Xu et al., 2021). The second

*Table 11.* Comparisons with each methods for constructing $\widehat{S}$ on CIFAR-100 with different ratio of noisy label and partial label. We report the mean and std values of last 5 epochs. *Env.*, *Sap.* denote the environment information and sample information, respectively.

| Type | Noisy Label | | | Partial Label | | |
|---|---|---|---|---|---|---|
| Ratio | 0.5 | 0.8 | 0.9 | 0.05 | 0.1 | 0.2 |
| WSC w/ *Env.* | 75.29±0.15 | 67.34±0.34 | 55.01±0.14 | 76.31±0.25 | 75.85±0.54 | 71.33±1.33 |
| WSC w/ *Env.* & *Sap.* | 77.51±0.07 | 71.92±0.17 | 61.32±0.15 | 77.88±0.19 | 77.26±0.30 | 75.13±0.24 |
| WSC w/o *Env.* | 76.83±0.91 | 71.18±0.75 | 59.58±0.38 | 78.01±0.45 | 76.88±0.25 | 74.50±0.25 |

method achieves the best results, as it leverages both the overall environmental information and the sample information. Additionally, because Xu et al. 2021 also uses the sample information, hence achieves performance similar to the second method. In contrast, using the environment information alone, even with the true $\sigma$, leading to suboptimal results because it neglects the sample information.

