# OpenReview forum: "Weakly-Supervised Contrastive Learning for Imprecise Class Labels"
_ICML.cc/2025/Conference — ICML 2025 spotlightposter_

### Official Review · Reviewer_LHXf · 2025-02-26

**Overall Recommendation:** 4

**Summary:**

This paper proposes a graph-theoretic framework for contrastive learning with weakly-supervised information. This framework is recognized as effective according to the superior results in noisy label learning and partial label learning by introducing the continuous semantics similarity to define positives and negatives. Theoretical analysis shows that the framework could approximate supervised contrastive learning under mild conditions.

**Claims And Evidence:**

Yes, this paper provides sufficient theorems and outstanding experimental results to support the convincing claims.

**Essential References Not Discussed:**

The key references are provided to follow the main idea of the paper, and I have no essential references to provide.

**Experimental Designs Or Analyses:**

The experimental designs adopt two weakly-supervised learning paradigms and several well-known datasets to validate the superiority of the model. However, the analyses are not in-depth enough.

**Methods And Evaluation Criteria:**

Yes, the proposed method adopts a novel view of contrastive learning by framing it as a graph spectral clustering problem, and the edge weights are constructed based on self-supervised connectivity and weakly-supervised information, thereby learning expressive weakly-supervised representations.

**Other Comments Or Suggestions:**

No, I have no other comments or suggestions.

**Other Strengths And Weaknesses:**

Strengths
- The technique is novel. This paper proposes a graph-theoretic framework for weakly-supervised contrastive learning, which mines the relationships between sample pairs by integrating the self-supervised and weakly-supervised information into the augmentation graph.
- The theoretical analysis is sound. This paper provides sound theorems to explain how to realize weakly-supervised contrastive learning, and offers a rigorous performance guarantee to show that the proposed framework can approximate supervised learning.
- The model performance is outstanding. Many SOTA models are adopted to compare with the proposed WSC, and the experiment results show the superiority of WSC in both noisy label learning and partial label learning.

Weaknesses
- Lack of in-depth analyses of experiment results. The analyses in Section 4.1 and Section 4.2 just focus on the performance improvements, but do not explain why the model has wonderful performance, e.g., how the model works in the weakly-supervised scenario. So more in-depth analyses should be provided.
- Some typos. There are some typos in the paper, for example, $\mathcal{D} and \mathcal{D}_{\mathcal{Q}} $ in line 625.

**Questions For Authors:**

- Question on hyperparameter settings. I find that there exist significant differences in the numerical settings of $alpha$ and $beta$, so I want to know how to balance the self-supervised and supervised information in Eq. (5), and find ideal $alpha$ and $beta$ jointly.

**Relation To Broader Scientific Literature:**

This paper is relative to the field of weakly-supervised learning. The excellent experiment results and rigorous theorems may contribute to the broader scientific literature.

**Theoretical Claims:**

I have checked all the theoretical claims and proofs, which are correct in my angle. However, there are still some typos in the proofs.

---

> ### Author Rebuttal · Authors · 2025-03-29
>
> Dear reviewer LHXf:
>
> Thanks for your valuable suggestions, we will try to address your concerns and we are eager to engage in a more detailed discussion with you.
>
> > **W1: Lack of in-depth analyses of experiment results...**
>
> - We analyze the effectiveness of the proposed method in detail in the theoretical understanding of Section 3. It can be roughly understood in the following order:
>   - **Supervised information improves representation learning.** Theorem 3.4 describes how much the quality of learned features can be improved when using accurate supervised information. It can be seen that the features learned by the graph constructed with supervised information usually have a smaller error upper bound than those learned by only self-connection.
>   - **Weakly supervised information can approximate supervised information.** Through Theorem 3.7, we try to show that the features estimated using weakly supervised information can approximate the features obtained by supervised information to a certain extent. This theorem also indirectly illustrates how the proposed method works in weakly supervised scenario.
> - Your proposal made us realize that there is a lack of some supplementary explanations between our theoretical analysis and experimental verification. We will emphasize the reasons why the model behind the theoretical results is valid in the experimental part when the revised version is uploaded.
>
> > **W2: Some typos.**
>
> We are very sorry for this. We will check typos carefully and correct them in the next revision.
>
> > **Q1: Question on hyperparameter settings.**
>
> The setting of the hyperparameters $\alpha$ and $\beta$ is related to the number of categories in the dataset. Denote $C$ is the number of class, setting the ratio of $\beta$ to $\alpha$ at $2C$ can achieve a promising performance in general.

---

### Official Review · Reviewer_kufB · 2025-03-11

**Overall Recommendation:** 4

**Summary:**

The paper introduces a graph-theoretic framework for weakly supervised contrastive learning, leveraging continuous semantic similarity to better utilize ambiguous supervisory signals from imprecise class labels. This approach enhances model performance on multiple benchmark datasets in noisy and partially labeled settings.

**Claims And Evidence:**

Yes

**Essential References Not Discussed:**

N/A

**Experimental Designs Or Analyses:**

The paper experimentally evaluates its weakly-supervised contrastive learning framework through extensive testing on noisy and partial label datasets (e.g., CIFAR-10 and CIFAR-100), demonstrating performance improvements over existing methods via quantitative metrics and t-SNE visualizations of learned representations.

However, for PLL, it is better to add the experiments on instance-dependent PLL data.

**Methods And Evaluation Criteria:**

Yes

**Other Comments Or Suggestions:**

Please refer to the weakness

**Other Strengths And Weaknesses:**

Strengths:
1. The paper introduces a graph-theoretic framework for weakly supervised contrastive learning named WSC.
2. Extensive experiments demonstrate that WSC consistently outperforms existing baselines across diverse noise levels.
3. The manuscript is clearly written and well-structured, ensuring ease of readability.

Weaknesses:
1. It looks like there's a discrepancy between the caption and the content of Table 1. The caption mentions that both the mean and standard deviation values are reported, but only the mean values are shown.
2. In Table 7, PiCO outperforms the proposed method with a significant performance advantage under the partial ratio setting of 0.1 on the CUB-200 dataset. The 5.89% performance advantage is remarkable. It would be beneficial to provide an explanation for why the proposed method performs poorly in this scenario.
3. For PLL, the authors lack of the validation on instance-dependent data on PLL.

**Questions For Authors:**

1. It looks like there's a discrepancy between the caption and the content of Table 1. The caption mentions that both the mean and standard deviation values are reported, but only the mean values are shown.
2. In Table 7, PiCO outperforms the proposed method with a significant performance advantage under the partial ratio setting of 0.1 on the CUB-200 dataset. The 5.89% performance advantage is remarkable. It would be beneficial to provide an explanation for why the proposed method performs poorly in this scenario.
3. For PLL, the authors lack of the validation on instance-dependent data on PLL.
4. Only one state-of-the-art (SOTA) method published in 2024 is adopted for comparison, while the others were published in 2022 or earlier. It would be better to adopt more SOTA methods as comparison methods to further validate the effectiveness of the proposed method.

**Relation To Broader Scientific Literature:**

N/A

**Theoretical Claims:**

I have gone through the theorem and found no obvious errors

---

> ### Author Rebuttal · Authors · 2025-03-29
>
> Dear reviewer kufB:
>
> Thank you for your valuable suggestions. We truly appreciate your review and are committed to addressing your concerns with care and attention. We sincerely look forward to engaging in a more in-depth discussion with you, as your insights are essential in helping us improve and refine our approach.
>
> > **W1 & Q1: There's a discrepancy between the caption and the content of Table 1.**
>
> We are very sorry for this, this is a typo on our part, and we will correct the inconsistency between the caption and the table in the next revision.
>
> >**W2 & Q2: In Table 7, PiCO outperforms the proposed method with a significant performance advantage under the partial ratio setting of 0.1 on the CUB-200 dataset. The 5.89% performance advantage is remarkable. It would be beneficial to provide an explanation for why the proposed method performs poorly in this scenario.**
>
> - In addition to using contrastive learning to learn similar representations for samples from the same class, PiCO also uses class prototype-based pseudo-labeling mechanism. The synergy between the contrastive learning and the above mechanism has been proven to be very effective in several fine-grained classification tasks [1].
> - Our approach avoids the need for additional complex techniques. By incorporating our contrastive loss into the training objective, we achieve an 8.69% improvement over GFWS in this setting, highlighting the potential for even greater performance gains.
> - Although no other complex techniques are used, our method achieves the best performance in most of the remaining experiments, which also shows the effectiveness of our method in most scenarios.
>
> Reference:
>
> [1] Partial label learning: Taxonomy, analysis and outlook. *Neural Networks* 161 (2023): 708-734.
>
> > **Q3: Lack of the validation on instance-dependent data on PLL.**
>
> Following the works in [1, 2], we have expanded our experimental results to include the settings of biased tags associated with four fine-grained instance-dependent partial label datasets. The table below demonstrates that our method achieves marginally better performance than the current state-of-the-art (SOTA) approaches, and notably, without relying on any additional tricks or techniques. In the next version of our paper, we plan to incorporate more comprehensive comparative experiments to further validate our findings.
>
> |         | WSC(Our)  |  CEL[1]   | DIRK[2] | NEPLL[3] | IDGP[4] |
> | :-----: | :-------: | :-------: | :-----: | :------: | :-----: |
> | CUB200  | **69.10** |   68.60   |  66.60  |  62.88   |  58.16  |
> | CARS196 | **87.96** |   86.22   |  85.31  |  85.05   |  79.56  |
> | DOGS120 | **79.65** |   78.18   |  75.97  |  74.84   |  74.86  |
> | FGVC100 |   77.94   | **78.36** |  76.86  |  75.36   |  72.48  |
>
> The experimental details are shown in the table below ($\mathrm{resnet34}^{*}$ denote resnet34 pre-trained on Imagenet)：
>
> | Hyper-parameter | CUB200 | CARS196 | DOGS120 |FGVC100|
> | :---: | :---: | :---: | :---: | :---: |
> | Model | $\mathrm{resnet34}^{*}$ | $\mathrm{resnet34}^{*}$  | $\mathrm{resnet34}^{*}$ |$\mathrm{resnet34}^{*}$|
> | Image Size | 224 | 224 | 224 |224|
> | Batch Size | 256 | 256 | 256 |256|
> | Learning Rate | 0.1 | 0.1 | 0.1 |0.1|
> | Weight Decay | 1e-4 | 1e-4 | 1e-4 |1e-4|
> | LR Scheduler | Cosine | Cosine | Cosine |Cosine|
> | Training Epochs | 500 | 500 | 500 |500|
> | Classes | 200 |196 | 120 |100	|
> |$\alpha$|2|2|2|2|
> |$\beta$|0-400|0-400|0-240|0-200|
>
> > **Q4: Only one state-of-the-art (SOTA) method published in 2024 is adopted for comparison, while the others were published in 2022 or earlier.**
>
> - Recent advances in partial label learning have primarily concentrated on instance-dependent partial label problems, exemplified by several notable methodologies: CEL [1], a class-wise embedding-guided disambiguation approach; DIRK [2], a self-distillation-based label disambiguation framework; NEPLL [3], which employs normalized entropy for sample selection; and IDGP [4], a generative method modeling candidate label generation processes.
> - While these methods provide thorough investigations into label disambiguation mechanisms for instance-dependent scenarios, they notably neglect the critical exploration of leveraging ambiguous information for representation learning—a research gap addressed by our work. Comprehensive comparative experiments under instance-dependent partial label settings will be included in our subsequent manuscript version, accompanied by detailed discussions in related work sections.
>
> Reference:
>
> [1] Mixed Blessing: Class-Wise Embedding guided Instance-Dependent Partial Label Learning (KDD'2025)
>
> [2] Distilling Reliable Knowledge for Instance-Dependent Partial Label Learning. (AAAI'2024)
>
> [3] Candidate-aware Selective Disam-biguation Based On Normalized Entropy for Instance-dependent Partial-label Learning. (ICCV'23)
>
> [4] Decompositional Generation Process for Instance-Dependent Partial Label Learning. (ICLR'2023)

---

> > ### Comment · Reviewer_kufB · 2025-04-02
> >
> > Thanks author for the response. They have answered my questions well. I will raise my score.

---

> > > ### Author Response · Authors · 2025-04-02
> > >
> > > Dear Reviewer,
> > >
> > > We appreciate your invaluable suggestions for enhancing the quality and clarity of the paper. We will incorporate your comments in the next version of the paper. Thank you again for your time and effort!
> > >
> > > Best regards,
> > >
> > > The Authors

---

### Official Review · Reviewer_x5zp · 2025-03-17

**Overall Recommendation:** 4

**Summary:**

This paper tackles a key challenge in contrastive learning: handling real-world datasets with messy or ambiguous labels. The authors propose replacing traditional binary positive/negative pairs with "continuous semantic similarity," modeled via a graph where edge weights reflect how likely two examples belong to the same class. This framework integrates weak supervision (e.g., noisy/partial labels) into contrastive learning and shows strong empirical results across tasks like noisy label learning (NLL) and partial label learning (PLL).

**Claims And Evidence:**

1. Continuous similarity > binary labels for weak supervision
The graph-based approach replaces rigid class labels with similarity scores, allowing gradual refinement of supervision. Theoretical analysis (Prop 2.1, Thm 3.4) shows this approximates supervised contrastive learning under mild conditions.

2. Versatility across weak supervision settings
The proposed framework adapts to both NLL (noisy labels) and PLL (ambiguous candidate labels). Results on CIFAR-10/100, CUB-200, and Clothing1M show consistent gains, especially under high noise/ambiguity.

**Essential References Not Discussed:**

All essential references are discussed.

**Experimental Designs Or Analyses:**

yes

**Methods And Evaluation Criteria:**

- The proposed method builds a graph where nodes are augmented samples, and edge weights blend self-supervised similarity (from augmentations) and weakly supervised similarity (derived from labels/noise patterns).
- The proposed method is tested on standard NLL/PLL benchmarks (CIFAR, CUB-200, Clothing1M) against 10+ baselines under varying noise/partial-label ratios.

**Other Comments Or Suggestions:**

None.

**Other Strengths And Weaknesses:**

## Strengths
1. This paper directly addresses the "label quality bottleneck" in real-world data. The graph framework elegantly unifies self-supervised and weakly supervised signals.
2. This paper proposes to connect the method to spectral graph theory, providing error bounds (Thm 3.4) and showing how label information improves clustering.
3. Outperforms specialized methods (e.g., DivideMix for NLL, PiCO for PLL) even in high-noise regimes.

## Weaknesses
1. It seems this paper relies on uniform class distribution for theoretical guarantees—how does this hold for imbalanced data?
2. While t-SNE plots (Fig 2) suggest better representations, deeper qualitative insights are missing.

**Questions For Authors:**

1. How does the method handle class imbalance, given the uniform class assumption in theory?
2. What’s the computational overhead of building/maintaining the similarity graph compared to standard contrastive learning?
3. Are there scenarios where discrete labels would still outperform continuous similarity (e.g., clean labeled data)?

**Relation To Broader Scientific Literature:**

Weak supervision is very common in the field of machine learning literature, but contrastive learning with weak supervision has not been studied in prior works.

**Theoretical Claims:**

I checked the proof of Thm 3.4 which seems okay. I skimmed through Appendix C (proof of Section 3) but did not check thoroughly.

---

> ### Author Rebuttal · Authors · 2025-03-29
>
> Dear reviewer x5zp:
>
> Thank you for your valuable suggestions. We truly appreciate your review and are committed to addressing your concerns with care and attention. We sincerely look forward to engaging in a more in-depth discussion with you, as your insights are essential in helping us improve and refine our approach.
>
> > **W1 & Q1: It seems this paper relies on uniform class distribution for theoretical guarantees—how does this hold for imbalanced data?**
>
> **(1) Practical Applicability to Imbalanced Data:**
>
> - Our Algorithm 2 explicitly handles imbalanced scenarios by estimating $\mathcal{L}_{wsc}$ without any class prior assumption ($\mathbb{P}(\boldsymbol{y})$). This ensures practical validity under arbitrary class distributions. Algorithm 1 and Proposition 2.3 are merely simplified special cases under class balance, not prerequisites for imbalance handling.
>
> **(2) Theoretical Scope:**
>
> - The theoretical guarantee of unbiased representation learning under weak supervision (as formalized in Theorem 3.7: "Learning from Weak Supervision Approximates Supervised Learning") is inherently class-prior-invariant, requiring no assumptions about label distribution.  However, the downstream generalization analysis under supervised information Theorem 3.4 currently requires balanced assumptions for tractability. Hence, the adverse impacts of extreme long-tailed distributions on our framework remain unclear.
>   We will conduct further research on the framework's application under extreme long-tailed conditions in future work.
>
>
> >**W2: While t-SNE plots (Fig 2) suggest better representations, deeper qualitative insights are missing.**
>
> In addition to Figure 2, we also theoretically analyze that the proposed method helps improve the quality of representation learning features.
>
> - Theorem 3.4 bounds linear probe error of feature derived from perturbation graph through properties of self-supervised connectivity graph and perturbation coefficients, and demonstrates that the perturbation graph increases the density of intra-class connections without adding extra-class connections.
> - In Theorem 3.7, we further show that the features learned through finite samples and weakly supervised information can approximate the features learned from supervised information.
> - Finally, we present the linear probe error of the features learned by our framework in Corollary 3.8, which can be regarded as a qualitative analysis of the learned features.
>
> >**Q2: What’s the computational overhead of building/maintaining the similarity graph compared to standard contrastive learning?**
>
> Compared with standard contrastive learning, our additional computational overhead is mainly reflected in the single-step matrix multiplication used to construct semantic similarity, that is, $S((\tilde{x},\tilde{q}), (\tilde{x}',\tilde{q}'))$. This step of calculation does not bring too much additional computational cost and is negligible compared to standard contrastive learning. It can be considered that the computational overhead of our proposed method is basically the same as that of standard contrastive learning.
>
> > **Q3: Are there scenarios where discrete labels would still outperform continuous similarity (e.g., clean labeled data)?**
>
> - The concept of continuous semantic similarity we proposed is designed for weakly supervised learning, and our method is very effective under weakly supervised information empirically.
>
> - Theorem 3.7 explains the difference between the two features. The third and fourth terms on the right side of the inequality characterize the error induced by the usage of weakly supervised information, where the third term can be regarded as the estimated error term, and the fourth term describes the extent to which the weakly supervised information affects the learned representation. If the supervision information is completely accurate, it can be seen from the analysis that our proposed framework can be approximately equivalent to supervised contrastive learning.

---

### Official Review · Reviewer_jhEr · 2025-03-20

**Overall Recommendation:** 4

**Summary:**

This work rethinks contrastive learning for noisy real-world settings by replacing rigid class-based positive/negative sampling with adaptive, graph-driven “semantic similarity”. By blending self-supervised augmentations with weak supervision signals (e.g., noisy/partial labels), the method achieves state-of-the-art performance while offering theoretical guarantees.

**Claims And Evidence:**

Claim #1: Instead of treating labels as binary (same/different class), the authors model similarity as a continuous measure derived from label noise patterns or partial candidate sets.

Evidence #1: This avoids brittle assumptions about label correctness, which is crucial for messy real-world data. On Clothing1M (real-world noisy labels), the method achieves best performance, outperforming prior works as reported in experiments.

Claim #2: Augmented samples form nodes in a graph, with edges weighted by both augmentation-based similarity (e.g., two crops of an image) and label-derived signals (e.g., estimated noise patterns).

Evidence #2: By framing the problem through spectral graph theory, the authors prove the learned representations approximate supervised contrastive learning (Corollary 3.8), a novel theoretical bridge.

**Essential References Not Discussed:**

All essential references are discussed.

**Experimental Designs Or Analyses:**

yes

**Methods And Evaluation Criteria:**

Method: Constructs a graph where nodes are augmented samples, and edge weights combine (1) self-supervised similarity (via data augmentations) and (2) weakly supervised similarity (derived from label patterns or noise estimates). From this graph, a contrastive learning objective is derived by solving a spectral clustering problem.

Evaluation: Evaluated on multiple benchmarks spanning synthetic noise (CIFAR), real-world noise (Clothing1M), and fine-grained ambiguity (CUB-200). This paper compares with many strong baselines in LNL and PLL, which demonstrates its superiority.

**Other Comments Or Suggestions:**

None.

**Other Strengths And Weaknesses:**

**Strengths**

S1. Weak supervision is ubiquitous in real-world data, yet contrastive learning under such conditions remains underexplored. This work fills that void.
S2. Connects spectral graph theory with representation learning, providing error bounds while delivering strong empirical results.
S3. Outperforms specialized methods even in extreme noise regimes (e.g., 90% label noise on CIFAR-100).

**Weaknesses**

W1. Since contrastive learning is designed for improving representation learning, however except Figure 2, the paper does not provide sufficient evidence that shows its superiority in representation learning.

**Questions For Authors:**

Q1. Can the proposed method be applied to weakly-supervised multi-label learning problems?

**Relation To Broader Scientific Literature:**

The paper builds on prior work (i.e., Provable guarantees for self-supervised deep learning with spectral contrastive loss) and extends it to weak supervision tasks.

**Theoretical Claims:**

I have checked the proof of Proposition 2.3. and Corollary 3.8 which appears valid.

---

> ### Author Rebuttal · Authors · 2025-03-29
>
> Dear reviewer jhEr:
>
> Thank you for your valuable suggestions. We truly appreciate your review and are committed to addressing your concerns with care and attention. We sincerely look forward to engaging in a more in-depth discussion with you, as your insights are essential in helping us improve and refine our approach.
>
> > **W1. The paper does not provide sufficient evidence that shows its superiority in representation learning.**
>
> In addition to Figure 2, we also theoretically analyze that the proposed method helps improve the quality of representation learning features.
>
> - Theorem 3.4 bounds linear probe error of feature derived from perturbation graph through properties of self-supervised connectivity graph and perturbation coefficients, and demonstrates that the perturbation graph increases the density of intra-class connections without adding extra-class connections.
> - In Theorem 3.7, we further show that the features learned through finite samples and weakly supervised information can approximate the features learned from supervised information.
> - Finally, we present the linear probe error of the features learned by our framework in Corollary 3.8, which can be regarded as a qualitative analysis of the learned features.
>
> > **Q1. Can the proposed method be applied to weakly-supervised multi-label learning problems?**
>
> - The method we proposed is based on the semantic similarity between samples. The construction of semantic similarity depends on the transfer conditions approximately satisfied by each sample. It is not specially designed for multi-label classification tasks and cannot be simply applied to multi-label classification.
>
> - We believe that by further expanding the concept of semantic similarity, it is possible to apply our method to weakly-supervised multi-label learning problems. We will explore this problem in future work.

---

### Decision · Program_Chairs · 2025-05-01

**Decision:**

Accept (spotlight poster)

**Comment:**

This paper proposes WSC (Weakly-Supervised Contrastive Learning), a graph-theoretic framework that replaces rigid binary positive/negative sampling in contrastive learning with continuous semantic similarity. By blending self-supervised augmentations with weak supervision signals (e.g., noisy/partial labels), WSC achieves state-of-the-art performance in noisy label learning (NLL) and partial label learning (PLL). Theoretical analysis shows WSC approximates supervised contrastive learning under mild conditions.

WSC makes a methodological contribution by bridging graph theory and weak supervision, with strong empirical/theoretical validation. The rebuttal strengthened the paper’s rigor and scope, addressing all major critiques.